# In-context Learning for Mixture of Linear Regression: Existence, Generalization and Training Dynamics

**Yanhao Jin**  *yahjin@ucdavis.edu*
*Department of Statistics*
*University of California, Davis*

**Krishnakumar Balasubramanian**  *kbala@ucdavis.edu*
*Department of Statistics*
*University of California, Davis*

**Lifeng Lai**  *lflai@ucdavis.edu*
*Department of Electrical and Computer Engineering,*
*University of California, Davis*

**Reviewed on OpenReview:** *https://openreview.net/forum?id=buZXVuTsHY*

## Abstract

We investigate the in-context learning capabilities of transformers for the $d$-dimensional mixture of linear regression model, providing theoretical insights into their existence, generalization bounds, and training dynamics. Specifically, we prove that there exists a transformer capable of achieving a prediction error of order $\mathcal{O}(\sqrt{d/n})$ with high probability, where $n$ represents the training prompt size in the high signal-to-noise ratio (SNR) regime. Moreover, we derive in-context excess risk bounds of order $\mathcal{O}(L/\sqrt{B})$ for the case of two mixtures, where $B$ denotes the number of training prompts, and $L$ represents the number of attention layers. The dependence of $L$ on the SNR is explicitly characterized, differing between low and high SNR settings. We further analyze the training dynamics of transformers with single linear self-attention layers, demonstrating that, with appropriately initialized parameters, gradient flow optimization over the population mean square loss converges to a global optimum. Extensive simulations suggest that transformers perform well on this task, potentially outperforming other baselines, such as the Expectation-Maximization algorithm.

## 1 Introduction

We investigate the in-context learning (ICL) ability of transformers for the Mixture of Regression (MoR) model (De Veaux, 1989; Jordan & Jacobs, 1994). The MoR model is widely applied in various domains, including federated learning, collaborative filtering, and healthcare (Deb & Holmes, 2000; Viele & Tong, 2002; Kleinberg & Sandler, 2008; Faria & Soromenho, 2010; Ghosh et al., 2020) to address heterogeneity in data, often arising from multiple data sources. In particualr, we consider linear MoR models where independent and identically distributed samples $(x_i, y_i) \in \mathbb{R}^d \times \mathbb{R}$, for $i = 1, \ldots, n$, are assumed to follow the model $y_i = \langle \beta_i, x_i \rangle + v_i$, where $v_i \sim \mathcal{N}(0, \vartheta^2)$ represents observation noise, independent of $x_i$, and $\beta_i \in \mathbb{R}^d$ is an unknown regression vector. Specifically, there are $K$ distinct regression vectors $\{\beta_k^*\}_{k=1}^K$, and each $\beta_i$ is independently drawn from these vectors according to the distribution $\{\pi_k^*\}_{k=1}^K$. The goal for a new test sample, $x_{n+1}$, is to predict its label $y_{n+1}$. Specifically, we are interested in the ICL setup for MoR (Kong et al., 2020; Pathak et al., 2024).

Classically, the Expectation Maximization (EM) algorithm is a widely used method for estimation and prediction in the MoR models (Balakrishnan et al., 2017; Kwon et al., 2019; Kwon & Caramanis, 2020; Wang et al., 2024). A major limitation of the EM algorithm is its tendency to converge to local maxima rather than the global maximum of the likelihood function. This issue arises because the algorithm's performance

crucially depends on the initialization (Jin et al., 2016). To mitigate this, favorable initialization strategies based on spectral methods (Chaganty & Liang, 2013; Zhang et al., 2016; Chen et al., 2020) are typically employed alongside the EM algorithm.

In an intriguing recent work, through a mixture of theory and experiments, Pathak et al. (2024) examined the performance of transformers for ICL MoR models. However, their theoretical result suffers from the following major drawback. They only showed that the existence of a transformer architecture that is capable of implementing the *oracle* Bayes optimal predictor for the linear MoR problem. That is, they assume the availability of $\{\beta_k^*\}_{k=1}^K$, which are in practice unknown and are to be estimated. Hence, there remains a gap in the theoretical understanding of how transformers actually perform parameter estimation and prediction in MoR. Furthermore, their theoretical result is rather disconnected from their empirical observations which focused on ICL. Indeed, they leave open a theoretical characterization of the problem of ICL MoR (Pathak et al., 2024, Section 4).

In this work, we first demonstrate that transformers are capable (in an existence sense) of in-context learning for linear MoRs by effectively implementing the EM algorithm, a double-loop algorithm in which each inner loop consists of multiple steps of gradient ascent. We derive in-context excess risk bounds for the global solution of the empirical in-context risk minimizer, precisely quantifying the number of pre-training tasks required to achieve accurate predictions. While the aforementioned existence result and in-context excess risk bounds provide insight into capability of transformers for in-context learning MoR modes, from the practical point of view, it is more important to understand the training dynamics of transformers under the MoR models. Towards that, we also analyze the performance of gradient flow (in the population setting) for ICL MoR models with linear self-attention transformers. Furthermore, through our experiments, we empirically show that trained transformers achieve efficient prediction and estimation in the MoR model while substantially mitigating the initialization challenges typically associated with the EM algorithm. To summarize, we make the following **contributions**:

- We demonstrate the existence of a transformer capable of learning MoR models by implementing the dual-loops of the EM algorithm. This construction involves the transformer performing multiple gradient ascent steps during each M-step of the EM algorithm. In Theorem 3.1, we derive precise bounds on the transformer's ability to make prediction in high signal-to-noise (SNR) regimes. In the special case of two mixtures, Theorem 3.2 also provides the precise high-probability bound for the estimation of the parameters by the constructed transformer in the high and low-SNR settings.

- In Theorem 4.1, we analyze the sample complexity associated with pretraining transformers using a finite number of ICL training instances. Additionally, Theorem 4.2 provides guarantee that the gradient flow of the parameters of single linear self-attention layers will eventually converge to the global optimum under population mean squared loss with appropriate initializations.

- As a byproduct of our analysis, we also derive convergence results with statistical guarantees for the gradient EM algorithm applied to a two-component mixture of regression models, where the M-step involves $T$ steps of gradient ascent. We extend this approach to the multi-component case, improving upon previous works, such as Balakrishnan et al. (2017), which considered only a single step of gradient ascent.

## 1.1 Related works

**Transformers and optimization algorithms:** Garg et al. (2022) successfully demonstrated that transformers can be trained to perform ICL for linear function classes, achieving results comparable to those of the optimal least squares estimator. Beyond their empirical success, numerous studies have sought to uncover the mechanisms by which transformers facilitate ICL. Recent investigations suggest that transformers may internally execute first-order Gradient Descent (GD) to perform ICL, a concept explored in depth by Akyürek et al. (2023), Bai et al. (2024), Von Oswald et al. (2023a), Von Oswald et al. (2023b), Ahn et al. (2024), Huang et al. (2024) and Zhang et al. (2024). Specifically, Akyürek et al. (2023) identified fundamental operations that transformers can execute, such as multiplication and affine transformations, showing that transformers can implement GD for linear regression using these capabilities. Building on

this, Bai et al. (2024) provided detailed constructions illustrating how transformers can implement convex risk minimization across a wide range of standard machine learning problems, including least squares, ridge, lasso, and generalized linear models (GLMs). Further, Ahn et al. (2024) demonstrated that a single-layer linear transformer, when optimally parameterized, can effectively perform a single step of preconditioned GD. Zhang et al. (2024) expanded on this by showing that every one-step GD estimator, with a learnable initialization, can be realized by a linear transformer block (LTB) estimator.

Moving beyond first-order optimization methods, Fu et al. (2023) revealed that transformers can achieve convergence rates comparable to those of the iterative Newton's method, which are exponentially faster than GD, particularly in the context of linear regression. These insights collectively highlight the sophisticated computational abilities of transformers in ICL, aligning closely with classical optimization techniques. In addition to exploring how transformers implement these mechanisms, recent studies have also focused on their training dynamics in the context of linear regression tasks; see, for example, Zhang et al. (2023) and Chen et al. (2024). In comparison to the aforementioned works, in the context of MoR, we demonstrate that transformers are capable of implementing double-loop algorithms such as the EM algorithm.

**EM Algorithm:** The analysis of the standard EM algorithm for mixture of Gaussian and linear MoR models has a long-standing history Wu (1983); McLachlan & Krishnan (2007); Tseng (2004). Recently, Balakrishnan et al. (2017) proved that EM algorithm converges at a geometric rate to a local region close to the maximum likelihood estimator with explicit statistical and computational rates of convergence. Subsequent works (Kwon et al., 2019; 2021) established improved convergence results for mixture of regression under different SNR conditions. Kwon & Caramanis (2020) extended these results to mixture of regression with many components. Gradient EM algorithm was first analyzed by Wang et al. (2015) and Balakrishnan et al. (2017). It is an immediate variant of the standard EM algorithm where the M-step is achieved by one-step gradient ascent rather than exact maximization. They proved that the gradient EM also can achieve the local convergence with explicit finite sample statistical rate of convergence. Global convergence for the case of two-components mixture of Gaussian model was show by Xu et al. (2016), Daskalakis et al. (2017) and Wu & Zhou (2021). The case of unbalanced mixtures was handled by Weinberger & Bresler (2022). Penalized EM algorithm for handling high-dimensional mixture models was analyzed by Zhu et al. (2017), Yi & Caramanis (2015) and Wang et al. (2024), showing that gradient EM can achieve linear convergence to the unknown parameter under mild conditions.

## 2 Preliminaries

**Mixture of regression model:** We now formally describe the MoR problem. The underlying true model is described by the equation:

$$y_i = x_i^\top \beta_i + v_i, \tag{1}$$

where $x_i \sim \mathcal{N}(0, I_d)$, $v_i \sim \mathcal{N}(0, \vartheta^2 I_d)$ denotes the noise term with variance $\vartheta^2$, and $\beta_i$'s are i.i.d. random vectors that taking the value $\beta_k^*$ with probability $\pi_k^*$ for $k = 1, \ldots, K$. The vectors $\beta_k^*$ are unknown. For the MoR model equation 1, we define $R_{ij}^* = \left\| \beta_i^* - \beta_j^* \right\|_2$ as pairwise distance between regression vectors, and $R_{\min} = \min_{i \neq j} R_{ij}^*$, $R_{\max} = \max_{i \neq j} R_{ij}^*$ as the smallest and largest distance respectively. The SNR of this problem is defined as the ratio of minimum pairwise distance versus standard deviation of noise

$$\eta := R_{\min}/\vartheta. \tag{2}$$

When the number of the components $K = 2$ and we represent $\beta_1^* = -\beta_2^* := \beta^*$, the SNR reduces to $\eta = 2\|\beta^*\|_2/\vartheta$. In Section 3, we will show that the performance of the constructed transformer solving the MoR problem in general depends on the SNR condition of the problem.

**Transformer architecture:** We focus on transformers that handle the input sequence $H \in \mathbb{R}^{D \times N}$ by integrating attention layers and multi-layer perceptrons (MLPs). These transformers are structured to process the input by effectively mapping the complex interactions and dependencies between data points in the sequence, utilizing the capabilities of attention mechanisms to dynamically weigh the importance of different features in the context of regression analysis.

**Definition 2.1.** An attention layer with $M$ heads is denoted as $\text{Attn}_{\boldsymbol{\theta}}(\cdot)$ with parameters $\boldsymbol{\theta} = \{(V_m, Q_m, K_m)\}_{m \in [M]} \subset \mathbb{R}^{D \times D}$. On any input sequence $H \in \mathbb{R}^{D \times N}$, we have

$$\widetilde{H} = \text{Attn}_{\boldsymbol{\theta}}(H) := H + \frac{1}{N} \sum_{m=1}^{M} (V_m H) \times \sigma\big((Q_m H)^\top (K_m H)\big) \in \mathbb{R}^{D \times N}$$

where $\sigma : \mathbb{R} \to \mathbb{R}$ is the activation function and $D$ is the hidden dimension. In the vector form,

$$\tilde{h}_i = h_i + \sum_{m=1}^{M} \frac{1}{N} \sum_{j=1}^{N} \sigma\big(\langle Q_m h_i, K_m h_j \rangle\big) \cdot V_m h_j.$$

**Remark 2.1.** The prevalent choices for the activation function include the softmax function and the ReLU function. In our analysis in Section 3, the attention layer (defined in Section 2.1) employs a normalized ReLU activation, $t \mapsto \sigma(t)/N$, which is used for technical convenience. This modification does not impact the fundamental nature of the study. There are various works on transformers that use non-softmax function as activation, (e.g. Shen et al. (2023), Bai et al. (2024) for ReLu activation, Luo et al. (2021) for kernel attention, Qin et al. (2022) for cosFormer).

**Definition 2.2** (Attention only transformer)**.** An $L$-layer transformer, denoted as $\text{TF}_\theta(\cdot)$, is a composition of $L$ self-attention layers,

$$\text{TF}_\theta(\cdot) = \text{Attn}_{\theta^L} \circ \text{Attn}_{\theta^{L-1}} \circ \cdots \circ \text{Attn}_{\theta^1}(H)$$

where $H \in \mathbb{R}^{D \times N}$ is the input sequence, and the parameter $\boldsymbol{\theta} = (\theta^1, \ldots, \theta^L)$ consists of the attention layers $\theta^{(\ell)} = \{(V_m^{(\ell)}, Q_m^{(\ell)}, K_m^{(\ell)})\}_{m \in [M^{(\ell)}]} \subset \mathbb{R}^{D \times D}$.

Our theory consists of two parts, (i) the existence of the theoretical transformer that can internally implement the EM algorithm, and (ii) the dynamics of transformers with a single linear self-attention layer trained by gradient flow on mixture of regression tasks. In the first part, the input sequence $H \in \mathbb{R}^{D \times (n+1)}$ has columns

$$\begin{aligned} h_i &= [x_i, y_i', \mathbf{0}_{D-d-3}, 1, t_i]^\top, \\ h_{n+1} &= [x_{n+1}, y_{n+1}', \mathbf{0}_{D-d-3}, 1, 1]^\top \end{aligned} \tag{3}$$

where $y_i' = y_i t_i$ and $t_i := \mathbb{1}\{i < n+1\}$ is the indicator for the training examples. Then the transformer $\text{TF}_\theta$ produces the output $\widetilde{H} = \text{TF}_\theta(H)$. The prediction $\hat{y}_{n+1}$ is derived from the $(d+1, n+1)$-th entry of $\widetilde{H}$, denoted as $\hat{y}_{n+1} = \text{read}_y(\widetilde{H}) := (\tilde{h}_{n+1})_{d+1}$. Our objective is to develop a fixed transformer architecture that efficiently conducts ICL for the mixture of regression problem, thereby providing a prediction $\hat{y}_{n+1}$ for $y_{n+1}$ under an appropriate loss framework. Besides, the constructed transformer in Section 2 can also extract an estimate of the regression components, which is realized by operator $\text{read}_\beta(\text{TF}(H)) = [\text{TF}(H)]_{d+2:2d+2, n+1}$ extracts the estimate of $\beta^*$ in the output matrix. In the second part, the embedded input matrix is given by

$$E = \begin{pmatrix} x_1 & x_2 & \cdots & x_n & x_{n+1} \\ y_1 & y_2 & \cdots & y_n & 0 \end{pmatrix} \in \mathbb{R}^{(d+1) \times (n+1)} \tag{4}$$

and is fed into a single-layer linear self-attention layer $f_{\text{LSA}} : \mathbb{R}^{(d+1) \times (n+1)} \to \mathbb{R}^{(d+1) \times (n+1)}$

$$f_{\text{LSA}}(E; \theta) = E + VE \cdot \frac{E^\top Q^\top K E}{n}, \tag{5}$$

where $\theta = \{K, Q, V\}$. The prediction on the query sample $x_{n+1}$ is given by the bottom-right entry of the matrix by $f_{\text{LSA}}$, i.e. $\hat{y}_{n+1} = [f_{\text{LSA}}(E; \theta)]_{d+1, n+1}$.

**Remark 2.2.** We highlight the key ideas of our work. In Section 3, Theorems 3.1 and 3.2 provide an explicit construction of a transformer that executes gradient-EM within its forward pass—an E-step followed by $T$ gradient updates in the M-step—along with the SNR and prompt-length regimes where the error bounds hold. This is an existence result. In Section 4, we pre-train transformers by minimizing MSE. Thus, the learning mechanism is not tied to our constructive parameterization and can match or surpass it. Our experiments report end-to-end excess MSE, showing the effectiveness of trained transformers, though this does not by itself certify that they use EM internally—even though they are capable of doing so.

**Remark 2.3.** Equation equation 3 is used only for theoretical convenience, not as a data-preprocessing recipe. It specifies a per-token "register" layout that lets the proof show how one attention layer can read/write quantities needed for an EM step, with zero blocks serving as scratch space and $t_i$ as position tags. In practice, we feed standard embeddings for $(x_i, y_i)$ and a masked query $x_{n+1}$, while a single input projection can create the scratch dimensions automatically. Writing the prompt as $E$ in equation 4 is just a compact reformatting of inputs (matching common implementations) for technical convenience and does not introduce new assumptions.

**Notation**: Given two functions $g(n)$ and $f(n)$, we say that $f(n) = \Omega(g(n))$, if there exist constants $c > 0$ and $n_0 >= 0$ such that $f(n) \geqslant c^*g(n)$ for all $n \geqslant n_0$. We say that $f(n)$ is $\mathcal{O}(g(n))$ if there exist positive constant $C$ and $n_0$ such that $0 \leqslant f(n) \leqslant Cg(n)$ for all $n \geqslant n_0$. For a vector $v \in \mathbb{R}^d$, its $\ell_2$ norm is denoted by $\|v\|_2$. For a matrix $A \in \mathbb{R}^{d \times d}$, $\|A\|_{\mathrm{op}}$ denotes the operator (spectral) norm of $A$. We denote the joint distribution of $(x, y)$ in model equation 1 by $\mathcal{P}_{x,y}$ and the distribution of $x$ by $\mathcal{P}_x$. Besides, we denote the joint distribution of $(x_1, y_1, \ldots, x_n, y_n, x_{n+1}, y_{n+1})$ by $\mathcal{P}$, where $\{x_i, y_i\}_{i=1}^n$ are the input in the training prompt and $x_{n+1}$ is the query sample. Besides, in Section 3, we use $y_i' \in \mathbb{R}$ defined as $y_i' = y_i t_i$ and $t_i = 1_{\{i < n+1\}}$ for $i = 1, \ldots, n, n+1$ to simplify our notation.

**Evaluation:** Let $f : H \mapsto \hat{y} \in \mathbb{R}$ be any procedure that takes a prompt $H$ as input and outputs an estimate $\hat{y}$ on the query $y_{n+1}$. We define the mean squared error (MSE) by $\mathrm{MSE}(f) := \mathbb{E}_{\mathcal{P}}\big[\big(f(H) - y_{n+1}\big)^2\big]$.

## 3 Existence of transformer for MoR

In this section, we show the existence of a transformer that can approximately implement the EM algorithm internally in Theorem 3.1 and Theorem 3.2. Note that under model Theorem 1, the oracle vector that minimizes the mean squared error of the prediction $\mathbb{E}_{\mathcal{P}}[(x_{n+1}^\top \beta - y_{n+1})^2]$ is given by

$$\beta^{\mathsf{OR}} := \arg\min_{\beta \in \mathbb{R}^d} \mathbb{E}_{\mathcal{P}_{x,y}}\big[(x_{n+1}^\top \beta - y_{n+1})^2\big] = \sum_{\ell=1}^K \pi_\ell^* \beta_\ell^*.$$

Generally, the transformer constructed in Theorem 3.1 will provide a prediction that is close to $x_{n+1}^\top \beta^{\mathsf{OR}}$.

**Theorem 3.1.** *Given the input matrix $H$ in the form of equation 3, there exists a transformer* $\mathrm{TF}$ *with the number of heads $M^{(\ell)} \leqslant M = 4$ in each attention layers. This transformer* $\mathrm{TF}$ *can make prediction on $y_{n+1}$ by implementing gradient EM algorithm of MoR problem where $T$ steps of gradient descent is used in each M-step. When $L$ is sufficiently large and the prompt length $n$ satisfies following condition*

$$n \geqslant C \max\left\{d \log^2 \frac{dK^2}{\delta}, \big(\frac{K^2}{\delta}\big)^{1/3}, \frac{d}{\pi_{\min}} \log\left(\frac{K^2}{\delta}\right)\right\},$$

*under the SNR condition*

$$\eta \geqslant CK\rho_\pi \log(K\rho_\pi), \quad \textit{for a sufficiently large } C > 0, \tag{6}$$

*equipped with $\mathcal{O}\big(T \log\big(n/d\big)\big)$ attention layers, the transformer has the prediction error $\Delta_y := |\mathrm{read}_y\big(\mathrm{TF}(H)\big) - x_{n+1}^\top \beta^{\mathsf{OR}}|$ upper-bounded by*

$$\mathcal{O}\left(\sqrt{\log(d/\delta)}\left(\sqrt{\frac{dK\rho_\pi^2}{n} \log^2\left(\frac{nK^2}{\delta}\right)} + \sqrt{\frac{dK \log(\frac{K^2}{\delta})}{n\pi_{\min}}}\right)\right),$$

*with probability at least $1 - 9\delta$, where $\rho_\pi = \max_j \pi_j^* / \min_j \pi_j^*$ is the ratio of maximum mixing weight and minimum mixing weight, $\pi_{\min} = \min_j \pi_j^*$ and $\mathrm{read}_y(\tilde{H}) := \big(\tilde{h}_{n+1}\big)_{d+1}$ extracts the prediction on query sample.*

Theorem 3.1 demonstrates the feasibility and theoretical guarantees of transformers in solving a general mixture of regression problems under the high SNR condition specified in equation 6. The error between

the transformer's prediction and the true response for the query sample is bounded with high probability in the order of $\sqrt{\log(d/\delta)\frac{d}{n}\log^2(n/\delta)}$. This error decreases as the prompt length $n$ increases and is affected by factors like the dimension $d$ and the number of components $K$. The ratio $\rho_\pi = \max_j \pi_j^* / \min_j \pi_j^*$ quantifies the imbalance in mixing proportions. Larger imbalances $\rho_\pi$ degrade the error bound, indicating that the transformer's performance could worsen when mixture components are highly unbalanced.

In the special case of MoR problems with two components $\beta_1^* = -\beta_2^* = \beta^*$ and $\pi_1^* = \pi_2^* = \frac{1}{2}$, we have $\beta^{\text{OR}} = 0$. While predicting zero is not quite meaningful, estimating the true regression coefficient vector $\beta^*$ is of interest. Hence, in Theorem 3.2 below, we provide more refined results focusing both on the low and high SNR regimes.

**Theorem 3.2.** *Given input matrix $H$ whose columns are given by equation 3, there exists a transformer $\text{TF}_\theta$, with the number of heads $M^{(\ell)} \leqslant M = 4$ in each attention layers, that can make prediction on $y_{n+1}$ by implementing gradient EM algorithm of MoR problem where $T$ steps of gradient descent are used in each M-step. When $T$ is sufficiently large and the prompt length $n$ satisfies*

$$n \geqslant C d \log^2\left(1/\delta\right), \tag{7}$$

*the transformer can approximates $\beta^*$ by the second-to-last layer with probability at least $1 - \delta$:*

- *When $\eta \leqslant C\left(d\log^2 n/n\right)^{\frac{1}{4}}$, $\|\operatorname{read}_\beta\left(\text{TF}(H)\right) - \beta^*\|_2 \leqslant \mathcal{O}\left(\left(d\log^2(n/\delta)/n\right)^{\frac{1}{4}}\right)$;*

- *When $\eta \geqslant C\left(d\log^2 n/n\right)^{\frac{1}{4}}$, then $\|\operatorname{read}_\beta\left(\text{TF}(H)\right) - \beta^*\|_2 \leqslant \mathcal{O}\left(\sqrt{d\log^2(n/\delta)/n}\right)$;*

*where $\operatorname{read}_\beta(\text{TF}(H)) = \left[\text{TF}(H)\right]_{d+2:2d+2,n+1}$ extracts the estimate of $\beta^*$.*

In Theorem 3.2, the error depends on the SNR, $\eta$, and exhibits two distinct behaviors: In low SNR settings, the error scales as $\mathcal{O}\left((d\log^2(n/\delta)/n)^{1/4}\right)$, reflecting the inherent difficulty of recovering $\beta^*$ in noisy environments. In high SNR settings, the error scales as $\mathcal{O}\left(\sqrt{d\log^2(n/\delta)/n}\right)$, showing better performance due to stronger signals dominating the noise. According to Theorem 3.2, the architecture of the constructed transformer varies primarily in the number of layers it includes. In general, with the prompt length $n$ and dimension $d$ held constant, the constructed transformer needs more training samples in the prompt in the low SNR settings to achieve the desired precision. The prediction error is order of $\tilde{\mathcal{O}}(\sqrt{d/n})$ under the high SNR settings, and is $\tilde{\mathcal{O}}((d/n)^{\frac{1}{4}})$ in the low SNR settings. Besides, under the high SNR settings, the constructed transformer needs $\mathcal{O}\left(\log(n/d)\right)$ attention layers, while it needs $\mathcal{O}\left(\sqrt{n/d}\log(\log(n/d))\right)$ attention layers in the low SNR settings.

The proof of Theorem 3.1 is provided in Appendix A.7 and details of the proof of Theorem 3.2 can be found in Appendix A.5. In Theorem 3.1 and 3.2, the variance parameter $\vartheta$ in equation 1 and equation 2 in the noise is assumed to be a fixed known constant, since it is used in the construction of the transformer. The SNR condition required in Theorem 3.1 is stricter than that in Theorem 3.2 due to technical reasons in the proof. However, in our simulations (presented in Figure 1 in Section 5), we see that the actual performance of the transformer is still good in the low SNR scenario when the number of components $K \geqslant 3$.

Finally, in Theorem 3.3, we provide the excess risk bound for the transformer constructed in Theorem 3.2.

**Theorem 3.3.** *For any $T$ being sufficiently large and the prompt length $n$ satisfies condition equation 7. Define the excess risk $\mathcal{R} := \mathbb{E}_\mathcal{P}\left[(y_{n+1} - \operatorname{read}_y(\text{TF}(H)))^2\right] - \inf_\beta \mathbb{E}_\mathcal{P}\left[(x_{n+1}^\top \beta - y_{n+1})^2\right]$. Then the ICL prediction $\operatorname{read}_y(\text{TF}(H))$ of the constructed transformer in Theorem 3.2 satisfies*

$$\mathcal{R} = \begin{cases} \mathcal{O}\left(\sqrt{d\log^2 n/n}\right) & 0 < \eta \leqslant C\left(d\log^2(n/\delta)/n\right)^{1/4} \\ \mathcal{O}\left(d\log^2 n/n\right) & \eta \geqslant C\left(d\log^2(n/\delta)/n\right)^{1/4} \end{cases}. \tag{8}$$

*Furthermore, $\inf_\beta \mathbb{E}_\mathcal{P}\left[(x_{n+1}^\top \beta - y_{n+1})^2\right] = \vartheta^2 + \|\beta^*\|_2^2$.*

Theorem 3.1, Theorem 3.2 and Theorem 3.3 provide the first quantitative framework for end-to-end ICL in the mixture of regression problems, achieving desired precision. The excess risk of the constructed transformer is $\mathcal{O}(d\log^2 n/n)$ under the high SNR settings, and is $\mathcal{O}(\sqrt{d/n}\log n)$ under the low SNR settings. These results represent an advancement over the findings in Pathak et al. (2024), which do not offer explicit error bounds like equation 8.

The transition on the error rate appears in Theorem 3.2 due to the switch of the analysis regime. In high SNR $\eta \gtrsim \left(d\log^2(n/\delta)/n\right)^{1/4}$ : the population EM map is strongly contractive (curvature $\approx \eta^2$ ). Balancing this contraction with sampling noise ( $\approx \sqrt{d\log^2(n/\delta)/n}$) gives the parametric rate $O(\sqrt{d\log^2(n/\delta)/n})$. In low SNR $\eta \lesssim \left(d\log^2(n/\delta)/n\right)^{1/4}$ : the map is nearly flat; progress is driven by a cubic drift, yielding an $n^{-1/4}$-scale neighborhood: $O\left(\left(d\log^2(n/\delta)/n\right)^{1/4}\right)$ The threshold at $\eta \asymp (d\log^2(n/\delta)/n)^{1/4}$ is exactly where curvature $(\propto \eta^2)$ overtakes sampling error. Plugging this $\eta$ into the high-SNR bound recovers the $n^{-1/4}$ order, so the two regimes meet smoothly.

## 4 Understanding Transformer Training on MoR tasks

### 4.1 Analysis of pre-training

We now analyze the sample complexity needed to pretrain the transformer with a limited number of ICL training instances. Prior results from Bai et al. (2024) are only applicable to linear models and are not immediately applicable to the linear MoR models that we focus on in this work. We consider the square loss between the in-context prediction and the ground truth label:

$$\ell_{\text{icl}}(\boldsymbol{\theta}; \mathbf{Z}) := \frac{1}{2}\Big[y_{n+1} - \text{clip}_R\left(\text{read}_y\left(\text{TF}_{\boldsymbol{\theta}}(H)\right)\right)\Big]^2,$$

where $\mathbf{Z} := \left(H, y_{n+1}\right)$ is the training prompt, $\boldsymbol{\theta} = \left\{(K_m^{(\ell)}, Q_m^{(\ell)}, V_m^{(\ell)}) : \ell = 1, \ldots, L, m = 1, \ldots, M\right\}$ is the collection of parameters of the transformer and $\text{clip}_R(t) := \text{Proj}_{[-R,R]}(t)$ is the standard clipping operator with (a suitably large) radius $R \geqslant 0$ that varies in different problem setups to prevent the transformer from blowing up on tail events, in all our results concerning (statistical) in-context prediction powers. Additionally, the clipping operator can be employed to control the Lipschitz constant of the transformer $\text{TF}_{\boldsymbol{\theta}}$ with respect to $\boldsymbol{\theta}$ (see Bai et al. (2024)). In practical applications, it is common to select a sufficiently large clipping radius $R$ to ensure that it does not alter the behavior of the transformer on any input sequence of interest. Applying clipping operator on the objective functions when training LLM is used in RLHF (see Ziegler et al. (2019), Ouyang et al. (2022)). Denote $\|\boldsymbol{\theta}\|$ as the norm of transformer given by

$$\|\boldsymbol{\theta}\| := \max_{\ell\in[L]}\left\{\max_{m\in[M]}\left\{\|Q_m^{(\ell)}\|_{\text{op}}, \|K_m^{(\ell)}\|_{\text{op}}\right\} + \sum_{m=1}^{M}\|V_m^{(\ell)}\|_{\text{op}}\right\}.$$

Our pretraining loss is the average ICL loss on $B$ pretraining instances $\mathbf{Z}^{(1:B)} \overset{\text{iid}}{\sim} \pi$, and we consider the corresponding test ICL loss on a new test instance:

$$\hat{L}_{\text{icl}}(\boldsymbol{\theta}) := \frac{1}{B}\sum_{j=1}^{B}\ell_{\text{icl}}\left(\boldsymbol{\theta}; \mathbf{Z}^{(j)}\right),$$

$$L_{\text{icl}}(\boldsymbol{\theta}) := \mathbb{E}_{\mathcal{P}}\left[\ell_{\text{icl}}\left(\boldsymbol{\theta}; \mathbf{Z}\right)\right].$$

Our pretraining algorithm is to solve a standard constrained empirical risk minimization problem over transformers with $L$ layers, $M$ heads, and norm bounded by $M'$:

$$\hat{\boldsymbol{\theta}} := \arg\min_{\boldsymbol{\theta}\in\Theta_{M'}} \hat{L}_{\text{icl}}(\boldsymbol{\theta}), \tag{9}$$

$$\Theta_{M'} = \left\{\boldsymbol{\theta} = (K_m^{(\ell)}, Q_m^{(\ell)}, V_m^{(\ell)}) : \max_{\ell\in[L]} M^{(\ell)} \leqslant M, \|\boldsymbol{\theta}\| \leqslant M'\right\}.$$

**Theorem 4.1** (Generalization for pretraining). *For MoR problem given by equation 1, suppose that* $\max_{i \leqslant K} \|\beta_i^*\|_2 \leqslant C$ *with some absolute constant $C$, then with probability at least $1 - 3\xi$ (over the pretraining instances $\{\mathbf{Z}^{(j)}\}_{j \in [B]}$), the solution $\widehat{\boldsymbol{\theta}}$ to equation 9 satisfies*

$$L_{\mathrm{icl}}(\widehat{\boldsymbol{\theta}}) \leqslant \inf_{\boldsymbol{\theta} \in \Theta_{B'}} L_{\mathrm{icl}}(\boldsymbol{\theta}) + \mathcal{O}\left( (1 + 4/(\pi_{\min} K \eta^2)) \times \log\left(\frac{2nB}{\xi}\right) \sqrt{\frac{(L)^2 (MD^2)\iota + \log(1/\xi)}{B}} \right)$$

*where* $\iota = \log\left(2 + \max\left\{M', B_x, B_y, (2B_y)^{-1}\right\}\right)$, $B_x = \sqrt{\log(ndB/\xi)}$, $B_y = \sqrt{\log\left(\frac{2nB}{\xi}\right)C^2 \log(2nB/\xi)}$, *D is the hidden dimension and $M$ is the number of heads.*

**Remark 4.1.** The proof of Theorem 4.1 is provided in Appendix B.1. Under the low SNR settings, the constructed transformer generally requires more attention layers than those in the high SNR settings to achieve the same level of excess risk. Besides, the theorem highlights a fundamental trade-off: while larger models (more layers, heads, and hidden dimensions) increase have the capacity to learn complex patterns, they also require more pretraining datasets ($B$).

## 4.2 Dynamics of single linear self-attention layer

Next, we investigate the training dynamics of gradient flow for MoR models. For this subsection, we consider transformers with linear self-attention layers. Given the input matrix in the form of equation 4, appropriately sized key, query and value matrices $K, Q, V$, the output of a linear attention block is given by $\hat{y}_{n+1} = \left[f_{\mathrm{LSA}}(E; \theta)\right]_{d+1, n+1}$. For our technical analysis, following Zhang et al. (2023), we only consider training the model equation 5 over population squared loss between $\hat{y}_{n+1}$ and $y_{n+1}$, i.e.

$$\theta^* := \arg\min_{\theta \in \Theta} \left\{ L_{\mathrm{SA}}(E, \theta) = \frac{1}{2}\mathbb{E}_{\mathcal{P}}\left[(\hat{y}_{n+1} - y_{n+1})^2\right] \right\}.$$

And we assume that $\mathbb{E}\beta = \sum_{i=1}^K \pi_i \beta_i^* = 0$ on the MoR task equation 1. The gradient descent gives

$$\theta_{t+1} = \theta_t - \epsilon \nabla L_{\mathrm{SA}}(E, \theta_t)$$

when minimizing $L_{\mathrm{SA}}(E, \theta)$ and this could simplified as

$$\frac{d\theta}{dt} = -\nabla L_{\mathrm{SA}}(E; \theta).$$

Letting $\epsilon \to 0$ and setting $\theta(t) = \theta_k$ at time $t = k\epsilon$, we recognize the left-hand side above as the discrete derivative of $\theta(t)$ at time $t$. Hence, we get a continuous-time ordinary differential equation (i.e. the gradient flow of the parameters)

$$\frac{d\theta}{dt} = -\nabla L_{\mathrm{SA}}(E; \theta)$$

It captures the behavior of gradient descent when we running the gradient descent with small step size. In the remaining of this section, we start by rewriting the output of the linear attention module in an alternative form. Following Zhang et al. (2023), we define

$$V = \begin{pmatrix} * & * \\ u_{21}^\top & u_{-1} \end{pmatrix}, \quad K^\top Q = \begin{pmatrix} U_{11} & * \\ u_{12}^\top & * \end{pmatrix}$$

therefore, the prediction on the query sample is given by

$$\hat{y}_{n+1} = \left[ u_{21}^\top \cdot \frac{1}{n} X^\top X \cdot U_{11} + u_{21}^\top \cdot \frac{1}{n} X^\top \mathbf{y} \cdot u_{12}^\top \right. \tag{10}$$
$$\left. + u_{-1} \cdot \frac{1}{n} \mathbf{y}^\top X \cdot U_{11} + u_{-1} \cdot \frac{1}{n} \mathbf{y}^\top \mathbf{y} \cdot u_{12}^\top \right] \cdot x_{n+1}$$

where $X = [x_1, \ldots, x_n]^\top$ and $\mathbf{y} = [y_1, \ldots, y_n]^\top$. We will consider gradient flow with an initialization that satisfies the following assumption

**Assumption 4.2.1** (Initialization). Let $\gamma > 0$ be a parameter, and let $\Theta \in \mathbb{R}^{d \times d}$ be any matrix satisfying $\Theta^\top \mathbb{E}\beta\beta^\top \neq 0$ and $\operatorname{tr}\left(\Theta\Theta^\top \left(\mathbb{E}\beta\beta^\top\right) \Theta\Theta^\top \left(\mathbb{E}\beta\beta^\top\right)\right) = 1$

$$
\begin{aligned}
u_{-1}(0) &= \gamma \cdot 1, \\
u_{12}(0) &= u_{21}(0) = 0, \\
U_{11}(0) &= \gamma \left(\mathbb{E}\beta\beta^\top\right)^{\frac{1}{2}} \Theta\Theta^\top \left(\mathbb{E}\beta\beta^\top\right)^{\frac{1}{2}}.
\end{aligned}
\tag{11}
$$

Theorem 4.2 below proves that gradient flow will converge to a global optimum under suitable initialization.

**Theorem 4.2.** *Under initialization condition equation 11, when the parameter $\gamma$ satisfies the condition*

$$
\gamma \leqslant \sqrt{\frac{2\lambda_{\min}\left(\mathbb{E}\beta\beta^\top\right)}{\sqrt{d}\left(\frac{n+d+1}{n}\lambda_{\max}\left(\mathbb{E}\beta\beta^\top\right) + \frac{\vartheta^2}{n}\right)}}
\tag{12}
$$

*and $\mathbb{E}\beta = \sum_{k=1}^K \pi_k^* \beta_k^* = 0$, we have $u_{21}(t) = u_{12}(t) = 0$ for all $t \geqslant 0$, and the gradient flow converges to a global minimum of the population loss. Moreover, $U_{11}$ and $u_{-1}$ converge to $U_{11}^*$ and $u_{-1}^*$ respectively, where*

$$
u_{-1}^* = \left\| \left(\frac{n+1}{n}\mathbb{E}\beta\beta^\top + \frac{\mathbb{E}\|\beta\|_2^2 + \vartheta^2}{n}I\right)^{-1} \mathbb{E}\beta\beta^\top \right\|_F^{\frac{1}{2}},
\tag{13}
$$

$$
U_{11}^* = \left(u_{-1}^*\right)^{-1} \left(\frac{n+1}{n}\mathbb{E}\beta\beta^\top + \frac{\mathbb{E}\|\beta\|_2^2 + \vartheta^2}{n}I\right)^{-1} \mathbb{E}\beta\beta^\top.
\tag{14}
$$

With $u_{-1}^*$ and $U_{11}^*$ specified in equation 13 and equation 14, the linear self-attention layer makes prediction on $x_{n+1}$ as

$$
\hat{y}_{n+1} = x_{n+1}^\top \left(\frac{n+1}{n}\mathbb{E}\beta\beta^\top + \frac{\mathbb{E}\|\beta\|_2^2 + \vartheta^2}{n}I\right)^{-1} \mathbb{E}\beta\beta^\top \left[\frac{1}{n}\sum_{i=1}^n y_i x_i\right].
$$

When $n$ is sufficiently large, it holds that $u_{-1}^* U_{11}^* \approx I_d$ and $\frac{1}{n}\sum_{i=1}^n y_i x_i^\top \approx \mathbb{E}_{x,y} yx$. Therefore, when $n$ is large, the prediction made by linear self-attention layer $\hat{y}_{n+1} \approx x_{n+1}^\top I_d \left(\mathbb{E}xx^\top\beta + \mathbb{E}vx\right) = x_{n+1}^\top \beta^{\mathsf{OR}}$. This shows that the linear self-attention layer effectively learns the optimal predictor in the large-sample limit.

Theorem 4.2 provides crucial insights into the convergence of gradient flow under structured initializations and zero mean assumption for the coefficients. Larger noise variance $\vartheta^2$ or smaller sample size $n$ necessitates smaller $\gamma$ (scaling of $u_{-1}(0)$). The initialization $U_{11}(0)$ and the trace condition on $\Theta$ encode prior knowledge of the input distribution $\mathbb{E}\left[\beta\beta^\top\right]$, acting as a preconditioner for efficient learning. In particular, the trace condition in equation 11 ensures $\Theta$ is scaled to interact stably with the data covariance, preventing exploding/vanishing updates. Finally, we remark that if $\mathbb{E}\beta \neq 0$, there are additional terms affecting the dynamics, possibly complicating the convergence. We leave this problem as a possible future direction.

Compared to Theorem 4.1 in Zhang et al. (2023), our results are applicable for the case when label noise is present. Furthermore, we generalize the distribution assumption proposed on the the coefficient $\beta$. Indeed, in Zhang et al. (2023), the sample on the prompt are generated based on the noiseless model $y_i = x_i^\top \beta$ with $\beta \sim N(0, I_d)$. Whereas, our analysis only relies on the moment information $\mathbb{E}\beta$ and $\mathbb{E}\beta\beta^\top$ on the distribution of $\beta$. Besides, as mentioned above, our assumption equation 11 precisely characterizes how the initialization condition on the parameters depends on the covariance structure $\mathbb{E}\beta\beta^\top$.

## 5 Simulation study

In this section, we present numeric results of training transformers on the prompts described in Section 2. We train our transformers using Adam, with a constant step size of 0.001. For the general settings in the experiments, the dimension of samples $d = 32$. The number of training prompts are $B = 64$ by default ($B$ is other value if otherwise stated). The hidden dimension are $D = 64$ by default ($D$ is other value if otherwise

stated). The training data $x_i$'s are i.i.d. sampled from standard multivariate Gaussian distribution and the noise $v_i$'s are i.i.d. sampled from normal distribution $\mathcal{N}(0, \vartheta^2)$. The regression coefficients are generated from standard multivariate normal and then normalized by its $l_2$ norm. Once the coefficient is generated, it is fixed. The excess MSE is reported. Each experiment is repeated by 20 times and the results is averaged over these 20 times. We also point out that assumptions regarding $n$ and $d$ in Section 3 is for theoretical constructions. It is not related to the trained transformer in practice. The performance of the trained transformer will be better than the constructed transformers in Theorem 3.1 and 3.2.

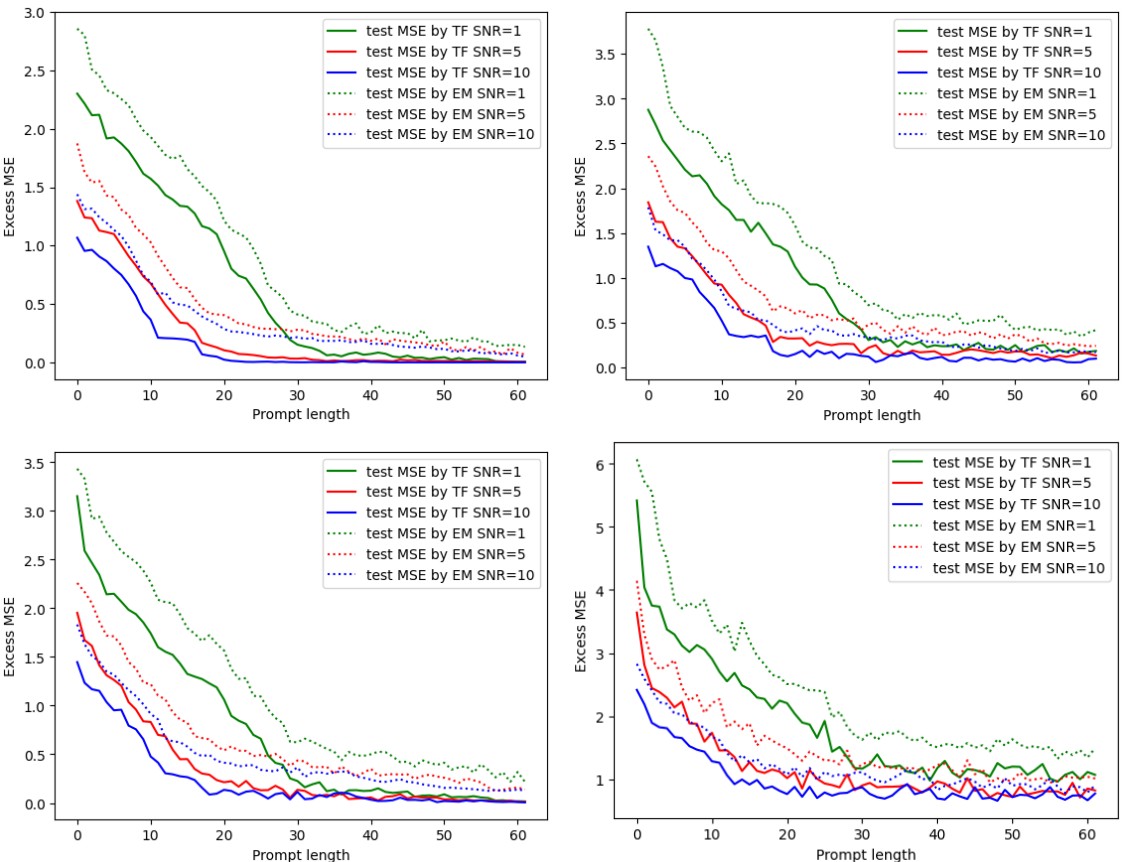

Figure 1: Plot of excess testing risk by the transformer (and EM algorithm) v.s. prompt length with different SNRs on MoR tasks with $K = 2, 3, 5, 20$ components.

The initializations of the transformer parameters for all our experiments are random standard Gaussian. As we will see from our results, transformers provide efficient prediction and estimation errors despite this global initialization. A possible explanation for this fact might be the overparametrization naturally available in the transformer architecture and the related need for overparametrization for estimation in mixture models (Dwivedi et al., 2020; Xu et al., 2024); we leave a theoretical investigation of this fact as intriguing future work.

**Performance with different prompt length:** In this experiment, we vary the number of components $K = 2, 3, 5, 20$. For each case, we run the experiment with different SNR ($\eta = 1, 5, 10$). The $x$-axis is the prompt length, and the $y$-axis is the test MSE. The number of attention layers is given by $L = 4$. The performance results of the transformer are presented in Figure 1.

From Figure 1, we observe the following trends: (1) With the number of prompt lengths and other parameters held constant, the trained transformer exhibits a higher excess MSE in the low SNR settings. (2) When the prompt length is very small, indicating an insufficient number of samples in the prompt, the resulting

excess test MSE is high. However, with a sufficiently large prompt length, the performance of the transformers stabilizes and is effective across all SNR settings, leading to a relatively small excess test MSE. (3) Additionally, when the prompt length and SNR are fixed, an increase in the number of components tends to result in a larger excess test MSE.

**Performance with different number of training prompts:** In this experiment, we vary the number of training prompts $B$ from 64 to 512. For each case, we run the experiment with two components ($K = 2$), different SNR ($\eta = 1, 5, 10$). The $x$-axis is the number of training prompts, and the $y$-axis is the test MSE. The length of training prompts is $n = 64$.

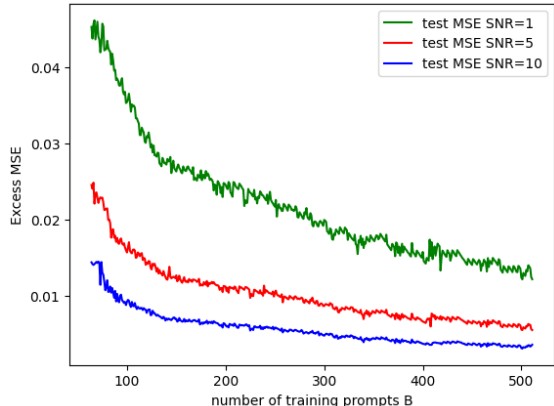

Figure 2: Plot of excess testing risk of the transformer v.s. the number of prompts with different SNRs.

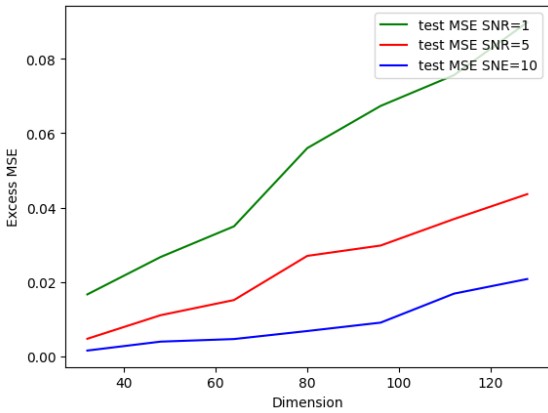

Figure 3: Plot of excess testing risk of the transformer v.s. the dimension $d$ with different SNRs.

Figure 2 gives the performance of trained transformer with different number of training prompts under three different SNR settings. Based on Figure 2, we observe that when the number of training prompts is already sufficiently large, the excess MSE is relatively small. Furthermore, as the number of training prompts increases, there is a general trend of decreasing in the excess MSE.

**Performance with different dimension $d$ of samples:** In this experiment, we fix the hidden dimension $D = 256$, the number of components $K = 2$, the number of prompts $B = 128$ and the prompt length is given by $n = 64$. The $x$-axis is the dimension $d$ of the input sample $x_i$ and $y$-axis is the excess test MSE. In this experiment, we evaluate the performance of the trained transformer for various dimensions $d = 32, 48, 64, 80, 96, 112, 128$. The performance of the transformer are presented in Figure 3. Observations

from this figure indicate that increasing the dimension $d$ significantly raises the excess test MSE. Notably, this increase becomes more pronounced at the lower SNR levels.

An additional experiment on the performance with different number of hidden dimension is provided in Section D.2.

## 6 Conclusions

We have explored the behavior of transformers in handling linear MoR problems, demonstrating their in-context learning capabilities through both theoretical analysis and empirical experiments. Specifically, we showed that transformers are capable of implementing the EM algorithm for linear MoR tasks. Additionally, we have examined the sample complexity involved in pretraining transformers with a finite number of ICL training instances and the training dynamics of gradient flow, offering valuable insights into their practical performance. Our empirical findings also reveal that transformer performance is less susceptible to initializations. For future work, understanding the training dynamics of general transformers for MoR problems remains a highly interesting and challenging task. Furthermore, extending our results to non-linear MoR models would be a natural direction.

Prior work (e.g., Raventós et al. (2023)) studied ICL for linear regression with coefficients sampled from a prior, measuring task difficulty by the minimum distance across all regression vectors in the pretraining corpus. In contrast, our paper defines SNR per task, based on the separation of the $K$ mixture components within a single prompt relative to noise, so having many distinct tasks in pretraining does not automatically reduce SNR. For technical convenience, we assume fixed ground-truth coefficients $\beta_k^*$, but one could instead model them with priors (e.g., Gaussian or uniform on the sphere), which would reduce to Raventós et al. (2023) in the $K = 1$ case. Extending our results to this random-effects setting requires re-deriving several lemmas and adapting the proofs of Theorems 3.1–3.2, which we leave as promising future work.

**Acknowledgments**

The work of Krishnakumar Balasubramanian was supported in part by National Science Foundation (NSF) under grant DMS-2413426. The work of Lifeng Lai was supported in part by NSF under grants CCF-22-32907 and ECCS-24-48268.

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

# A    Proof of Theorems in Section 3

In this section, we illustrate that the transformers constructed in Theorem 3.1 and Theorem Theorem 3.2 can solve the MoR problem by implementing the EM algorithm internally while GD is used in each M-step. Previous works (e.g. Balakrishnan et al. (2017), Kwon et al. (2019) and Kwon et al. (2021)) focused on the sample-based EM algorithm, typically employing closed-form solutions or one-step gradient approaches in the M-step. We start with detailed explanation on the existence of the transformer in MoR problem with two symmetric components ($K = 2, \pi_1^* = \pi_2^* = \frac{1}{2}$ and $\beta_1^* = -\beta_2^* = \beta^*$). For general analysis, we explore the performance of the transformer using a gradient descent in steps $T$ within the EM algorithm. To simplify the analysis, we restrict our stepsize $\alpha \in (0, 1)$ in each gradient descent step in M-step.

**Attention layer can implement the one-step gradient descent.** We first recall how the attention layer can implement one-step GD for a certain class of loss functions as demonstrated by Bai et al. (2024). Let $\ell : \mathbb{R}^2 \to \mathbb{R}$ be a loss function. Let $\hat{L}_n(\beta) := \frac{1}{n} \sum_{i=1}^n \ell(\beta^\top x_i, y_i)$ denote the empirical risk with loss function $\ell$ on dataset $\{(x_i, y_i)\}_{i \in [n]}$, and we denote

$$\beta_{k+1} := \beta_k - \alpha \nabla \hat{L}_n(\beta_k) \tag{15}$$

as the GD trajectory on $\hat{L}_n$ with initialization $\beta_0 \in \mathbb{R}^d$ and learning rate $\alpha > 0$. The foundational concept of the construction presented in Theorem 3.2 is derived from Bai et al. (2024). It hinges on the condition that the partial derivative of the loss function, $\partial_s \ell : (s, t) \mapsto \partial_s \ell(s, t)$ (considered as a bivariate function), can be approximated by a sum of ReLU functions, which are defined as follows:

**Definition A.1** (Approximability by sum of ReLUs). A function $g : \mathbb{R}^k \to \mathbb{R}$ is $(\varepsilon_{\text{approx}}, R, M, C)$-approximable by sum of ReLUs, if there exists a "$(M, C)$-sum of ReLUs" function

$$f_{M,C}(\mathbf{z}) = \sum_{m=1}^M c_m \sigma(\mathbf{a}_m^\top [\mathbf{z}; 1]) \quad \text{with} \quad \sum_{m=1}^M |c_m| \leqslant C, \max_{m \in [M]} \|\mathbf{a}_m\|_1 \leqslant 1, \mathbf{a}_m \in \mathbb{R}^{k+1}, c_m \in \mathbb{R}$$

such that $\sup_{\mathbf{z} \in [-R,R]^k} |g(\mathbf{z}) - f_{M,C}(\mathbf{z})| \leqslant \varepsilon_{\text{approx}}$.

Suppose that the partial derivative of the loss function, $\partial_s \ell(s, t)$, is approximable by a sum of ReLUs. Then, $T$ steps of GD, as described in equation 15, can be approximately implemented by employing $T$ attention layers within the transformer. This result is formally presented in Proposition C.1.

**Transformer can implement the gradient-EM algorithm:** Proposition C.1 illustrates how the transformer described in Theorem 3.2 is capable of learning from the MoR problem. Using Proposition C.1, we can construct a transformer whose architecture consists of attention layers that implement GD for each M-step, followed by additional attention layers responsible for computing the necessary quantities in the E-step. Here is a summary of how the transformer implements the EM algorithm for the mixture of regression problem. Following the notation from Balakrishnan et al. (2017), we consider the mixture of regression with two symmetric components and define the weight function:

$$w_\beta(x, y) = \frac{\exp\left\{-\frac{1}{2\vartheta^2}\left(y - x^\top \beta\right)^2\right\}}{\exp\left\{-\frac{1}{2\vartheta^2}\left(y - x^\top \beta\right)^2\right\} + \exp\left\{-\frac{1}{2\vartheta^2}\left(y + x^\top \beta\right)^2\right\}}.$$

Denote $\beta^{(t)}$ as the current parameter estimates of $\beta^*$ in the EM algorithm for the MoR problem. During each M-step, the objective is to maximize the following loss function:

$$Q_n(\beta' \mid \beta^{(t)}) = \frac{-1}{2n} \sum_{i=1}^n \left(w_{\beta^{(t)}}(x_i, y_i)(y_i - x_i^\top \beta')^2 + (1 - w_{\beta^{(t)}}(x_i, y_i))(y_i + x_i^\top \beta')^2\right). \tag{16}$$

The update $\beta^{(t+1)}$ is given by $\beta^{(t+1)} = \arg\max_{\beta' \in \Omega} Q_n(\beta' \mid \beta^{(t)})$. Lemma A.1 below, demonstrates that the function $\hat{L}_n^{(t)(\beta)}$ minimized in each M-step is approximable by a sum of ReLUs.

**Lemma A.1.** For the function $\hat{L}_n^{(t)}(\beta) = \frac{1}{n}\sum_{i=1}^{n} l^{(t)}(x_i^\top \beta, y_i)$, where

$$l^{(t)}(x_i^\top \beta, y_i) = w_{\beta^{(t)}}(x_i, y_i)(y_i - x_i^\top \beta)^2 + (1 - w_{\beta^{(t)}}(x_i, y_i))(y_i + x_i^\top \beta)^2,$$

it holds that (1) $l^{(t)}(s, t)$ is convex in first argument; and (2) $\partial_s l^{(t)}(s, t)$ is $(0, +\infty, 4, 16)$-approximable by sum of ReLUs.

*Proof of Lemma A.1.* Note that

$$l^{(t)}(s, t) = w_{\beta^{(t)}}(x_i, y_i)(t - s)^2 + (1 - w_{\beta^{(t)}}(x_i, y_i))(t + s)^2.$$

Taking derivative w.r.t. the first argument yields

$$\partial_s l^{(t)}(s, t) = w_{\beta^{(t)}}(x_i, y_i)(-2)(t - s) + (1 - w_{\beta^{(t)}}(x_i, y_i))2(t + s),$$
$$\partial_s^2 l^{(t)}(s, t) = 2w_{\beta^{(t)}}(x_i, y_i) + 2(1 - w_{\beta^{(t)}}(x_i, y_i)) = 2.$$

Hence, $l(s, t)$ is convex in the first argument and

$$\begin{aligned}
\partial_s l^{(t)}(s, t) &= 2w_{\beta^{(t)}}(x_i, y_i)(s - t) + 2(1 - w_{\beta^{(t)}}(x_i, y_i))(s + t) \\
&= 2w_{\beta^{(t)}}(x_i, y_i)[2\sigma((s - t)/2) - 2\sigma(-(s - t)/2)] \\
&\quad + 2(1 - w_{\beta^{(t)}}(x_i, y_i))[2\sigma((s + t)/2) - 2\sigma(-(s + t)/2)].
\end{aligned}$$

Here $c_1 = 4w_{\beta^{(t)}}(x_i, y_i)$, $c_2 = -4w_{\beta^{(t)}}(x_i, y_i)$, $c_3 = 4(1 - w_{\beta^{(t)}}(x_i, y_i))$ and $c_4 = -4(1 - w_{\beta^{(t)}}(x_i, y_i))$. Now, we have $|c_1| + |c_2| + |c_3| + |c_4| \leqslant 16$ and $\partial_s l(s, t)$ is $(0, +\infty, 4, 16)$-approximable by sum of ReLUs. $\square$

By Lemma A.1, we can design attention layers with $T$ layers that implement the $T$ steps of GD for the empirical loss $\hat{L}_n^{(t)}(\beta')$ as outlined in Proposition C.1. We provide a concise demonstration of the entire process for MoR with two symmetric components below. Starting with an appropriate initialization $\beta^{(0)}$, the first M-step minimizes the loss function:

$$\hat{L}_n^{(0)}(\beta) = \frac{1}{n}\sum_{i=1}^{n} \left\{w_{\beta^{(0)}}(x_i, y_i)(y_i - x_i^\top \beta)^2 + (1 - w_{\beta^{(0)}}(x_i, y_i))(y_i + x_i^\top \beta)^2\right\}.$$

Following Proposition C.1, given the input sequence formatted as $h_i = [x_i; y_i'; 0_d; 0_{D-2d-3}; 1; t_i]$, there exists a transformer with $T$ attention layers that gives the output $\tilde{h}_i = [x_i; y_i'; \beta_T^{(0)}, 0_{D-2d-3}; 1; t_i]$. Furthermore, the existence of a transformer capable of computing the necessary quantities in the M-step is guaranteed by Proposition 1 from Pathak et al. (2024) and we restate this proposition in section C in appendix.

It is worth mentioning that computing $w_{\beta^{(t)}}(x_i, y_i)$ in each M-step can be easily implemented by affine and softmax operation in Proposition C.2. Similar arguments can be made for the upcoming iterations of the EM algorithm and we summarize these results in Lemma A.2 and Lemma A.3.

**Lemma A.2.** In each E-step, given the input $H^{(T+1)} = [h_1^{(T+1)}, \ldots, h_{n+1}^{(T+1)}]$ where

$$h_i^{(T+1)} = [x_i; y_i'; \beta_T^{(t)}; \mathbf{0}_{D-2d-3}; 1; t_i; w_{\beta_T^{(t-1)}}(x_i, y_i)]^\top, \quad i = 1, \ldots, n,$$
$$h_{n+1}^{(T+1)} = [x_i; x_{n+1}^\top \beta_T^{(t)}; \beta_T^{(t)}; \mathbf{0}_{D-2d-3}; 1; 1; 0]^\top,$$

there exists a transformer $\text{TF}_E^{(t)}$ that can compute $w_{\beta_T^{(t)}}(x_i, y_i)$. Furthermore, the output sequence takes the form of

$$\tilde{h}_i^{(T+1)} = [x_i; y_i'; \beta_T^{(t)}; \mathbf{0}_{D-2d-4}; 1; t_i; w_{\beta_T^{(t)}}(x_i, y_i)]^\top, \quad i = 1, \ldots, n, \tag{17}$$
$$\tilde{h}_{n+1}^{(T+1)} = [x_i; x_{n+1}^\top \beta_T^{(t)}; \beta_T^{(t)}; \mathbf{0}_{D-2d-4}; 1; 1; 0]^\top. \tag{18}$$

*Proof of Lemma A.2.* Note that the output of M-step after $t$-th iteration is given by $H^{(T+1)} = \left[h_1^{(T+1)}, \ldots, h_{n+1}^{(T+1)}\right]$ where

$$h_i^{(T+1)} = \left[x_i; y_i'; \beta_T^{(t)}; \mathbf{0}_{D-2d-3}; 1; t_i; w_{\beta_T^{(t-1)}}(x_i, y_i)\right]^\top, \quad i = 1, \ldots, n$$

$$h_{n+1}^{(T+1)} = \left[x_i; x_{n+1}^\top \beta_T^{(t)}; \beta_T^{(t)}; \mathbf{0}_{D-2d-3}; 1; 1; 0\right]^\top,$$

i.e.

$$H^{(T+1)} = \begin{bmatrix} x_1 & x_2 & \ldots & x_n & x_{n+1} \\ y_1' & y_2' & \ldots & y_n' & x_{n+1}^\top \beta_T^{(t)} \\ \beta_T^{(t)} & \beta_T^{(t)} & \ldots & \beta_T^{(t)} & \beta_T^{(t)} \\ \mathbf{0}_{D-2d-3} & \mathbf{0}_{D-2d-3} & \ldots & \mathbf{0}_{D-2d-3} & \mathbf{0}_{D-2d-3} \\ 1 & 1 & \ldots & 1 & 1 \\ t_1 & t_2 & \ldots & t_n & 1 \\ w_{\beta_T^{(t-1)}}(x_1, y_1) & w_{\beta_T^{(t-1)}}(x_2, y_2) & \ldots & w_{\beta_T^{(t-1)}}(x_n, y_n) & 0 \end{bmatrix}.$$

After copy down and scale operation, the output is given by

$$H^{(T+1)}(1) = \begin{bmatrix} x_1 & x_2 & \ldots & x_n & x_{n+1} \\ y_1' & y_2' & \ldots & y_n' & x_{n+1}^\top \beta_T^{(t)} \\ \beta_T^{(t)} & \beta_T^{(t)} & \ldots & \beta_T^{(t)} & \beta_T^{(t)} \\ -\beta_T^{(t)} & -\beta_T^{(t)} & \ldots & -\beta_T^{(t)} & -\beta_T^{(t)} \\ \mathbf{0}_{D-3d-3} & \mathbf{0}_{D-3d-3} & \ldots & \mathbf{0}_{D-3d-3} & \mathbf{0}_{D-3d-3} \\ 1 & 1 & \ldots & 1 & 1 \\ t_1 & t_2 & \ldots & t_n & 1 \\ w_{\beta_T^{(t-1)}}(x_1, y_1) & w_{\beta_T^{(t-1)}}(x_2, y_2) & \ldots & w_{\beta_T^{(t-1)}}(x_n, y_n) & 0 \end{bmatrix}.$$

In this step, the negative of $\beta_T^{(t)}$ is copied down. After affine operation, the output is given by

$$H^{(T+1)}(2) = \begin{bmatrix} x_1 & x_2 & \ldots & x_n & x_{n+1} \\ y_1' & y_2' & \ldots & y_n' & x_{n+1}^\top \beta_L^{(t)} \\ \beta_T^{(t)} & \beta_T^{(t)} & \ldots & \beta_T^{(t)} & \beta_T^{(t)} \\ -\beta_T^{(t)} & -\beta_T^{(t)} & \ldots & -\beta_T^{(t)} & -\beta_T^{(t)} \\ r_1 & r_2 & \ldots & r_n & 0 \\ \mathbf{0}_{D-3d-4} & \mathbf{0}_{D-3d-4} & \ldots & \mathbf{0}_{D-3d-4} & \mathbf{0}_{D-3d-4} \\ 1 & 1 & \ldots & 1 & 1 \\ t_1 & t_2 & \ldots & t_n & 1 \\ w_{\beta_T^{(t-1)}}(x_1, y_1) & w_{\beta_T^{(t-1)}}(x_2, y_2) & \ldots & w_{\beta_T^{(t-1)}}(x_n, y_n) & 0 \end{bmatrix}.$$

Here, $r_1$ is computed in the next row. After another affine operation, the output is given by

$$H^{(T+1)}(3) = \begin{bmatrix} x_1 & x_2 & \ldots & x_n & x_{n+1} \\ y_1' & y_2' & \ldots & y_n' & x_{n+1}^\top \beta_T^{(t)} \\ \beta_T^{(t)} & \beta_T^{(t)} & \ldots & \beta_T^{(t)} & \beta_T^{(t)} \\ -\beta_T^{(t)} & -\beta_T^{(t)} & \ldots & -\beta_T^{(t)} & -\beta_T^{(t)} \\ r_1 & r_2 & \ldots & r_n & 0 \\ \tilde{r}_1 & \tilde{r}_2 & \ldots & \tilde{r}_n & 0 \\ \mathbf{0}_{D-3d-5} & \mathbf{0}_{D-3d-5} & \ldots & \mathbf{0}_{D-3d-5} & \mathbf{0}_{D-3d-5} \\ 1 & 1 & \ldots & 1 & 1 \\ t_1 & t_2 & \ldots & t_n & 1 \\ w_{\beta_T^{(t-1)}}(x_1, y_1) & w_{\beta_T^{(t-1)}}(x_2, y_2) & \ldots & w_{\beta_T^{(t-1)}}(x_n, y_n) & 0 \end{bmatrix}.$$

Similarly, $\tilde{r}_1$ is computed in the next row. After softmax operation, the output is given by

$$H^{(T+1)}(4) = \begin{bmatrix} x_1 & x_2 & \ldots & x_n & x_{n+1} \\ y_1' & y_2' & \ldots & y_n' & x_{n+1}^\top \beta_T^{(t)} \\ \beta_T^{(t)} & \beta_T^{(t)} & \ldots & \beta_T^{(t)} & \beta_T^{(t)} \\ -\beta_T^{(t)} & -\beta_T^{(t)} & \ldots & -\beta_T^{(t)} & -\beta_T^{(t)} \\ r_1 & r_2 & \ldots & r_n & 0 \\ \tilde{r}_1 & \tilde{r}_2 & \ldots & \tilde{r}_n & 0 \\ \mathbf{0}_{D-3d-5} & \mathbf{0}_{D-3d-5} & \ldots & \mathbf{0}_{D-3d-5} & \mathbf{0}_{D-3d-5} \\ 1 & 1 & \ldots & 1 & 1 \\ t_1 & t_2 & \ldots & t_n & 1 \\ w_{\beta_T^{(t)}}(x_1,y_1) & w_{\beta_T^{(t)}}(x_2,y_2) & \ldots & w_{\beta_T^{(t)}}(x_n,y_n) & 0 \end{bmatrix}.$$

The only change is that the responsibility register is updated: the row/column containing $w_{\beta_T^{(t-1)}}(x_i,y_i)$ is overwritten by $w_{\beta_T^{(t)}}(x_i,y_i)$ for all tokens $i$. This update is exactly the E-step computed by the forward pass at that stage. After copy over operation, the output is given by

$$H^{(T+1)}(5) = \begin{bmatrix} x_1 & x_2 & \ldots & x_n & x_{n+1} \\ y_1' & y_2' & \ldots & y_n' & x_{n+1}^\top \beta_T^{(t)} \\ \beta_T^{(t)} & \beta_T^{(t)} & \ldots & \beta_T^{(t)} & \beta_T^{(t)} \\ \mathbf{0}_{D-2d-3} & \mathbf{0}_{D-2d-3} & \ldots & \mathbf{0}_{D-2d-3} & \mathbf{0}_{D-2d-3} \\ 1 & 1 & \ldots & 1 & 1 \\ t_1 & t_2 & \ldots & t_n & 1 \\ w_{\beta_T^{(t)}}(x_1,y_1) & w_{\beta_T^{(t)}}(x_2,y_2) & \ldots & w_{\beta_T^{(t)}}(x_n,y_n) & 0 \end{bmatrix}. \tag{19}$$

Finally, this transformer gives the output matrix $H_M^{(T+1)}$ as equation 19. $\qquad\square$

**Lemma A.3.** In each M-step, given the input matrix as equation 17 and equation 18, there exists a transformer $\mathrm{TF}_M^{(t)}$ with $T + 1$ attention layers that can implement $T$ steps of GD on the loss function $\hat{L}_n^{(t)}(\beta) = \frac{1}{n}\sum_{i=1}^n l^{(t)}(x_i^\top \beta, y_i)$, where $l^{(t)}(x_i^\top \beta, y_i) = w_{\beta_T^{(t)}}(x_i,y_i)(y_i - x_i^\top \beta)^2 + (1 - w_{\beta_T^{(t)}}(x_i,y_i))(y_i + x_i^\top \beta)^2$. Furthermore, the output sequence takes the form of

$$h_i^{(T+1)} = \left[x_i; y_i'; \beta_T^{(t+1)}; \mathbf{0}_{D-2d-3}; 1; t_i; w_{\beta_T^{(t)}}(x_i,y_i)\right]^\top, \quad i = 1,\ldots,n,$$
$$h_{n+1}^{(T+1)} = \left[x_i; x_{n+1}^\top \beta_T^{(t+1)}; \beta_T^{(t+1)}; \mathbf{0}_{D-2d-3}; 1; 1; 0\right]^\top.$$

*Proof of Lemma A.3.* The conceptual basis of the proof draws from the theorem discussed in Bai et al. (2024). By Proposition C.2 in Bai et al. (2024), there exists a function $f : \mathbb{R}^2 \to \mathbb{R}$ of form

$$f(s,t) = \sum_{m=1}^4 c_m \sigma(a_m s + b_m t + d_m) \quad \text{with} \quad \sum_{m=1}^4 |c_m| \leq 16, |a_m| + |b_m| + |d_m| \leq 1, \forall m \in [4],$$

such that $\sup_{(s,t)\in\mathbb{R}^2}|f(s,t) - \partial_s \ell(s,t)| \leq \varepsilon$. Next, in each attention layer, for every $m \in [4]$, we define matrices $Q_m, K_m, V_m \in \mathbb{R}^{D\times D}$ such that

$$Q_m h_i = \begin{bmatrix} a_m \beta \\ b_m \\ d_m \\ -2 \\ 0 \\ 0 \end{bmatrix}, \quad K_m h_j = \begin{bmatrix} x_j \\ y_j' \\ 1 \\ R(1-t_j) \\ 0 \\ 0 \end{bmatrix}, \quad V_m h_j = -\frac{(N+1)\eta c_m}{N} \cdot \begin{bmatrix} \mathbf{0}_d \\ 0 \\ x_j \\ \mathbf{0}_{D-2p-1} \\ 0 \end{bmatrix}$$

where $D$ is the hidden dimension which is a constant multiple of $d$. In the last attention layers, the heads $\left\{(Q_m^{(T+1)}, K_m^{(T+1)}, V_m^{(T+1)})\right\}_{m=1,2}$ satisfies

$$Q_1^{(T+1)}h_i^{(T)} = [x_i; \mathbf{0}_{D-d+1}], \quad K_1^{(T+1)}h_j^{(T)} = [\beta_T^{(t+1)}; \mathbf{0}_{D-d+1}], \quad V_1^{(T+1)}h_j^{(T)} = [\mathbf{0}_d; 1; \mathbf{0}_{D-d}],$$
$$Q_2^{(T+1)}h_i^{(T)} = [x_i; \mathbf{0}_{D-d+1}], \quad K_2^{(T+1)}h_j^{(T)} = [-\beta_T^{(t+1)}; \mathbf{0}_{D-d}], \quad V_2^{(T+1)}h_j^{(T)} = [\mathbf{0}_d; -1; \mathbf{0}_{D-d}].$$

The output of this transformer gives the matrix

$$H^{(T+1)} = \begin{bmatrix} x_1 & x_2 & \dots & x_n & x_{n+1} \\ y_1' & y_2' & \dots & y_n' & x_{n+1}^\top \beta_T^{(t+1)} \\ \beta_T^{(t+1)} & \beta_T^{(t+1)} & \dots & \beta_T^{(t+1)} & \beta_T^{(t+1)} \\ \mathbf{0}_{D-2d-3} & \mathbf{0}_{D-2d-3} & \dots & \mathbf{0}_{D-2d-3} & \mathbf{0}_{D-2d-3} \\ 1 & 1 & \dots & 1 & 1 \\ t_1 & t_2 & \dots & t_n & 1 \\ w_{\beta_T^{(t)}}(x_1, y_1) & w_{\beta_T^{(t)}}(x_2, y_2) & \dots & w_{\beta_T^{(t)}}(x_n, y_n) & 0 \end{bmatrix}.$$

$\square$

Combining all the architectures into one transformer, we have that there exists a transformer that can implement gradient descent EM algorithm for $T_0$ iterations (outer loops) and in each M-step (inner loops), it implements $T$ steps of GD for function defined by equation 16. Finally, following similar procedure in Theorem 1 of Pathak et al. (2024), the output of the transformer will give $\hat{\beta}^{\text{OR}} := \pi_1 \beta_T^{(T_0+1)} - \pi_2 \beta_T^{(T_0+1)}$, which is an estimate of $\beta^{\text{OR}} = \pi_1 \beta^* - \pi_2 \beta^*$ that minimizes the prediction MSE. The output is given by $\tilde{H} \in \mathbb{R}^{D \times (n+1)}$, whose columns are

$$\tilde{h}_i = [x_i, y_i', \beta_T^{(T_0+1)}, \mathbf{0}_{D-2d-4}, 1, t_i]^\top, \qquad i = 1, \dots, n,$$
$$\tilde{h}_{n+1} = [x_{n+1}, x_{n+1}^\top \hat{\beta}^{\text{OR}}, \beta_T^{(T_0+1)}, \mathbf{0}_{D-2d-4}, 1, 1]^\top.$$

In the remaining part of Section 3, we give the proof of the estimation and prediction bound presented in Theorem 3.1 and Theorem 3.2. The transformer described in Theorem A.3, which is equipped with $T + 1$ layers, implements the M-step of the EM algorithm by performing $T$ steps of gradient descent on the empirical loss $\hat{L}_n^{(t)}(\beta)$. Therefore, it is sufficient to analyze the behavior of the sample-based EM algorithm in which $T$ steps of gradient descent are implemented during each M-step.

To begin, we define some notations that are utilized in the proof. We denote $\tilde{\beta}^{(0)}$ as any fixed initialization for the EM algorithm. The transformer described in Theorem 3.2 addresses the following optimization problem:

$$\text{argmin}\left\{\hat{L}_n^{(0)}(\beta) = \frac{1}{n}\sum_{i=1}^{n}\left\{w_{\beta^{(0)}}(x_i, y_i)(y_i - x_i^\top\beta)^2 + (1 - w_{\beta^{(0)}}(x_i, y_i))(y_i + x_i^\top\beta)^2\right\}\right\}$$

for some weight function $w_{\beta^{(0)}} \in (0, 1)$. The transformer generates a sequence $\beta_1^{(0)}, \beta_2^{(0)}, \dots$, with $\beta_k^{(0)} \to \tilde{\beta}^{(1)} = \arg\min \hat{L}_n^{(0)}(\beta)$ as $k \to \infty$. More generally, we denote $\tilde{\beta}^{(t)}$ as the minimizer of the loss function $\hat{L}_n^{(t-1)}(\beta)$ at each M-step. Additionally, $\beta_1^{(t-1)}, \cdots, \beta_T^{(t-1)}$ represents the sequence generated by applying $T + 1$ attention layers of the constructed transformer in Lemma A.3 on the loss $\hat{L}_n^{(t-1)}(\beta)$.

The approach to analyzing the convergence behavior of the transformer's output, $\text{TF}(H)$, involves examining the performance of the sample-based gradient EM algorithm. This analysis is conducted by coupling the finite sample EM with the population EM, drawing on methodologies from Balakrishnan et al. (2017) and Kwon et al. (2019).

## A.1 Results in population gradient EM algorithm for MoR problem

In this section, we present some results regarding the population EM algorithm. Given the current estimator of the parameter $\beta^*$ to be $\beta^{(t)}$. The population gradient EM algorithm maximizes (see Balakrishnan et al.

(2017) and Kwon et al. (2019))

$$Q\big(\beta \mid \beta^{(t)}\big) = -\frac{1}{2}\mathbb{E}\Big[w_{\beta^{(t)}}(X,Y)\big(Y - \langle X,\beta\rangle\big)^2 + \big(1 - w_{\beta^{(t)}}(X,Y)\big)\big(Y + \langle X,\beta\rangle\big)^2\Big],$$

whose gradient is given by $\mathbb{E}\big[\tanh\big(\frac{1}{\vartheta^2}YX^\top\beta^{(t)}\big)YX - \beta\big]$. Rather than using the standard population EM update

$$\tilde{\beta}^{(t+1)} = \arg\max_\beta Q\big(\beta \mid \beta^{(t)}\big) = \mathbb{E}\Big[\tanh\Big(\frac{1}{\vartheta^2}YX^\top\beta^{(t)}\Big)YX\Big] \tag{20}$$

the output after applying $T$ steps of gradient descent is employed as the subsequent estimator for the parameter $\beta^*$, i.e.

$$\beta^{(t+1)} = (1-\alpha)^T\beta^{(t)} + \big(1 - (1-\alpha)^T\big)\mathbb{E}\Big[\tanh\Big(\frac{1}{\vartheta^2}YX^\top\beta^{(t)}\Big)YX\Big], \tag{21}$$

where $\alpha \in (0,1)$ is the step size of the gradient descent.

In each iteration of the population gradient EM algorithm, the current iterate is denoted by $\beta$, the next iterate by $\beta'$ and the standard EM update based on equation 20 by $\tilde{\beta}'$. We concentrate on a single iteration of the population EM, which yields the next iterate $\beta'$. Consequently, equation 21 becomes:

$$\beta' = (1-\alpha)^T\beta + \big(1 - (1-\alpha)^T\big)\tilde{\beta}'. \tag{22}$$

We employ techniques similar to those used in Kwon et al. (2019) for basis transformation. By selecting $v_1 = \beta/\|\beta\|_2$ in the direction of the current iterate and $v_2$ as the orthogonal complement of $v_1$ within the span of $\{\beta, \beta^*\}$, we extend these vectors to form an orthonormal basis $\{v_1, \ldots, v_d\}$ in $\mathbb{R}^d$. To simplify notation, we define:

$$b_1 := \langle\beta, v_1\rangle = \|\beta\|_2, \qquad b_1^* := \langle\beta^*, v_1\rangle \qquad b_2^* := \langle\beta^*, v_2\rangle, \tag{23}$$

which represent the coordinates of the current estimate $\beta$ and $\beta^*$. The next iterate $\beta'$ can then be expressed as:

$$\beta' = (1-\alpha)^T b_1 v_1 + \big(1 - (1-\alpha)^T\big)\mathbb{E}\Big[\tanh\Big(\frac{\alpha_1 b_1}{\vartheta^2}Y\Big)Y\sum_{i=1}^d \alpha_i v_i\Big] \tag{24}$$

based on spherical symmetry of Gaussian distribution. The expectation is taken over $\alpha_i \sim \mathcal{N}(0,1)$ and $Y \mid \alpha_i \sim \mathcal{N}(\alpha_1 b_1^* + \alpha_2 b_2^*, \vartheta^2)$. Without loss of generality, we assume that $b_1, b_1^*, b_2^* \geq 0$.

Lemma A.4 is analogous to Lemma 1 from Kwon et al. (2019). It provides an explicit expression for $\beta'$ within the established basis system, demonstrating among other insights that $\beta'$ resides within the span$\{\beta, \beta^*\}$. Consequently, all estimators of $\beta^*$ generated by the population gradient EM algorithm remain confined within the span$\{\beta^{(0)}, \beta^*\}$

**Lemma A.4.** Suppose that $\alpha \in (0,1)$. Define $\vartheta_2^2 := \vartheta^2 + b_2^{*2}$. We can write $\beta' = b_1' v_1 + b_2' v_2$, where $b_1'$ and $b_2'$ satisfy

$$b_1' = (1-\alpha)^T b_1 + \big(1 - (1-\alpha)^T\big)\big(b_1^* S + R\big), \tag{25}$$
$$b_2' = \big(1 - (1-\alpha)^T\big)b_2^* S. \tag{26}$$

Here, $S \geq 0$ and $R > 0$ are given explicitly by

$$S := \mathbb{E}_{\alpha_1 \sim \mathcal{N}(0,1), y \sim \mathcal{N}(0,\vartheta_2^2)}\Big[\tanh\big(\tfrac{\alpha_1 b_1}{\vartheta^2}(y + \alpha_1 b_1^*)\big) + \tfrac{\alpha_1 b_1}{\vartheta^2}(y + \alpha_1 b_1^*)\tanh'\big(\tfrac{\alpha_1 b_1}{\vartheta^2}(y + \alpha_1 b_1^*)\big)\Big] \tag{27}$$

and

$$R := \big(\vartheta^2 + \|\beta^*\|_2^2\big)\mathbb{E}_{\alpha_1 \sim \mathcal{N}(0,1), y \sim \mathcal{N}(0,\vartheta_2^2)}\Big[\frac{\alpha_1^2 b_1}{\vartheta^2}\tanh'\Big(\frac{\alpha_1 b_1}{\vartheta^2}\big(y + \alpha_1 b_1^*\big)\Big)\Big]. \tag{28}$$

Moreover, $S = 0$ iff $b_1 = 0$ or $b_1^* = 0$.

*Proof.* The proof of Lemma A.4 is directly adapted from the argument used in Lemma 1 from Kwon et al. (2019), applying equation 24 for our specific context. In equation 24, the inner expectation over $y$ is independent of $\alpha_i$ for $i \geqslant 3$. Consequently, taking the expectation over $\alpha_i$ for $i \geqslant 3$ results in zero, confirming that $\beta'$ remains within the plane spanned by $v_1, v_2$. This allows us to express $\beta'$ as $\beta' = b_1'v_1 + b_2'v_2$ with

$$b_1' = (1-\alpha)^T b_1 + \left(1 - (1-\alpha)^T\right)\mathbb{E}_{\alpha_1,\alpha_2}\left[\mathbb{E}_{Y|\alpha_1,\alpha_2}\left[\tanh\left(\frac{b_1\alpha_1}{\vartheta^2}Y\right)Y\right]\alpha_1\right], \tag{29}$$

$$b_2' = \left(1 - (1-\alpha)^T\right)\mathbb{E}_{\alpha_1,\alpha_2}\left[\mathbb{E}_{Y|\alpha_1,\alpha_2}\left[\tanh\left(\frac{b_1\alpha_1}{\vartheta^2}Y\right)Y\right]\alpha_2\right], \tag{30}$$

where the expectation is taken over $\alpha_i \sim \mathcal{N}(0,1)$, and $y \mid \alpha_i \sim \mathcal{N}(\alpha_1 b_1^* + \alpha_2 b_2^*, \vartheta^2)$. The computation from equation 29 and equation 30 to equation 27 and equation 28 is identical to that in Lemma 1 of Kwon et al. (2019). $\square$

The findings in Lemma A.4 align with Lemma 1 from Klusowski et al. (2019). As the number of iterations $T$ approaches infinity, the estimator $\beta'$ converges to the standard population EM update

$$\beta^{(t)} \to \mathbb{E}_{X\sim\mathcal{N}(0,I)}\left[\left(\mathbb{E}_{Y|X\sim N(\langle X,\beta*\rangle,\vartheta^2)}\left[\tanh\left(\frac{\langle X,\beta^{(t-1)}\rangle}{\vartheta^2}Y\right)Y\right]\right)X\right].$$

For any number of steps $T$, the angle between $\beta'$ and $\beta^*$ is consistently smaller than that between $\beta$ and $\beta^*$. This can be observed by noting that:

$$0 \leqslant \tan\angle\left(\beta',\beta\right) = \frac{b_2'}{b_1'} = \frac{\left(1-(1-\alpha)^T\right)b_2^* S}{(1-\alpha)^T b_1 + \left(1-(1-\alpha)^T\right)(b_1^* S + R)} \leqslant \frac{b_2^*}{b_1^*} = \tan\angle\left(\beta^*,\beta\right). \tag{31}$$

These relationships demonstrate the geometric convergence properties of the estimation process. Motivated by equation 31, we examine the behavior of the angle between the iterates $\beta^{(t)}$ and $\beta^*$. For clarity, we use $\theta_0, \theta$, and $\theta'$ to denote the angles formed by $\beta^*$ with $\beta^{(0)}$ (the initial iterate), $\beta$ (the current iterate), and $\beta'$ (the next iterate), respectively. Using the coordinate representation of $\beta'$ equation 25 and equation 26, the cosine and sine of $\theta'$ can be expressed by

$$\cos\theta' = \frac{(1-\alpha)^T b_1 b_1^* + (1-(1-\alpha)^T)(S\|\beta*\|_2^2 + Rb_1^*)}{\|\beta*\|_2\sqrt{(1-\alpha)^{2T} b_1^2 + (1-(1-\alpha)^T)^2\left(R^2 + S^2\|\beta*\|_2^2 + 2RSb_1^*\right) + 2(1-\alpha)^T b_1(1-(1-\alpha)^T)\left(b_1^* S + R\right)}},$$

$$\sin\theta' = \frac{(1-\alpha)^T b_1 b_2^* + (1-(1-\alpha)^T)Rb_2^*}{\|\beta*\|_2\sqrt{(1-\alpha)^{2T} b_1^2 + (1-(1-\alpha)^T)^2\left(R^2 + S^2\|\beta*\|_2^2 + 2RSb_1^*\right) + 2(1-\alpha)^T b_1(1-(1-\alpha)^T)\left(b_1^* S + R\right)}}.$$

**Lemma A.5.** There exists a non-decreasing function $\varphi(\lambda)$ on $\lambda \in [0,1]$ such that

$$\varphi(0) = \frac{1}{\sqrt{1 + (S/R)^2\|\beta*\|_2^2 + 2(S/R)b_1^*}},$$

$$\varphi(1) = 1.$$

As long as $\theta \in \left[\frac{\pi}{3}, \frac{\pi}{2}\right)$ and $\alpha \in (0,1)$, it holds that

$$\sin\theta' \leqslant \varphi((1-\alpha)^T)\sin\theta$$

and

$$\varphi(0) = \frac{1}{\sqrt{1 + (S/R)^2\|\beta*\|_2^2 + 2(S/R)b_1^*}} \leqslant \left(\sqrt{1 + \frac{2\eta^2}{1+\eta^2}\cos^2\theta}\right)^{-1} < 1.$$

Similarly,

$$\cos\theta' \geqslant \phi((1-\alpha)^T)\cos\theta$$

where

$$\phi(0) = \sqrt{1 + \frac{b_2^{*2}(3\|\beta^*\|_2^2 + 2\vartheta^2)}{(\|\beta^*\|_2^2 + \vartheta^2)^2 + b_1^{*2}(3\|\beta^*\|_2^2 + 2\vartheta^2)}} > 1,$$

$$\phi(1) = 1.$$

*Proof.* We provide the proof for the sine case, and the proof for the cosine case follows a similar approach. Define $\lambda = (1-\alpha)^T \in (0,1]$, we have

$$\sin\theta' = \frac{b_2^*}{\|\beta^*\|_2} \frac{\lambda b_1 + (1-\lambda)R}{\sqrt{\left(\lambda b_1 + (1-\lambda)(b_1^* S + R)\right)^2 + \left(\lambda \cdot 0 + (1-\lambda)(b_2^* S)\right)^2}}$$

$$= \sin\theta \frac{\lambda b_1 + (1-\lambda)R}{\sqrt{\left(\lambda b_1 + (1-\lambda)(b_1^* S + R)\right)^2 + \left(\lambda \cdot 0 + (1-\lambda)(b_2^* S)\right)^2}}.$$

Then we define the function $\varphi(\lambda)$ to be

$$\varphi(\lambda) := \frac{\lambda b_1 + (1-\lambda)R}{\sqrt{\left(\lambda b_1 + (1-\lambda)(b_1^* S + R)\right)^2 + \left(\lambda \cdot 0 + (1-\lambda)(b_2^* S)\right)^2}}.$$

By symmetry, one can assume that the angles $\angle\langle\beta,\beta^*\rangle, \angle\langle\tilde{\beta}',\beta^*\rangle < \frac{\pi}{2}$. The non-decreasing property of $\varphi(\lambda)$ can be easily verified by the fact that $\beta'$ is located on the line segment between the current iterate $\beta$ and standard population EM updates $\tilde{\beta}$ based on equation 22. $\square$

In the remainder of this section, we discuss the convergence of the gradient population EM algorithm in terms of distance, as presented in Lemma A.1.

**Theorem A.1.** *Assume that $\theta < \pi/8$, and define $\vartheta_2^2 = \vartheta^2 + b_2^{*2}$. If $b_2^* < \vartheta$ or $\frac{\vartheta_2^2}{\vartheta^2}b_1 < b_1^*$, then we have*

$$\|\beta' - \beta^*\|_2 \leqslant \left((1-\alpha)^T + \left(1 - (1-\alpha)^T\right)\kappa\right)\|\beta - \beta^*\|_2$$

$$+ \left(1 - (1-\alpha)^T\right)\kappa\left(16\sin^3\theta\right)\|\beta^*\|_2 \frac{\eta^2}{1+\eta^2},$$

*where $\kappa = \left(\sqrt{1 + \min\left(\frac{\vartheta_2^2}{\vartheta^2}b_1, b_1^*\right)^2/\vartheta_2^2}\right)^{-1}$. Otherwise, we have*

$$\|\beta' - \beta^*\|_2 \leqslant \left((1-\alpha)^T + 0.6\left(1 - (1-\alpha)^T\right)\right)\|\beta - \beta^*\|_2.$$

*Proof.* The proof of this theorem is a direct corollary of Theorem 4 from Kwon et al. (2019) by noticing that

$$\|\beta' - \beta^*\|_2 = \left\|(1-\alpha)^T\beta + \left(1 - (1-\alpha)^T\right)\tilde{\beta}' - \beta^*\right\|_2$$

$$\leqslant (1-\alpha)^T\|\beta - \beta^*\|_2 + \left(1 - (1-\alpha)^T\right)\|\tilde{\beta}' - \beta^*\|_2.$$

$\square$

## A.2    Results in sample-based EM algorithm for MoR problem

In this section, we present results concerning the convergence of the sample-based gradient EM algorithm. We begin by deriving the update rule for the sample-based gradient EM algorithm, which incorporates $T$ steps of gradient descent. Starting from the previous estimate, $\beta^{(t-1)}$, we define $\hat{\Sigma} = \frac{1}{n}\sum_{i=1}^{n} x_i x_i^\top$. The new estimate, $\beta^{(t)}$, is obtained by applying $T$ steps of gradient descent to the loss function $\hat{L}_n^{(t-1)}(\beta)$, specifically:

$$\beta^{(t)} = \beta_T^{(t-1)}$$

$$= \left(I - \frac{\alpha}{n}\sum_{i=1}^{n}x_i x_i^\top\right)\beta_{T-1}^{(t-1)} + \frac{\alpha}{n}\sum_{i=1}^{n}\tanh\left(\frac{1}{\vartheta^2}y_i x_i^\top \beta^{(t-1)}\right)y_i x_i$$

$$= (I - \alpha\hat{\Sigma})\left[(I - \alpha\hat{\Sigma})\beta_{T-2}^{(t-1)} + \frac{\alpha}{n}\sum_{i=1}^{n}\tanh\left(\frac{1}{\vartheta^2}y_i x_i^\top \beta^{(t-1)}\right)y_i x_i\right]$$

$$\quad + \frac{\alpha}{n}\sum_{i=1}^{n}\tanh\left(\frac{1}{\vartheta^2}y_i x_i^\top \beta^{(t-1)}\right)y_i x_i$$

$$= (I - \alpha\hat{\Sigma})^T \beta^{(t-1)} + \alpha \cdot (\alpha\hat{\Sigma})^{-1}\left(I - (I - \alpha\hat{\Sigma})^T\right)\frac{1}{n}\sum_{i=1}^{n}\tanh\left(\frac{1}{\vartheta^2}y_i x_i^\top \beta^{(t-1)}\right)y_i x_i.$$

For the analysis in the remainder of this section, we denote the current iteration as $\beta$, the subsequent iteration resulting from $T$ steps of sample-based gradient descent as $\tilde{\beta}'$, and the iteration following $T$ steps of population-based gradient descent as $\beta'$. By define $\hat{\mu} := \frac{1}{n}\sum_{i=1}^{n}\tanh\left(\frac{1}{\vartheta^2}y_i x_i^\top \beta\right)y_i x_i$ and $\mu := \mathbb{E}\tanh\left(\frac{1}{\vartheta^2}YX^\top \beta\right)YX$, we have

$$\tilde{\beta}' = (I - \alpha\hat{\Sigma})^T \beta + \hat{\Sigma}^{-1}\left(I - (I - \alpha\hat{\Sigma})^T\right)\hat{\mu},$$
$$\beta' = (I - \alpha I)^T \beta + \left(I - (I - \alpha I)^T\right)\mu.$$

In the previous analysis,

$$\tilde{\beta}' - \beta^* = (I - \alpha\hat{\Sigma})^T(\beta - \beta^*) + \left(I - (I - \alpha\hat{\Sigma})^T\right)\left(\hat{\Sigma}^{-1}\hat{\mu} - \beta^*\right),$$

$$\hat{\Sigma}^{-1}\hat{\mu} - \beta^* = \hat{\Sigma}^{-1}\left(\frac{1}{n}\sum_{i=1}^{n}y_i x_i \tanh\left(\frac{y_i\langle x_i, \beta\rangle}{\vartheta^2}\right) - \mathbb{E}_y \frac{1}{n}\sum_{i=1}^{n}y_i x_i \tanh\left(\frac{y_i\langle x_i, \beta^*\rangle}{\vartheta^2}\right)\right)$$

$$= \underbrace{\hat{\Sigma}^{-1}}_{:=I}\underbrace{\left(\frac{1}{n}\sum_{i=1}^{n}y_i x_i \tanh\left(\frac{y_i\langle x_i, \beta\rangle}{\vartheta^2}\right) - \mathbb{E}_y \frac{1}{n}\sum_{i=1}^{n}y_i x_i \tanh\left(\frac{y_i\langle x_i, \beta\rangle}{\vartheta^2}\right)\right)}_{:=II}$$

$$+ \hat{\Sigma}^{-1}\underbrace{\left(\mathbb{E}_y \frac{1}{n}\sum_{i=1}^{n}y_i x_i \tanh\left(\frac{y_i\langle x_i, \beta\rangle}{\vartheta^2}\right) - \mathbb{E}_y \frac{1}{n}\sum_{i=1}^{n}y_i x_i \tanh\left(\frac{y_i\langle x_i, \beta^*\rangle}{\vartheta^2}\right)\right)}_{:=III}.$$

Then $\|I\|_{\mathrm{op}} = 1 + \mathcal{O}\left(\sqrt{\frac{d}{n}}\right)$ by standard concentration result and it requires $n \geq \mathcal{O}\left(d\log^2(1/\delta)\right)$ in the end. Conditioning on the sample covariance matrix has bounded spectral norm, $\|II\|_2 = O\left(\sqrt{\frac{d}{n}}\right)$. Finally, for each fixed $\beta$ satisfying $\|\beta\|_2 \geq \frac{\|\beta^*\|_2}{10}$, and its angle with $\beta^*$, $\theta$ is less than $\frac{\pi}{70}$, with $n = \mathcal{O}\left(\frac{d}{\epsilon^2}\right)$, $\|III\|_2 \leq \left(0.95 + \epsilon/\sqrt{d}\right)\|\beta - \beta^*\|_2$.

This can be improved by

$$\tilde{\beta}' - \beta^* = (I - \alpha\hat{\Sigma})^T \beta + \hat{\Sigma}^{-1}\left(I - (I - \alpha\hat{\Sigma})^T\right)\frac{1}{n}\sum_{i=1}^{n}\tanh\left(\frac{1}{\vartheta^2}y_i x_i^\top \beta\right)y_i x_i - \beta^*$$

$$= (I - \alpha\hat{\Sigma})^T(\beta - \beta^*) + \left(I - (I - \alpha\hat{\Sigma})^T\right)\underbrace{\left[\frac{1}{n}\sum_{i=1}^{n}\hat{\Sigma}^{-1}\tanh\left(\frac{1}{\vartheta^2}y_i x_i^\top \beta\right)y_i x_i - \beta^*\right]}_{:=A},$$

$$A = \hat{\Sigma}^{-1}\left[\underbrace{\mathbb{E}_{X,Y}\left[XY\Delta_{(X,Y)}(\beta)\right]}_{:=A_1} + \underbrace{\frac{1}{n}\sum_i X_i Y_i \Delta_{(X_i,Y_i)}(\beta) - \mathbb{E}_{X,Y}\left[XY\Delta_{(X,Y)}(\beta)\right]}_{:=A_2}\right.$$

$$+\frac{1}{n}\sum_i x_i y_i \tanh\left(y_i x_i^\top \beta^*/\vartheta^2\right) - \underbrace{\mathbb{E}_{y_i|x_i}\left[\frac{1}{n}\sum_i x_i y_i \tanh\left(y_i x_i^\top \beta^*/\vartheta^2\right)\right]}_{:=A_3}\Bigg],$$

where $\Delta_{(X,Y)}(\beta) := \tanh\left(yx^\top\beta/\vartheta^2\right) - \tanh\left(yx^\top\beta^*/\vartheta^2\right)$. Then

$$A_1 < 0.9\|\beta - \beta^*\|_2,$$

$$A_2 \leqslant (\|\beta - \beta^*\|_2 + 1)\sqrt{d\log^2\left(n\|\beta^*\|_2/\delta\right)/n},$$

$$A_3 \leqslant C\sqrt{d\log(1/\delta)/n},$$

with probability at least $1 - \delta$.

### A.3 Convergence result in Theorem 3.2 under the high SNR setting

We first present the results for parameter estimation under the high SNR regime.

**Lemma A.6** (Lemma 2 in Kwon et al. (2021))**.** For any given $r > 0$, there exists a universal constant $c > 0$ such that with probability at least $1 - \delta$.

$$\sup_{\|\beta\|_2 \leqslant r}\left\|\hat\Sigma^{-1}\hat\mu - \mu\right\|_2 \leqslant cr\sqrt{d\log^2(n/\delta)/n}$$

where

$$\mu = \mathbb{E}\left[XY\tanh\left(YX^\top\beta\right)\right],$$

$$\hat\mu = \frac{1}{n}\sum_{i=1}^n \tanh\left(\frac{y_i x_i^\top \beta}{\vartheta^2}\right)y_i x_i,$$

$$\hat\Sigma = \frac{1}{n}\sum_i x_i x_i^\top.$$

**Lemma A.7** (Lemma 5 in Kwon et al. (2021))**.** For each fixed $\beta$, with probability at least $1 - \exp(-cn) - 6^d\exp\left(-\frac{nt^2}{72}\right)$

$$\left\|\frac{1}{n}\sum_{i=1}^n y_i \boldsymbol{x}_i \tanh\left(y_i\langle x_i, \beta\rangle\right) - \frac{1}{n}\sum_{i=1}^n \mathbb{E}_{y_i}\left[y_i x_i \tanh\left(y_i\langle x_i, \beta\rangle\right)\right]\right\|_2 \leqslant t$$

for some absolute constant $c > 0$.

**Theorem A.2.** *Suppose that $\eta \geqslant \mathcal{O}\left(d\log^2(n/\delta)/n\right)^{1/4}$ for some absolute constant $C$ and $\|\beta^{(0)}\| \geqslant 0.9\|\beta^*\|$ and $\cos\angle\left(\beta^*, \beta^{(0)}\right) \geqslant 0.95.$, let $\{\beta^{(t)}\}$ be the iterates of sample-based gradient EM algorithm, then there exists a constant $\gamma_2 \in (0,1)$ such that*

$$\|\beta^{(t)} - \beta^*\|_2 \leqslant \gamma_2^t + \frac{1}{1-\gamma_2}\mathcal{O}\left(\sqrt{d\log^2(n/\delta)/n}\right)$$

*holds with probability at least $1 - 5\delta$.*

*Proof.* Without loss of generality, we can assume that $\vartheta = 1$. Denote $\beta$ as the current iterate, and $\tilde\beta'$ as the next sample-based iterate. We first consider

$$\tilde\beta' - \beta^* = (I - \alpha\hat\Sigma)^T\beta + \hat\Sigma^{-1}\left(I - (I - \alpha\hat\Sigma)^T\right)\frac{1}{n}\sum_{i=1}^n \tanh\left(\frac{1}{\vartheta^2}y_i x_i^\top\beta\right)y_i x_i - \beta^*$$

$$= (I - \alpha\hat\Sigma)^T(\beta - \beta^*) + \left(I - (I - \alpha\hat\Sigma)^T\right)\underbrace{\left[\frac{1}{n}\sum_{i=1}^n \hat\Sigma^{-1}\tanh\left(\frac{1}{\vartheta^2}y_i x_i^\top\beta\right)y_i x_i - \beta^*\right]}_{:=A}.$$

We prove the results in two cases, i.e. $\eta \geqslant 1$ and $C_0\big(d\log^2(n/\delta)/n\big)^{1/4} \leqslant \eta \leqslant 1$ for some universal constant $C_0$. When $\eta \geqslant 1$, based on the analysis in Kwon et al. (2021), with probability at least $1 - 5\delta$,

$$\|A\|_2 \leqslant \Big(0.9 + \sqrt{d\log^2\left(n\|\beta^*\|_2/\delta)/n\right)}\Big)\|\beta - \beta^*\| + C_1\sqrt{d\log^2\left(n\|\beta^*\|_2/\delta)/n\right)}$$

$$\leqslant \gamma\|\beta - \beta^*\|_2 + C_1\sqrt{d\log^2\left(n\|\beta^*\|_2/\delta)/n\right)} \tag{32}$$

where $\gamma = 0.9 + \sqrt{d\log^2\left(n\|\beta^*\|_2/\delta)/n\right)}$. By standard concentration results on $\hat{\Sigma} - I$, it holds that with $n \geqslant \mathcal{O}(d\log^2(1/\delta))$, $\|(I - \alpha\hat{\Sigma})^T\|_{\text{op}} \leqslant (1 - \alpha/2)^T$ with probability at least $1 - \delta$ for appropriately small $\alpha$. Along with equation 32,

$$\|\tilde{\beta}' - \beta^*\|_2 \leqslant \big(1 - \frac{\alpha}{2}\big)^T\|\beta - \beta^*\|_2 + \big(1 - \big(1 - \frac{\alpha}{2}\big)^T\big)\|A\|_2$$

$$\leqslant \Big[\big(1 - \frac{\alpha}{2}\big)^T + \big(1 - \big(1 - \frac{\alpha}{2}\big)^T\big)\gamma\Big]\|\beta - \beta^*\|_2$$

$$+ \big(1 - \big(1 - \frac{\alpha}{2}\big)^T\big)C_1\sqrt{d\log^2\left(n\|\beta^*\|_2/\delta)/n\right)}. \tag{33}$$

Define $\epsilon(n, \delta) = \big(1 - \big(1 - \frac{\alpha}{2}\big)^T\big)C_1\sqrt{d\log^2\left(n\|\beta^*\|_2/\delta)/n\right)}$ and $\gamma_2 = \big(1 - \frac{\alpha}{2}\big)^T + \big(1 - \big(1 - \frac{\alpha}{2}\big)^T\big)\gamma$. As long as $\gamma < 1$, we can iterate over $t$ based on equation 33 and obtain

$$\|\beta^{(t)} - \beta^*\| \leqslant \gamma_2\|\beta^{(t-1)} - \beta^*\|_2 + \epsilon(n, \delta) \leqslant \gamma_2^2\|\beta^{(t-2)} - \beta^*\|_2 + (1 + \gamma_2)\epsilon(n, \delta)$$

$$\leqslant \gamma_2^t\|\beta^{(0)} - \beta^*\|_2 + \frac{1}{1 - \gamma_2}\epsilon(n, \delta).$$

In the remaining part of the proof, we present an analysis of the convergence behavior of the sample-based gradient EM algorithm when $C_0\big(d\log^2(n/\delta)/n\big)^{1/4} \leqslant \eta \leqslant 1$. By Lemma 3 from Kwon et al. (2021), it holds that

$$\big\|\mathbb{E}\big[\tanh\big(YX^\top\beta\big)YX\big] - \beta^*\big\|_2 \leqslant \big(1 - \frac{1}{8}\|\beta^*\|_2^2\big)\|\beta - \beta^*\|_2.$$

To systematically analyze the convergence, we categorize the iterations into several epochs. We define $\bar{C}_0 = \|\beta^{(0)} - \beta^*\|_2$ and assume that during each $l^{\text{th}}$ epoch, the distance $\|\beta - \beta^*\|_2$ lies within the interval $[\bar{C}_0 2^{-l-1}, \bar{C}_0 2^{-l}]$. This stratification is conceptual and does not impact the practical implementation of the EM algorithm. The key idea in this part is the same as Kwon et al. (2021). During the $l^{\text{th}}$ epoch, the improvement in the population gradient EM updates must exceed the statistical error for convergence to occur, formalized as:

$$\frac{1}{8}\big(1 - (1 - \frac{\alpha}{2})^T\big)\|\beta^*\|_2^2\|\beta - \beta^*\|_2 \geqslant 2cr\sqrt{d\log^2(n/\delta)/n}$$

where $c$ is the constant in Lemma A.6. By setting $r = \|\beta^*\| + \bar{C}_0 2^{-l}$ and using triangle inequality $\|\beta\|_2 \leqslant \|\beta^*\|_2 + \|\beta - \beta^*\|_2$, in $l^{\text{th}}$ epoch when

$$\frac{1}{8}\big(1 - (1 - \frac{\alpha}{2})^T\big)\|\beta^*\|^2\bar{C}_0 2^{-l-1} \geqslant 2cr\sqrt{d\log^2(n/\delta)/n}$$

$$\geqslant 4c\big(\|\beta^*\| + \bar{C}_0 2^{-l}\big)\sqrt{d\log^2(n/\delta)/n},$$

is guaranteed to be true, then it holds that

$$\|A\|_2 \leqslant \big(1 - \frac{1}{16}\|\beta^*\|_2^2\big)\|\beta - \beta^*\|_2$$

$$\|\beta' - \beta^*\|_2 \leqslant \Big[\big(1 - \frac{\alpha}{2}\big)^T + \big(1 - \big(1 - \frac{\alpha}{2}\big)^T\big)\big(1 - \frac{1}{16}\|\beta^*\|_2^2\big)\Big]\|\beta - \beta^*\|_2.$$

Recall that $\eta \geqslant \mathcal{O}\big((d\log^2(n/\delta)/n)^{\frac{1}{4}}\big)$, then with appropriately set constants

$$\|\beta^*\|^2 \geqslant (c_1 + 1)\sqrt{d\log^2(n/\delta)/n},$$

we can deduce that $\beta$ moves progressively closer to $\beta^*$ as long as $\bar{C}_0 2^{-l} \leqslant c_2\|\beta^*\|_2^{-1}\sqrt{d\log^2(n/\delta)/n}$. This process requires $\mathcal{O}\big(\|\beta^*\|_2^{-2}\big)$ iterations per epoch, and after $\mathcal{O}(\log(n/d))$ epochs, the error bound $\|\beta - \beta^*\|_2 \leqslant c_2\|\beta^*\|_2^{-1}\sqrt{\frac{d\log^2(n/\delta)}{n}}$ is expected to hold. Thus, the convergence rate for $\beta^{(t)}$ towards $\beta^*$ is quantified as:

$$\|\beta^{(t)} - \beta^*\|_2 \leqslant \gamma_2^t\|\beta^{(0)} - \beta^*\|_2 + \frac{1}{1-\gamma_2}\sqrt{d\log^2(n/\delta)/n}.$$

$\square$

## A.4 Convergence result in Theorem 3.1 under Low SNR settings

We present several auxiliary lemmas that will be utilized in analyzing the convergence results for sample-based gradient EM iterates.

**Lemma A.8** (Lemma 6 in Kwon et al. (2021)). *There exists some universal constants $c_u > 0$ such that,*

$$\|\beta\|_2\big(1 - 4\|\beta\|_2^2 - c_u\|\beta^*\|_2^2\big) \leqslant \big\|\mathbb{E}\big[\tanh\big(YX^\top\beta\big)YX\big]\big\|_2 \leqslant \|\beta\|_2\big(1 - \|\beta\|_2^2 + c_u\|\beta^*\|_2^2\big).$$

**Theorem A.3.** *When $\eta \leqslant C_0(d\log^2(n/\delta)/n)^{1/4}$, there exist universal constants $C_3, C_4 > 0$ such that the sample-based gradient EM updates initialized with $\|\beta^{(0)}\|_2 \leqslant 0.2$ return $\beta^{(t)}$ that satisfies*

$$\|\beta^{(t)} - \beta^*\|_2 \leqslant \mathcal{O}\Big(\big(d\log^2 n/n\big)^{\frac{1}{4}}\Big)$$

*with probability at least $1 - \delta$ after $t \geqslant C_4\big(1 - \big(1 - \alpha/2\big)^T\big)^{-1}\log(\log(n/d))\sqrt{n/\big(d\log^2(n/\delta)\big)}$ iterations.*

*Proof.* The proof argument follows the similar localization argument used in Theorem A.2. Define $\epsilon(n, \delta) := C\sqrt{d\log^2(n/\delta)/n}$ with some absolute constant $C > 0$. We assume that we start from the initialization region where $\|\beta\|_2 \leqslant \epsilon^{\alpha_0}(n, \delta)$ for some $\alpha_0 \in [0, 1/2]$. Suppose that $\epsilon^{\alpha_{l+1}}(n, \delta) \leqslant \|\beta\|_2 \leqslant \epsilon^{\alpha_l}(n, \delta)$ at the $l^{\text{th}}$ epoch for $l \geqslant 0$. We let $C > 0$ sufficiently large such that

$$\epsilon(n, \delta) \geqslant 4c_u\|\beta^*\|_2^2 + 4\sup_{\beta \in \mathbb{B}(\beta^*, r_l)}\big\|\mu - \hat{\Sigma}^{-1}\hat{\mu}\big\|_2/r_l$$

with $r_l = \epsilon_n^{\alpha_l}$. During this period, from Lemma A.8 on contraction of population EM, and Lemma A.6 concentration of finite sample EM, we can check that

$$\|\hat{\Sigma}^{-1}\hat{\mu}\|_2 \leqslant \|\beta\|_2 - 0.5\|\beta\|_2^3 + c_u\|\beta\|_2\|\beta^*\|_2^2 + \sup_{\beta \in \mathbb{B}(\beta^*, r)}\big\|\mu - \hat{\Sigma}^{-1}\hat{\mu}\big\|$$

$$\leqslant \|\beta\|_2 - \frac{1}{2}\epsilon^{3\alpha_{l+1}}(n, \delta) + \frac{1}{4}\epsilon^{\alpha_{l+1}}(n, \delta),$$

$$\|\tilde{\beta}'\|_2 \leqslant \big(1 - \frac{\alpha}{2}\big)^T\|\beta\|_2 + \big(1 - \big(1 - \frac{\alpha}{2}\big)^T\big)\|\hat{\Sigma}^{-1}\hat{\mu}\|_2$$

$$\leqslant \|\beta\|_2 + \big(1 - \big(1 - \frac{\alpha}{2}\big)^T\big)\Big[-\frac{1}{2}\epsilon^{3\alpha_{l+1}}(n, \delta) + \frac{1}{4}\epsilon^{\alpha_{l+1}}(n, \delta)\Big].$$

Note that this inequality is valid as long as $\epsilon^{\alpha_{l+1}}(n, \delta) \leqslant \|\beta\|_2 \leqslant \epsilon^{\alpha_l}(n, \delta)$. Now we define a sequence $\alpha_l$ by

$$\alpha_l = (1/3)^l(\alpha_0 - 1/2) + 1/2$$

and $\alpha_l \to 1/2$ as $l \to \infty$. With this choice of $\alpha_l$, $\epsilon_n^{\alpha_l} \to (d/n)^{1/4}$. Hence during the $l^{th}$ epoch, we have

$$\|\tilde{\beta}'\|_2 \leqslant \|\beta\|_2 - \frac{1}{4}\big(1 - \big(1 - \frac{\alpha}{2}\big)^T\big)\epsilon^{\alpha_{l+1}}(n, \delta).$$

Furthermore, the number of iterations required in $l^{\text{th}}$ epoch is

$$t_l := \frac{\left(\epsilon^{\alpha_l}(n,\delta) - \epsilon^{\alpha_{l+1}}(n,\delta)\right)}{\left(1 - \left(1 - \frac{\alpha}{2}\right)^T\right)\epsilon^{\alpha_{l+1}}(n,\delta)} \leqslant \left(1 - \left(1 - \frac{\alpha}{2}\right)^T\right)^{-1}\epsilon^{-1}(n,\delta).$$

When it gets into $(l+1)^{th}$ epoch. the behavior can be analyzed in the same way and after going through $l$ epochs in total, we have $\|\beta\|_2 \leqslant \epsilon^{\alpha_{l+1}}(n,\delta)$. At this point, the total number of iterations (counted in terms of steps of gradient descent) is bounded by

$$l\left(1 - \left(1 - \frac{\alpha}{2}\right)^T\right)^{-1}\epsilon^{-1}(n,\delta).$$

By taking $l = C\left(1 - \left(1 - \alpha/2\right)^T\right)\log(1/\theta)$ for some universal constant $C$ such that $\alpha_l$ is $1/2 - \theta$ for arbitrarily small $\theta > 0$, it holds that

$$\|\beta^{(t)}\|_2 \leqslant \epsilon^{1/2-\theta}(n,\delta) \leqslant c\left(d\log^2(n/\delta)/n\right)^{1/4-\theta/2}$$

with high probability as long as $t \geqslant \epsilon^{-1}(n,\delta)l \gtrsim \sqrt{d/n}\left(1 - \left(1 - \alpha/2\right)^T\right)\log(1/\theta)$ where $c$ is some universal constant. By taking $\theta = C/\log(d/n)$ and using triangle inequalities, it holds that $\|\beta^{(t)}\|_2 \leqslant c\left(d\log^2(n/\delta)/n\right)^{1/4}$ and $\|\beta^{(t)} - \beta^*\|_2 \leqslant c_1\left(d\log^2(n/\delta)/n\right)^{1/4}$ where $c_1$ is some universal constant under low SNR settings.

To finish the proof, we replace $\delta$ by $\delta/\log(1/\theta)$ and take the union bound of the concentration of sample gradient EM operators for all $l = 1, \ldots, C\left(1 - \left(1 - \alpha/2\right)^T\right)\log(1/\theta)$, such that the argument holds for all epochs. $\qquad\square$

### A.5 Proof of Theorem 3.2

*Proof of Theorem 3.2.* The existence of the transformer follows directly from Lemma A.2 and Lemma A.3. $\qquad\square$

### A.6 Proof of Theorem 3.3

For the data generated based on model equation 1 with two components, $\beta_{n+1} = -\beta^*$ with probability $\frac{1}{2}$ and $\beta_{n+1} = \beta^*$ with probability $\frac{1}{2}$. For any choice of $\beta \in \mathbb{R}^d$,

$$\begin{aligned}
\mathbb{E}_{\mathcal{P}_{x,y}}\left[(y_{n+1} - x_{n+1}^\top\beta)^2\right] &= \mathbb{E}_{\mathcal{P}_{x,y}}\left[(x_{n+1}^\top\beta_{n+1} - x_{n+1}^\top\beta + v_{n+1})^2\right] \\
&= \vartheta^2 + \mathbb{E}_{\mathcal{P}_{x,y}}\left[\left(x_{n+1}^\top\beta_{n+1} - x_{n+1}^\top\beta\right)^2\right] \\
&= \vartheta^2 + \mathbb{E}_{\mathcal{P}_{x,y}}\operatorname{tr} x_{n+1}x_{n+1}^\top(\beta_{n+1} - \beta)(\beta_{n+1} - \beta)^\top \\
&= \vartheta^2 + \mathbb{E}_{\mathcal{P}_{x,y}}\operatorname{tr}(\beta_{n+1} - \beta)(\beta_{n+1} - \beta)^\top \\
&= \vartheta^2 + \mathbb{E}_{\mathcal{P}_{x,y}}\|\beta_{n+1} - \beta\|_2^2 \\
&= \vartheta^2 + \frac{1}{2}\|\beta^* - \beta\|_2^2 + \frac{1}{2}\|\beta^* + \beta\|_2^2.
\end{aligned}$$

Therefore, $\mathbb{E}_{\mathcal{P}_{x,y}}\left[(y_{n+1} - x_{n+1}^\top)^2\right]$ is minimized at

$$\beta^{\mathsf{OR}} := \frac{1}{2}\beta^* - \frac{1}{2}\beta^* \tag{34}$$

the optimal risk is given by $\vartheta^2 + \|\beta^*\|_2^2$. And same results holds if the estimator $\beta$ depends on previous training instance $(x_1, y_1, \ldots, x_n, y_n)$ and the expectation is taken w.r.t. $\mathcal{P}$. And in general MoR problem, the vector that minimizes the mean squared error of the prediction is given by

$$\beta^{\mathsf{OR}} := \arg\min_{\beta\in\mathbb{R}^d}\mathbb{E}_{\mathcal{P}_{x,y}}\left[(x^\top\beta - y)^2\right] = \sum_{\ell=1}^K \pi_\ell^*\beta_\ell^*.$$

*Proof of Theorem 3.3.* The oracle estimator that minimizes the MSE, i.e. $\mathrm{MSE}(f) = \mathbb{E}_{\mathcal{P}}\big[\big(f(H) - y_{n+1}\big)^2\big]$ is given by equation 34. The output of the transformer is given by

$$\hat{y}_{n+1} = \mathrm{read}_y\big(\mathrm{TF}(H)\big) = x_{n+1}^{\top}\hat{\beta}^{\mathsf{OR}}$$

where $\hat{\beta}^{\mathsf{OR}}$ is given by

$$\hat{\beta}^{\mathsf{OR}} := \pi_1\hat{\beta} - (1 - \pi_1)\hat{\beta}$$

with $\hat{\beta} = \mathrm{read}_\beta(\mathrm{TF}(H))$ for $L = \mathcal{O}\Big(T\big(1 - \big(1 - \alpha/2\big)^T\big)^{-1}\log(\log(n/d))\sqrt{n/\big(d\log^2(n/\delta)\big)}\Big)$ in the low SNR settings and $\mathcal{O}\Big(T\log\big(\frac{n\log n}{d}\big)\Big)$ in the high SNR settings. Note that $\|\hat{\beta}^{\mathsf{OR}} - \beta^{\mathsf{OR}}\|_2 \leqslant \pi_1\|\beta^* - \hat{\beta}\|_2 + (1 - \pi_1)\|\beta^* - \hat{\beta}\|_2 \leqslant \|\beta^* - \hat{\beta}\|_2$.

- Under thelow SNR regime, after $T_0 \geqslant \mathcal{O}\Big(\log(\log(n/d))\sqrt{n/\big(d\log^2(n/\delta)\big)}\Big)$ outer loops,

$$\|\beta^{\mathsf{OR}} - \hat{\beta}^{\mathsf{OR}}\|_2 \leqslant \mathcal{O}\Big(\Big(\frac{d\log(n/\delta)}{n}\Big)^{\frac{1}{4}}\Big)$$

with probability at least $1 - 5\delta$.

- Under the high SNR regime, after $T_0 \geqslant \mathcal{O}\Big(\log\big(\frac{n\log n}{d}\big)\Big)$ outer loops,

$$\|\beta^{\mathsf{OR}} - \hat{\beta}^{\mathsf{OR}}\|_2 \leqslant \mathcal{O}\Bigg(\sqrt{\frac{d\log^2(n/\delta)}{n}}\Bigg)$$

with probability at least $1 - 5\delta$.

We would like to bound

$$\mathbb{E}_{\mathcal{P}}\Big[\big(y_{n+1} - \mathrm{read}_y(\mathrm{TF}(H))\big)^2\Big] - \inf_{\beta}\mathbb{E}_{\mathcal{P}}\Big[\big(x_{n+1}^{\top}\beta - y_{n+1}\big)^2\Big].$$

Note that the $\mathbb{E}_{\mathcal{P}}\Big[\big(y_{n+1} - \mathrm{read}_y(\mathrm{TF}(H))\big)^2\Big]$ is given by

$$\mathbb{E}_{\mathcal{P}}\Big[\big(x_{n+1}^{\top}\hat{\beta}^{\mathsf{OR}} - y_{n+1}\big)^2\Big] = \mathbb{E}_{\mathcal{P}}\Big[\big(x_{n+1}^{\top}\big(\hat{\beta}^{\mathsf{OR}} - \beta^{\mathsf{OR}} + \beta^{\mathsf{OR}}\big) - y_{n+1}\big)^2\Big]$$
$$=\mathbb{E}_{\mathcal{P}}\Big[\big(x_{n+1}^{\top}\big(\hat{\beta}^{\mathsf{OR}} - \beta^{\mathsf{OR}}\big)\big)^2\Big] + 2\mathbb{E}_{\mathcal{P}}\Big[\big(\hat{\beta}^{\mathsf{OR}} - \beta^{\mathsf{OR}}\big)^{\top}x_{n+1}\big(x_{n+1}^{\top}\beta^{\mathsf{OR}} - y_{n+1}\big)\Big] + \mathbb{E}_{\mathcal{P}}\Big[\big(x_{n+1}^{\top}\beta^{\mathsf{OR}} - y_{n+1}\big)^2\Big].$$

Hence, when $\pi_1 = \pi_2 = \frac{1}{2}$, $\beta^{\mathsf{OR}} = \pi_1\beta^* - \pi_2\beta^* = 0$,

$$\mathbb{E}_{\mathcal{P}}\Big[\big(y_{n+1} - \mathrm{read}_y(\mathrm{TF}(H))\big)^2\Big] - \inf_{\beta}\mathbb{E}_{\mathcal{P}}\Big[\big(x_{n+1}^{\top}\beta - y_{n+1}\big)^2\Big]$$
$$=\mathbb{E}_{\mathcal{P}}\Big[\big(x_{n+1}^{\top}\big(\hat{\beta}^{\mathsf{OR}} - \beta^{\mathsf{OR}}\big)\big)^2\Big] + 2\mathbb{E}_{\mathcal{P}}\Big[\big(\hat{\beta}^{\mathsf{OR}} - \beta^{\mathsf{OR}}\big)^{\top}x_{n+1}x_{n+1}^{\top}\beta^{\mathsf{OR}}\Big]$$
$$=\mathbb{E}_{\mathcal{P}}\Big[\big(\hat{\beta}^{\mathsf{OR}} - \beta^{\mathsf{OR}}\big)^{\top}x_{n+1}x_{n+1}^{\top}\big(\hat{\beta}^{\mathsf{OR}} - \beta^{\mathsf{OR}}\big)\Big]$$
$$=\mathbb{E}_{\mathcal{P}}\Big[\mathrm{tr}\Big(x_{n+1}x_{n+1}^{\top}\big(\hat{\beta}^{\mathsf{OR}} - \beta^{\mathsf{OR}}\big)\big(\hat{\beta}^{\mathsf{OR}} - \beta^{\mathsf{OR}}\big)^{\top}\Big)\Big]$$
$$=\mathbb{E}_{\mathcal{P}}\big\|\hat{\beta}^{\mathsf{OR}} - \beta^{\mathsf{OR}}\big\|_2^2.$$

- Under the high SNR settings, it holds that

$$\mathbb{P}\Big(\|\hat{\beta}^{\mathsf{OR}} - \beta^{\mathsf{OR}}\|_2 \leqslant \mathcal{O}\Big(\sqrt{d\log^2(n/\delta)/n}\Big)\Big) \geqslant 1 - \delta.$$

Hence, by integrating the tail probabilities we have

$$
\begin{aligned}
\mathbb{E}\|\hat{\beta}^{\mathsf{OR}} - \beta^{\mathsf{OR}}\|_2^2 &= \int_0^{+\infty} \mathbb{P}\big(\|\hat{\beta}^{\mathsf{OR}} - \beta^{\mathsf{OR}}\|_2 \geqslant \sqrt{t}\big)dt \\
&= \Big[\int_0^{c_1} + \int_{c_1}^{+\infty}\Big]\mathbb{P}\big(\|\hat{\beta}^{\mathsf{OR}} - \beta^{\mathsf{OR}}\|_2 \geqslant \sqrt{t}\big)dt \\
&\leqslant \int_0^{c_1} 1 dt + \int_{c_1}^{+\infty} \mathbb{P}\big(\|\hat{\beta}^{\mathsf{OR}} - \beta^{\mathsf{OR}}\|_2 \geqslant \sqrt{t}\big)dt \\
&\leqslant c_1 + \int_{c_1}^{+\infty} \mathbb{P}\big(\|\hat{\beta}^{\mathsf{OR}} - \beta^{\mathsf{OR}}\|_2 \geqslant \sqrt{t}\big)dt.
\end{aligned}
$$

Setting $\sqrt{t} = \mathcal{O}\big(\sqrt{d\log^2(n/\delta)/n}\big)$ and solving for $\delta$ give us $\delta \leqslant n\exp\big\{-\sqrt{nt/d}\big\}$. By taking $c_1 = \frac{Cd\log^2 n}{n}$ for some absolute constant $C$, it holds that

$$
\begin{aligned}
\mathbb{E}\|\hat{\beta}^{\mathsf{OR}} - \beta^{\mathsf{OR}}\|_2^2 &\leqslant \mathcal{O}\bigg(\frac{d\log^2 n}{n}\bigg) + \int_{c_1}^{+\infty} n\exp\big\{-\sqrt{nt/d}\big\}dt \\
&= \mathcal{O}\bigg(\frac{d\log^2 n}{n}\bigg) + \mathcal{O}\bigg(\frac{(2d+1)\log n}{n}\bigg) \\
&= \mathcal{O}\bigg(\frac{d\log^2 n}{n}\bigg).
\end{aligned}
$$

- Under the low SNR settings, it holds that

$$
\mathbb{P}\Big(\|\hat{\beta}^{\mathsf{OR}} - \beta^{\mathsf{OR}}\|_2 \leqslant \mathcal{O}\big(d^{\frac{1}{4}}\log^{\frac{1}{2}}(n/\delta)/n^{\frac{1}{4}}\big)\Big) \geqslant 1 - \delta.
$$

Hence,

$$
\begin{aligned}
\mathbb{E}\|\hat{\beta}^{\mathsf{OR}} - \beta^{\mathsf{OR}}\|_2^2 &= \int_0^{+\infty} \mathbb{P}\big(\|\hat{\beta}^{\mathsf{OR}} - \beta^{\mathsf{OR}}\|_2 \geqslant \sqrt{t}\big)dt \\
&= \Big[\int_0^{c_1} + \int_{c_1}^{+\infty}\Big]\mathbb{P}\big(\|\hat{\beta}^{\mathsf{OR}} - \beta^{\mathsf{OR}}\|_2 \geqslant \sqrt{t}\big)dt \\
&\leqslant \int_0^{c_1} 1 dt + \int_{c_1}^{+\infty} \mathbb{P}\big(\|\hat{\beta}^{\mathsf{OR}} - \beta^{\mathsf{OR}}\|_2 \geqslant \sqrt{t}\big)dt \\
&\leqslant c_1 + \int_{c_1}^{+\infty} \mathbb{P}\big(\|\hat{\beta}^{\mathsf{OR}} - \beta^{\mathsf{OR}}\|_2 \geqslant \sqrt{t}\big)dt.
\end{aligned}
$$

Similarly, setting $\sqrt{t} = \mathcal{O}\big(d^{\frac{1}{4}}\log^{\frac{1}{2}}(n/\delta)/n^{\frac{1}{4}}\big)$ and solving for $\delta$ give us $\delta \leqslant n\exp\big\{-\sqrt{n/dt}\big\}$. By taking $c_1 = C\sqrt{d\log^2 n/n}$ for some absolute constant $C$, it holds that

$$
\begin{aligned}
\mathbb{E}\|\hat{\beta}^{\mathsf{OR}} - \beta^{\mathsf{OR}}\|_2^2 &\leqslant \mathcal{O}\bigg(\sqrt{\frac{d\log^2 n}{n}}\bigg) + \int_{c_1}^{+\infty} n\exp\big\{-d^{-\frac{1}{4}}\sqrt{t}\big\}dt \\
&= \mathcal{O}\big(\sqrt{d/n}\log n\big).
\end{aligned}
$$

Combining everything together, it holds that

$$
\mathbb{E}_{\mathcal{P}}\Big[(y_{n+1} - \mathrm{read}_y(\mathrm{TF}(H)))^2\Big] - \inf_\beta \mathbb{E}_{\mathcal{P}}\big[(x_{n+1}^\top\beta - y_{n+1})^2\big]
$$

$$
= \begin{cases} \mathcal{O}\left(\frac{d \log^2 n}{n}\right) & \eta \geqslant C\big(d \log^2(n/\delta)/n\big)^{\frac{1}{4}} \\ \mathcal{O}\left(\sqrt{d/n} \log n\right) & \eta \leqslant C\big(d \log^2(n/\delta)/n\big)^{\frac{1}{4}} \end{cases}.
$$

$\square$

### A.7 Proof of Theorem 3.1

In this section, we illustrate the existence of a transformer that can solve MoR problem with $K \geqslant 3$ components in general. Given the input matrix $H$ as equation 3 and initialization of $\pi_j^{(0)} = \frac{1}{K}$, there exists a transformer that implements $E$-steps and computes

$$
\gamma_{ij}^{(t+1)} = \frac{\pi_j^{(t)} \prod_{\ell=1}^{n} \exp\left(-\frac{1}{2\vartheta^2}\big(y_\ell - x_\ell^\top \beta_j^{(t)}\big)^2\right)}{\sum_{j'=1}^{k} \pi_{j'}^{(t)} \prod_{\ell=1}^{n} \exp\left(-\frac{1}{2\vartheta^2}\big(y_\ell - x_\ell^\top \beta_{j'}^{(t)}\big)^2\right)}, \quad \pi_j^{(t+1)} = \frac{1}{n} \sum_{i=1}^{n} \gamma_{ij}^{(t+1)}, \tag{35}
$$

since the computation in equation 35 only contains scalar product, linear transformation and softmax operation. Next, following same procedure as before, one can construct $T$ attention layers that implement gradient descent of the optimization problem

$$
\min_{\beta \in \mathbb{R}^d} \left\{ \sum_{i=1}^{K} \sum_{\ell=1}^{n} \gamma_{ij}^{(t+1)} \big(y_\ell - \beta^\top x_\ell\big)^2 \right\}, \quad \text{for all } j \in [K],
$$

as the gradient of loss $l(x_\ell^\top \beta, y_\ell) := \sum_{i=1}^{k} \gamma_{ij}^{(t+1)} \big(y_\ell - \beta^\top x_\ell\big)^2$ is convex in first argument and $\partial_s l(s, t)$ is $(0, +\infty, 4, 16)$ approximable by sum of ReLUs. Hence, the construction in Lemma A.2 and Lemma A.3 also holds.

Given the estimate $\beta_j^{(t-1)}$ and $\pi_j^{(t-1)}$, at step $t-1$, the population EM algorithm is defined by the updates

$$
w_j^{(t)}(X, Y) = \frac{\pi_j^{(t-1)} \exp\left\{-\frac{1}{2}\big(Y - X^\top \beta_j^{(t-1)}\big)^2\right\}}{\sum_{\ell \in [K]} \pi_\ell^{(t-1)} \exp\left\{-\frac{1}{2}\big(Y - X^\top \beta_\ell\big)^2\right\}},
$$
$$
\tilde{\beta}_j^{(t)} = \big(\mathbb{E}\big[w_j^{(t)}(X, Y) X X^\top\big]\big)^{-1} \big(\mathbb{E}\big[w_j^{(t)}(X, Y) X Y\big]\big),
$$
$$
\tilde{\pi}_j^{(t)} = \mathbb{E}\big[w_j^{(t)}(X, Y)\big].
$$

In the sample version of the gradient EM algorithm, we define $\hat{\Sigma}_w^{(t)} = \frac{1}{n} \sum_{i=1}^{n} w_{ij}^{(t)}(x_i, y_i) x_i x_i^\top$. The new estimate, $\beta^{(t)}$, is obtained by applying $L$ steps of gradient descent to the loss function

$$
\hat{L}_n^{(t)}(\beta) = \frac{1}{2n} \sum_{i=1}^{n} w_{ij}^{(t)}(x_i, y_i) \big(y_i - x_i^\top \beta\big)^2
$$

starting from $\beta_j^{(t-1)}$. Specifically,

$$
\beta_j^{(t)} = \big(I - \alpha \hat{\Sigma}_w^{(t)}\big)^T \beta_j^{(t-1)} + \big(I - \big(I - \alpha \hat{\Sigma}_w^{(t)}\big)^T\big) \frac{1}{n} \sum_{i=1}^{n} \big[\hat{\Sigma}_w^{(t)}\big]^{-1} w_{ij}^{(t)}(x_i, y_i) y_i x_i.
$$

In the finite sample gradient version of EM, the estimation error at the next iteration in this problem is

$$
\beta_j^{(t)} - \beta_j^* = \big(I - \alpha \hat{\Sigma}_w^{(t)}\big)^T \big(\beta_j^{(t-1)} - \beta_j^*\big) + \big(I - \big(I - \alpha \hat{\Sigma}_w^{(t)}\big)^T\big) \left[\frac{1}{n} \sum_{i=1}^{n} \big[\hat{\Sigma}_w^{(t)}\big]^{-1} w_{ij}^{(t)}(x_i, y_i) y_i x_i - \beta_j^*\right].
$$

Define

$$
w_j^*(X, Y) = \frac{\pi_j^* \exp\left(-\frac{1}{2}\big(Y - X^\top \beta_j^*\big)^2\right)}{\sum_{l=1}^{K} \pi_j^* \exp\left(-\frac{1}{2}\big(Y - X^\top \beta_j^*\big)^2\right)},
$$

then we have

$$\mathbb{E}\big[w_j^*(X,Y)X\big(Y - X^\top \beta_j^*\big)\big] = \pi_1^* \mathbb{E}\big[X\big(Y - X^\top \beta_j^*\big)\big] = 0,$$

since true parameters are a fixed point of the EM iteration. Hence,

$$\beta_j^{(t)} - \beta_j^* = \big(I - \alpha \hat{\Sigma}_w^{(t-1)}\big)^T \big(\beta_j^{(t-1)} - \beta_j^*\big) + \Big(I - \big(I - \alpha \hat{\Sigma}_w^{(t)}\big)^T\Big)\big(\hat{\Sigma}_w^{(t)}\big)^{-1}\big[e_B + B\big],$$

$$e_B = \frac{1}{n}\sum_{i=1}^n w_{ij}^{(t)}(x_i, y_i)\big(y_i - x_i^\top \beta_j^*\big)x_i - \mathbb{E}\big[w_j^{(t)}(X,Y)X\big(Y - X^\top \beta_j^*\big)\big],$$

$$B = \mathbb{E}\big[w_j^{(t)}(X,Y)X\big(Y - X^\top \beta_j^*\big)\big] - \mathbb{E}\big[w_j^*(X,Y)X\big(Y - X^\top \beta_j^*\big)\big].$$

In Kwon & Caramanis (2020), the following results are proved.

**Lemma A.9** ((Kwon & Caramanis, 2020)). Under SNR condition

$$\eta \geqslant CK\rho_\pi \log(K\rho_\pi)$$

with sufficiently large $C > 0$ and initialization condition

$$\max_\ell |\pi_\ell^{(t-1)} - \pi_\ell^*| \leqslant \frac{\pi_{\min}}{2},$$
$$\max_\ell \big\|\beta_\ell^{(t-1)} - \beta_\ell^*\big\|_2 \leqslant \frac{c\eta}{K\rho_\pi \log(K\rho_\pi)},$$

for sufficiently small $c > 0$. Given $n \geqslant \mathcal{O}\big(\max\big\{d\log^2(dK^2/\delta), \big(K^2/\delta\big)^{1/3}\big\}\big)$ samples, we get

$$\|e_B\|_2 \leqslant \sqrt{\frac{K\pi_j^{*2}}{\pi_{\min}}}\sqrt{\frac{d}{n}\log^2\big(nK^2/\delta\big)}\max_\ell \big\|\beta_\ell^{(t-1)} - \beta_l^*\big\|_2 + \sqrt{\frac{K\pi_j^{*2}}{\pi_{\min}}}\sqrt{\frac{d}{n}\log^2\big(nK^2/\delta\big)}$$

with probability at least $1 - \delta$.

**Lemma A.10** ((Kwon & Caramanis, 2020)). Under SNR condition

$$\eta \geqslant CK\rho_\pi \log(K\rho_\pi)$$

with sufficiently large $C > 0$ and initialization condition

$$\max_\ell |\pi_\ell^{(t-1)} - \pi_\ell^*| \leqslant \frac{\pi_{\min}}{2},$$
$$\max_\ell \big\|\beta_\ell^{(t-1)} - \beta_\ell^*\big\|_2 \leqslant \frac{c\eta}{K\rho_\pi \log(K\rho_\pi)},$$

for sufficiently small $c > 0$. There exits some universal constant $c_B' \in (0, 1/2)$

$$B \leqslant c_B' \pi_j^* \max_\ell \big\|\beta_\ell^{(t-1)} - \beta_\ell^*\big\|_2.$$

Now, it remains to bound the maximum eigenvalue and minimum eigenvalue of the weighted sample covariance matrix $\hat{\Sigma}_w^{(t)}$. Define the event

$$\mathcal{E}_j = \big\{\text{the sample comes from } j\text{-th component}\big\}.$$

and $\rho_{j\ell} := \pi_\ell^*/\pi_j^*$ for $j \neq \ell$. Note that

$$\frac{1}{n}\sum_{i=1}^n w_{ij}^{(t)}(x_i, y_i)x_i x_i^\top 1_{\mathcal{E}_j} \preceq \hat{\Sigma}_w^{(t)} = \frac{1}{n}\sum_{i=1}^n w_{ij}^{(t)}(x_i, y_i)x_i x_i^\top \preceq \frac{1}{n}\sum_{i=1}^n x_i x_i^\top.$$

By standard concentration results on $\hat{\Sigma} - I$, it holds that with $n \geqslant \mathcal{O}(d \log(1/\delta))$,

$$\lambda_{\max}(\hat{\Sigma}_w^{(t)}) \leqslant \lambda_{\max}(\hat{\Sigma}) \leqslant \frac{3}{2}$$

with probability at least $1 - \delta$. The concentration of $\frac{1}{n} \sum_{i=1}^{n} w_{ij}^{(t)}(x_i, y_i) x_i x_i^\top 1_{\mathcal{E}_j}$ comes from standard concentration argument for random matrix with sub-exponential norm Vershynin (2018). Since $w_{ij}^{(t)} \in (0, 1)$ and $x_i$ is standard multivariate Gaussian, then by Appendix B.2 in Kwon & Caramanis (2020), it holds that

$$\left\| \frac{1}{n} \sum_i w_{ij}^{(t)} x_i x_i^\top 1_{\mathcal{E}_j} - \mathbb{E}\big[ w_j^{(t)}(X, Y) XX^\top 1_{\mathcal{E}_j} \big] \right\|_2 \leqslant O\left( \sqrt{\pi_j^*} \sqrt{\frac{d \log(K^2/\delta)}{n}} \right)$$

with probability at least $1 - \delta$. By Lemma A.3 in Kwon & Caramanis (2020), it holds that under the same SNR condition

$$\lambda_{\min}\big( \mathbb{E}\big[ w_j^{(t)}(X, Y) XX^\top \big] \big) \geqslant \frac{\pi_j^*}{2}.$$

Therefore, as long as $n \geqslant \mathcal{O}\big( d \log(K^2/\delta)/\pi_{\min} \big)$, it holds that

$$\lambda_{\min}\big( \hat{\Sigma}_w^{(t)} \big) \geqslant \lambda_{\min}\left( \frac{1}{n} \sum_i w_{ij}^{(t)} x_i x_i^\top 1_{\mathcal{E}_1} \right) \geqslant \frac{\pi_j^*}{4}.$$

Therefore, we have under the same SNR and initialization condition, as long as

$$n \geqslant \mathcal{O}\Big( \max \big\{ d \log^2(dK^2/\delta), \big( K^2/\delta \big)^{1/3}, d \log(K^2/\delta)/\pi_{\min} \big\} \Big),$$

it holds that for appropriately small $\alpha$,

$$\big\| \big( I - \alpha \hat{\Sigma}_w^{(t)} \big)^T \big\|_2 \leqslant \max\{ |1 - 3\alpha/2|, (1 - \pi_{\min}\alpha/4) \}^T := \gamma_T, \tag{36}$$

$$\|e_B\|_2 \leqslant \sqrt{\frac{K\pi_j^{*2}}{\pi_{\min}}} \sqrt{\frac{d}{n} \log^2 \big( nK^2/\delta \big)} \max_\ell \big\| \beta_\ell^{(t-1)} - \beta_\ell^* \big\|_2 + \sqrt{\frac{K\pi_j^{*2}}{\pi_{\min}}} \sqrt{\frac{d}{n} \log^2 \big( nK^2/\delta \big)}, \tag{37}$$

$$B \leqslant \frac{\pi_j^*}{2} \max_\ell \big\| \beta_\ell^{(t-1)} - \beta_\ell^* \big\|_2, \tag{38}$$

$$\big\| \big[ \hat{\Sigma}_w^{(t)} \big]^{-1} \big\|_2 \leqslant \frac{4}{\pi_{\min}}. \tag{39}$$

For appropriately small $\alpha$, we have $\gamma_T \in (0, 1)$ Therefore, combining equation 36, equation 37, equation 38 and equation 39 together, we have

$$\beta_j^{(t)} - \beta_j^{(*)} \leqslant \left[ \gamma_T + (1 - \gamma_T) \left( \sqrt{\frac{K\pi_j^{*2}}{\pi_{\min}}} \sqrt{\frac{d}{n} \log^2 \big( nK^2/\delta \big)} + \frac{\pi_j^*}{2} \right) \right] \max_\ell \big\| \beta_\ell^{(t-1)} - \beta_\ell^* \big\|_2$$
$$+ \sqrt{\frac{K\pi_j^{*2}}{\pi_{\min}}} \sqrt{\frac{d}{n} \log^2 \big( nK^2/\delta \big)}$$

with probability at least $1 - 5\delta$.

To derive the concentration results for $\big| \frac{1}{n} \sum_i w_{ij}^{(t)}(x_i, y_i) - \mathbb{E}\big[ w_j^{(t)}(X, Y) \big] \big|$, we define following events

$$\mathcal{E}_{\ell,1} = \big\{ |v| \leqslant \tau_\ell \big\},$$
$$\mathcal{E}_{\ell,2} = \Big\{ 4(|\langle X, \Delta_j \rangle| \vee |\langle X, \Delta_\ell \rangle|) \leqslant |\langle X, \beta_\ell^* - \beta_j^* \rangle| \Big\},$$
$$\mathcal{E}_{\ell,3} = \Big\{ |\langle X, \beta_\ell^* - \beta_j^* \rangle| \geqslant 4\sqrt{2}\tau_\ell \Big\},$$

$$\mathcal{E}_{\ell,\text{ good}} = \mathcal{E}_{\ell,1} \cap \mathcal{E}_{\ell,2} \cap \mathcal{E}_{\ell,3},$$

where $\Delta_\ell = \beta_\ell^{(t-1)} - \beta_\ell^*$, then we have the decomposition

$$w_{ij}^{(t)}(x_i, y_i) = \left( \sum_{\ell \neq j}^{K} w_{ij}^{(t)}(x_i, y_i) 1_{\mathcal{E}_\ell \cap \mathcal{E}_{\ell,good}} + w_{ij}^{(t)}(x_i, y_i) 1_{\mathcal{E}_\ell \cap \mathcal{E}_{\ell,good}^c} \right) + w_{ij}^{(t)}(x_i, y_i) 1_{\mathcal{E}_j}.$$

Therefore, we could bound

$$\left| \frac{1}{n} \sum_i w_{ij}^{(t)}(x_i, y_i) 1_{\mathcal{E}_\ell \cap \mathcal{E}_{\ell,good}} - \mathbb{E}\left[ w_{ij}^{(t)}(x_i, y_i) 1_{\mathcal{E}_\ell \cap \mathcal{E}_{\ell,\text{ good}}} \right] \right|,$$

$$\left| \frac{1}{n} \sum_i w_{ij}^{(t)}(x_i, y_i) 1_{\mathcal{E}_\ell \cap \mathcal{E}_{\ell,good}^c} - \mathbb{E}\left[ w_{ij}^{(t)}(x_i, y_i) 1_{\mathcal{E}_\ell \cap \mathcal{E}_{\ell,good}^c} \right] \right|,$$

$$\left| \frac{1}{n} \sum_i w_{ij}^{(t)}(x_i, y_i) 1_{\mathcal{E}_j} - \mathbb{E}\left[ w_{ij}^{(t)}(x_i, y_i) 1_{\mathcal{E}_j} \right] \right|,$$

respectively. For the first part, note that

$$\left\| w_{ij}^{(t)}(x_i, y_i) 1_{\mathcal{E}_\ell \cap \mathcal{E}_{\ell,good}^c} \right\|_{\psi_2} = \sup_{p \geq 1} p^{-1/2} \mathbb{E}\left[ |w_{ij}^{(t)}(x_i, y_i)|^p \mid \mathcal{E}_\ell \cap \mathcal{E}_{\ell,good} \right]^{1/p}$$
$$\leq C \rho_{\ell j} \exp\left( -\tau_\ell^2 \right).$$

Therefore, with probability at least $1 - \delta/K^2$,

$$\left| \frac{1}{n} \sum_i w_{ij}^{(t)}(x_i, y_i) 1_{\mathcal{E}_\ell \cap \mathcal{E}_{\ell,good}} - \mathbb{E}\left[ w_{ij}^{(t)}(x_i, y_i) 1_{\mathcal{E}_\ell \cap \mathcal{E}_{\ell,good}} \right] \right| \leq \mathcal{O}\left( \rho_{\ell j} \exp\left( -\tau_\ell^2 \right) \sqrt{\pi_\ell^*} \sqrt{\frac{1}{n} \log(K^2/\delta)} \right).$$

For the second part, note that

$$\left\| w_{ij}^{(t)}(x_i, y_i) 1_{\mathcal{E}_\ell \cap \mathcal{E}_{\ell,good}^c} \right\|_{\psi_2} = \sup_{p \geq 1} p^{-1/2} \mathbb{E}_{\mathcal{D}}\left[ |w_{ij}^{(t)}(x_i, y_i)|^p \mid \mathcal{E}_\ell \cap \mathcal{E}_{\ell,\text{ good}}^c \right]^{1/p} \leq 1,$$
$$\mathbb{P}\left( \mathcal{E}_\ell \cap \mathcal{E}_{\ell,\text{ good}}^c \right) \leq \mathcal{O}\left( \pi_\ell^*/(K\rho_\pi) \right).$$

Therefore,

$$\left| \frac{1}{n} \sum_i w_{ij}^{(t)}(x_i, y_i) 1_{\mathcal{E}_\ell \cap \mathcal{E}_{\ell,good}^c} - \mathbb{E}\left[ w_{ij}^{(t)}(x_i, y_i) 1_{\mathcal{E}_\ell \cap \mathcal{E}_{\ell,good}^c} \right] \right| \leq \mathcal{O}\left( \sqrt{\frac{\pi_\ell^*}{K\rho_\pi} \vee \frac{\log(K^2/\delta)}{n}} \sqrt{\frac{\log(K^2/\delta)}{n}} \right).$$

Similar to the second part, we have the following concentration result for the last part:

$$\left| \frac{1}{n} \sum_i w_{ij}^{(t)}(x_i, y_i) 1_{\mathcal{E}_j} - \mathbb{E}\left[ w_{ij}^{(t)}(x_i, y_i) 1_{\mathcal{E}_j} \right] \right| \leq \mathcal{O}\left( \sqrt{\pi_j^* \vee \frac{\log(K^2/\delta)}{n}} \sqrt{\frac{\log(K^2/\delta)}{n}} \right).$$

Combining three parts together, we have

$$\left| \frac{1}{n} \sum_i w_{ij}^{(t)}(x_i, y_i) - \mathbb{E}\left[ w_j^{(t)}(X, Y) \right] \right| \leq \mathcal{O}\left( \sqrt{\frac{1}{n} \log(K^2/\delta)} \left( \sum_{\ell \neq j}^{K} \rho_{\ell j} \exp\left( -\tau_\ell^2 \right) \sqrt{\pi_\ell^*} + \sqrt{\frac{\pi_j^*}{K}} \right) + \sqrt{\frac{\pi_j^* \log(K^2/\delta)}{n}} \right)$$

$$\leq \mathcal{O}\left( \sqrt{\frac{K \log(K^2/\delta)}{n\pi_{\min}}} \sqrt{\frac{\pi_j^*}{K}} \left( \sum_{\ell \neq j}^{K} \frac{\sqrt{\rho_{\ell,j}} \sqrt{\pi_j^*}}{K\rho_\pi} + \sqrt{\frac{\pi_j^*}{k}} \right) + \sqrt{\frac{K \log(K^2/\delta)}{n\pi_{\min}}} \pi_j^* \right)$$

$$\leq \mathcal{O}\left( \sqrt{\frac{K \log(K^2/\delta)}{n\pi_{\min}}} \pi_j^* \right).$$

with probability at least $1 - 3\delta$. Therefore,

$$
\begin{aligned}
|x_{n+1}^\top \hat{\beta}^{\mathsf{OR}} - x_{n+1}^\top \beta^{\mathsf{OR}}| &\leqslant \|x_{n+1}\|_2 \|\hat{\beta}^{\mathsf{OR}} - \beta^{\mathsf{OR}}\|_2 \\
&\leqslant \|x_{n+1}\|_2 \Big( \max_j |\hat{\pi}_j - \pi_j^*| \max_j \|\beta_j^*\|_2 + \max_j\{\hat{\pi}_j\} \max_j \|\hat{\beta}_j - \beta_j^*\|_2 \Big) \\
&\leqslant \sqrt{\log(d/\delta)} \left( \sqrt{\frac{K \log(K^2/\delta)}{n\pi_{\min}}} \pi_j^* + \sqrt{\frac{K\pi_j^{*2}}{\pi_{\min}}} \sqrt{\frac{d}{n}} \log^2\big(nK^2/\delta\big) \right)
\end{aligned}
$$

with probability at least $1 - 9\delta$.

## B  Proof of Theorems in Section 4

### B.1  Proof of Theorem 4.1 in Section 4.1

**Proposition B.1** (Proposition A.4 Bai et al. (2024))**.** *Suppose that $\{X_\theta\}_{\theta\in\Theta}$ is a zero-mean random process given by*

$$
X_\theta := \frac{1}{N} \sum_{i=1}^N f(z_i; \theta) - \mathbb{E}_z[f(z;\theta)],
$$

*where $z_1, \cdots, z_N$ are i.i.d samples from a distribution $\mathbb{P}_z$ such that the following assumption holds:*

(a) *The index set $\Theta$ is equipped with a distance $\rho$ and diameter $D$. Further, assume that for some constant $A$, for any ball $\Theta'$ of radius $r$ in $\Theta$, the covering number admits upper bound $\log N\big(\delta; \Theta', \rho\big) \leqslant d\log(2Ar/\delta)$ for all $0 < \delta \leqslant 2r$.*

(b) *For any fixed $\theta \in \Theta$ and $z$ sampled from $\mathbb{P}_z$, the random variable $f(z;\theta)$ is a $\mathrm{SG}\big(B^0\big)$-sub-Gaussian random variable.*

(c) *For any $\theta, \theta' \in \Theta$ and $z$ sampled from $\mathbb{P}_z$, the random variable $f(z;\theta) - f(z;\theta')$ is a $\mathrm{SG}\big(B^1\rho(\theta, \theta')\big)$-subGaussian random variable.*

*Then with probability at least $1 - \delta$, it holds that*

$$
\sup_{\theta\in\Theta} |X_\theta| \leqslant CB^0 \sqrt{\frac{d\log(2A\kappa) + \log(1/\delta)}{N}},
$$

*where $C$ is a universal constant, and we denote $\kappa = 1 + B^1 D/B^0$.*

*Furthermore, if we replace the $\mathrm{SG}$ in assumption (b) and (c) by $\mathrm{SE}$, then with probability at least $1 - \delta$, it holds that*

$$
\sup_{\theta\in\Theta} |X_\theta| \leqslant CB^0 \left[ \sqrt{\frac{d\log(2A\kappa) + \log(1/\delta)}{N}} + \frac{d\log(2A\kappa) + \log(1/\delta)}{N} \right].
$$

For any $p \in [1, \infty]$, let $\|H\|_{2,p} := \left( \sum_{i=1}^n \|h_i\|_2^p \right)^{1/p}$ denote the column-wise $(2, p)$-norm of $H$. For any radius $R > 0$, we denote $\mathcal{H}_R := \Big\{ H : \|H\|_{2,\infty} \leqslant R \Big\}$ be the ball of radius $R$ under norm $\|\cdot\|_{2,\infty}$.

**Lemma B.1** (Corollary J.1 Bai et al. (2024))**.** *For a single attention layer $\boldsymbol{\theta}_{\mathrm{attn}} = \big\{(V_m, Q_m, K_m)\big\}_{m\in[M]} \subset \mathbb{R}^{D\times D}$ and any fixed dimension $D$, we consider*

$$
\Theta_{\mathrm{attn}, B'} := \big\{ \boldsymbol{\theta}_{\mathrm{attn}} : \|\boldsymbol{\theta}_{\mathrm{attn}}\| \leqslant B' \big\}.
$$

Then for $H \in \mathcal{H}_R, \boldsymbol{\theta}_{\mathrm{attn}} \in \Theta_{\mathrm{attn},B}$, the function $(\boldsymbol{\theta}_{\mathrm{attn}}, H) \mapsto \mathrm{Attn}_{\boldsymbol{\theta}_{\mathrm{attn}}}(H)$ is $(B^2 R^3)$-Lipschitz w.r.t. $\boldsymbol{\theta}_{\mathrm{attn}}$ and $(1 + B^3 R^2)$-Lipschitz w.r.t. $H$. Furthermore, for the function $\mathrm{TF}^R$ given by

$$\mathrm{TF}^R : (\boldsymbol{\theta}, H) \mapsto \mathrm{clip}_R\left(\mathrm{Attn}_{\boldsymbol{\theta}_{\mathrm{attn}}}(H)\right).$$

$\mathrm{TF}^R$ is $B_\Theta$-Lipschitz w.r.t $\boldsymbol{\theta}$ and $L_H$-Lipschitz w.r.t. $H$, where $B_\Theta := B^2 R^3$ and $B_H := 1 + B^3 R^2$.

**Proposition B.2** (Proposition J.1 Bai et al. (2024)). *For a fixed number of heads $M$ and hidden dimension $D$, we consider*

$$\Theta_{\mathrm{TF},L,B'} = \left\{\boldsymbol{\theta} = \boldsymbol{\theta}_{\mathrm{attn}}^{(1:L)} : M^{(\ell)} = M, D^{(\ell)} = D, \|\boldsymbol{\theta}\| \leqslant B'\right\}.$$

*Then the function $\mathrm{TF}^R$ is $\left(L B_H^{L-1} B_\Theta\right)$-Lipschitz w.r.t $\boldsymbol{\theta} \in \Theta_{\mathrm{TF},L,B}$ for any fixed $\mathbf{H}$.*

*Proof of Theorem 4.1.* Define events

$$\mathcal{E}_y := \left\{\max_{i \in [n+1], j \in [B]}\left\{|y_i^{(j)}|\right\} \leqslant B_y\right\},$$

$$\mathcal{E}_x := \left\{\max_{i \in [n+1], j \in [B]}\left\{\|x_i^{(j)}\|_2\right\} \leqslant B_x\right\},$$

and the random process

$$X_{\boldsymbol{\theta}} := \frac{1}{B}\sum_{j=1}^B \ell_{\mathrm{icl}}\left(\boldsymbol{\theta}; \mathbf{Z}^{(j)}\right) - \mathbb{E}_{\mathbf{Z}}\left[\ell_{\mathrm{icl}}(\boldsymbol{\theta}; \mathbf{Z})\right]$$

where $\mathbf{Z}^{(1:B)}$ are i.i.d. copies of $\mathbf{Z} \sim \mathrm{P}$, drawn from the distribution $\mathrm{P}$. The next step involves applying Proposition B.1 to the process $\{X_{\boldsymbol{\theta}}\}$ conditioning on events $\mathcal{E}_x \cap \mathcal{E}_y$. To proceed, we must verify the following preconditions:

(a) By [Wainwright (2019), Example 5.8], it holds that $\log N\left(\delta; \mathrm{B}_{\|\cdot\|}(r), \|\cdot\|\right) \leqslant L(3MD^2)\log(1 + 2r/\delta)$, where $\mathrm{B}_{\|\cdot\|}(r)$ is any ball of radius $r$ under norm $\|\cdot\|$.

(b) $\left|\ell_{\mathrm{icl}}(\boldsymbol{\theta}; \mathbf{Z})\right| \leqslant 4B_y^2$ and hence $4B_y^2$-sub-Gaussian.

(c) $\left|\ell_{\mathrm{icl}}(\boldsymbol{\theta}; \mathbf{Z}) - \ell_{\mathrm{icl}}(\widetilde{\boldsymbol{\theta}}; \mathbf{Z})\right| \leqslant 2B_y\left(L B_H^{L-1} B_\Theta\right)\|\boldsymbol{\theta} - \widetilde{\boldsymbol{\theta}}\|$ by Proposition B.2, where $B_\Theta := B'^2 R^3$ and $B_H := 1 + B'^3 R^2$.

Therefore, by Proposition B.1, conditioning on $\mathcal{E}_x \cap \mathcal{E}_y$ with probability at least $1 - \xi$,

$$\sup_{\boldsymbol{\theta}}|X_{\boldsymbol{\theta}}| \leqslant \mathcal{O}\left(B_y^2\sqrt{\frac{L(MD^2)\iota + \log(1/\xi)}{B}}\right)$$

where $\iota = 20L\log\left(2 + \max\left\{B', R, (2B_y)^{-1}\right\}\right)$. Note that

$$\mathrm{Var}(x_i^\top \beta_i) = \mathbb{E}_{\beta_i}\left[\mathrm{Var}(x_i^\top \beta_i \mid \beta_i)\right] + \mathrm{Var}\left(\mathbb{E}_{x_i}\left[x_i^\top \beta_i \mid \beta_i\right]\right) = \sum_{k=1}^K \pi_k^* \|\beta_k^*\|_2^2,$$

$$\mathbb{E}\left[e^{\lambda(x_i^\top \beta_i)}\right] = \sum_{k=1}^K \pi_k^* \mathbb{E}\left[e^{\lambda x_i^\top \beta_k^*}\right] \leqslant \sum_{k=1}^K \pi_k^* \exp\left\{\frac{1}{2}\lambda^2 \|\beta_k^*\|_2^2\right\}.$$

Denote $\tau_i$ to be the sub-Gaussian parameter of $x_i^\top \beta_i$. Then we have $\sqrt{\sum_{k=1}^K \pi_k^* \pi_k^* \|\beta_k^*\|_2^2} \leqslant \tau_i \leqslant \max_{i \leqslant K} \|\beta_i^*\|_2$. Besides, $y_i$ is sub-Gaussian with parameter at most $\sqrt{\tau_i^2 + \vartheta^2}$. Finally, note that $\|\beta_i^* - \beta_j^*\|_2^2 \leqslant 2(\|\beta_i^*\|_2^2 + \|\beta_j^*\|_2^2)$. Summing over $i$ and $j$ for all $i \neq j$, we have

$$\sum_{i \neq j} \|\beta_i^* - \beta_j^*\|_2^2 \leqslant 4(K-1) \sum_{i=1}^K \|\beta_i^*\|_2^2,$$

which implies $K(K-1) \min_{i \neq j} \|\beta_i^* - \beta_j^*\|_2^2 \leqslant 4(K-1) \sum_{i=1}^K \|\beta_i^*\|_2^2$. Thus, we can derive a lower bound of the subGaussian parameter $\tau_i$ as below

$$\tau_i^2 = \sum_{i=1}^K \pi_i^* \|\beta_i^*\|_2^2 \geqslant \pi_{\min} \sum_{i=1}^K \|\beta_i^*\|_2^2$$

$$\geqslant \frac{\pi_{\min} K}{4} \min_{i \neq j} \|\beta_i^* - \beta_j^*\|_2^2 = \frac{\pi_{\min} K}{4} R_{\min}.$$

Therefore, $\sqrt{\tau_i^2 + \vartheta^2} = \sqrt{(1 + (\vartheta/\tau)^2)} \tau_i \leqslant \sqrt{1 + \frac{4\vartheta^2}{\pi_{\min} K R_{\min}^2}} C = \sqrt{1 + 4/(\pi_{\min} K \eta^2)} C$. Then by taking

$$B_x = \sqrt{d \log(nB/\xi)},$$
$$B_y = \sqrt{2(1 + 4/(\pi_{\min} K \eta^2)) C^2 \log(2nB/\xi)},$$
$$R = 2 \max\{B_x, B_y\},$$

we have $\mathbb{P}(\mathcal{E}_y) \geqslant 1 - \xi$ and $\mathbb{P}(\mathcal{E}_x) \geqslant 1 - \xi$ by union bound. Hence, with probability at least $1 - 3\xi$,

$$\sup_{\boldsymbol{\theta}} |X_{\boldsymbol{\theta}}| \leqslant \mathcal{O}\left( (1 + 4/(\pi_{\min} K \eta^2)) \log(2nL/\xi) \sqrt{\frac{L(MD^2)\iota + \log(1/\xi)}{B}} \right)$$

where

$$\iota = 20L \log\left( 2 + \max\left\{ B', R, (2B_y)^{-1} \right\} \right)$$

is a log factor. $\qquad\square$

## B.2 Proof of Theorem 4.2

### B.2.1 Computation of the risk

Given the prompt in the form of

$$E = \begin{bmatrix} x_1 & x_2 & \dots & x_n & x_{n+1} \\ y_1 & y_2 & \dots & y_\ell & 0 \end{bmatrix} \in \mathbb{R}^{(d+1) \times (n+1)} \tag{40}$$

appropriately sized key, query, and value matrices $K, Q, V$, the output of a linear-attention block is given by

$$A := E + \frac{1}{n} V E (KE)^\top (QE).$$

We start by rewriting the output of the linear attention module in an alternative form. Following Zhang et al. (2023), we define

$$W_{PV} = \begin{pmatrix} * & * \\ u_{21}^\top & u_{-1} \end{pmatrix}, \quad W_{KQ} = \begin{pmatrix} U_{11} & * \\ u_{12}^\top & * \end{pmatrix}, \tag{41}$$

therefore, the prediction on the query sample is given by

$$\hat{y}_{n+1} = \left[ u_{21}^\top \cdot \frac{1}{n} X^\top X \cdot U_{11} + u_{21}^\top \cdot \frac{1}{n} X^\top \mathbf{y} \cdot u_{12}^\top + u_{-1} \cdot \frac{1}{n} \mathbf{y}^\top X \cdot U_{11} + u_{-1} \cdot \frac{1}{n} \mathbf{y}^\top \mathbf{y} \cdot u_{12}^\top \right] \cdot x_{n+1}$$

$$= \left[ (u_{21} + u_{-1}\beta_{n+1})^\top \cdot \frac{1}{n} X^\top X \cdot \left( U_{11} + \beta_{n+1} u_{12}^\top \right) \right] \cdot x_{n+1}$$

$$+ \left[ u_{21}^\top \cdot \frac{1}{n} X^\top v \cdot u_{12}^\top + u_{-1} \cdot \frac{1}{n} v^\top X \cdot U_{11} + u_{-1} \cdot \frac{2}{n} v^\top X \beta_{n+1} u_{12}^\top + \frac{1}{n} v^\top v \cdot u_{-1} u_{12}^\top \right] \cdot x_{n+1}.$$

### B.2.2 Gradient of Loss w.r.t. Parameters

In this section, we compute the gradient of the population loss w.r.t. parameters $\{u_{21}, u_{12}, U_{11}, u_{-1}\}$. To simplify the presentation, we define $v = [v_1, \ldots, v_n]^\top \in \mathbb{R}^n$, $X = [x_1^\top, \ldots, x_n^\top]^\top \in \mathbb{R}^{n \times d}$ and following notation

$$z_1^\top = (u_{21} + u_{-1}\beta_{n+1})^\top \cdot \frac{1}{n} X^\top X \cdot \left( U_{11} + \beta_{n+1} u_{12}^\top \right),$$

$$z_2^\top = u_{21}^\top \cdot \frac{1}{n} X^\top v \cdot u_{12}^\top + u_{-1} \cdot \frac{1}{n} v^\top X \cdot U_{11} + u_{-1} \cdot \frac{2}{n} v^\top X \beta_{n+1} u_{12}^\top,$$

$$z_3^\top = \frac{1}{n} v^\top v \cdot u_{-1} u_{12}^\top.$$

Since $X, v, \beta_{n+1}$ are independent, we have

$$2L_{\mathrm{SA}}(E, \theta) = \mathbb{E}(\hat{y}_{n+1} - \langle \beta_{n+1}, X \rangle - v_{n+1})^2$$

$$= \mathbb{E}\left[ \langle z_1 + z_2 + z_3 - \beta_{n+1}, X \rangle^2 \right] + \vartheta^2$$

$$= \left\langle I_d, \mathbb{E}\left(z_1 + z_2 + z_3 - \beta_{n+1}\right)\left(z_1 + z_2 + z_3 - \beta_{n+1}\right)^\top \right\rangle + \vartheta^2.$$

$$= \vartheta^2 + \underbrace{\left\langle I_d, \mathbb{E}\left(z_1 - \beta_{n+1}\right)\left(z_1 - \beta_{n+1}\right)^\top \right\rangle}_{S_1} + \underbrace{\left\langle I_d, \mathbb{E} z_2 z_2^\top \right\rangle}_{S_2} + \underbrace{\left\langle I_d, \mathbb{E} z_3 z_3^\top \right\rangle}_{S_3} + \underbrace{2\left\langle I_d, \mathbb{E}\left(z_1 - \beta_{n+1}\right) z_3^\top \right\rangle}_{S_4}.$$

We first compute $S_1$. Note that $X$ is independent of $v$ and $\beta_{n+1}$, then by making use of Lemma C.1

$$\left\langle I_d, \mathbb{E} z_1 z_1^\top \right\rangle$$

$$= \mathbb{E}\,\mathrm{tr}\left[ \left( U_{11} + \beta_{n+1} u_{12}^\top \right)^\top \cdot \frac{1}{n} X^\top X \cdot (u_{21} + u_{-1}\beta_{n+1})(u_{21} + u_{-1}\beta_{n+1})^\top \cdot \frac{1}{n} X^\top X \cdot \left( U_{11} + \beta_{n+1} u_{12}^\top \right) I_d \right]$$

$$= \frac{n+1}{n} \mathbb{E}_{\beta_{n+1}}\,\mathrm{tr}\left[ \left( U_{11} + \beta_{n+1} u_{12}^\top \right)^\top (u_{21} + u_{-1}\beta_{n+1})(u_{21} + u_{-1}\beta_{n+1})^\top \cdot I_d \cdot \left( U_{11} + \beta_{n+1} u_{12}^\top \right) I_d \right]$$

$$\quad + \frac{1}{n} \mathbb{E}_{\beta_{n+1}}\,\mathrm{tr}\left[ \mathrm{tr}\left( (u_{21} + u_{-1}\beta_{n+1})(u_{21} + u_{-1}\beta_{n+1})^\top I_d \right) \left( U_{11} + \beta_{n+1} u_{12}^\top \right)^\top I_d \left( U_{11} + \beta_{n+1} u_{12}^\top \right) I_d \right]$$

$$= \frac{n+1}{n} \mathbb{E}_{\beta_{n+1}}\,\mathrm{tr}\left[ \left( U_{11} + \beta_{n+1} u_{12}^\top \right)^\top (u_{21} + u_{-1}\beta_{n+1})(u_{21} + u_{-1}\beta_{n+1})^\top \left( U_{11} + \beta_{n+1} u_{12}^\top \right) \right]$$

$$\quad + \frac{1}{n} \mathbb{E}_{\beta_{n+1}}\,\mathrm{tr}\left[ (u_{21} + u_{-1}\beta_{n+1})(u_{21} + u_{-1}\beta_{n+1})^\top I_d \right] \mathrm{tr}\left[ \left( U_{11} + \beta_{n+1} u_{12}^\top \right)^\top \left( U_{11} + \beta_{n+1} u_{12}^\top \right) I_d \right]$$

$$= \frac{n+1}{n}(\mathrm{I}) + \frac{1}{n}(\mathrm{II}),$$

where

$$(\mathrm{I}) = \mathbb{E}_{\beta_{n+1}}\,\mathrm{tr}\left[ U_{11}^\top u_{21} u_{21}^\top U_{11} \right] + 2\mathbb{E}_{\beta_{n+1}}\,\mathrm{tr}\left[ \left( \beta_{n+1} u_{12}^\top \right)^\top u_{21} u_{21}^\top U_{11} \right] + 2\mathbb{E}_{\beta_{n+1}}\,\mathrm{tr}\left[ U_{11}^\top (u_{-1}\beta_{n+1}) u_{21}^\top U_{11} \right]$$

$$\quad + 2\mathbb{E}_{\beta_{n+1}}\,\mathrm{tr}\left[ \left( \beta_{n+1} u_{12}^\top \right)^\top (u_{-1}\beta_{n+1}) u_{21}^\top U_{11} \right] + 2\mathbb{E}_{\beta_{n+1}}\,\mathrm{tr}\left[ \left( \beta_{n+1} u_{12}^\top \right)^\top u_{21} (u_{-1}\beta_{n+1}^\top) U_{11} \right]$$

$$\quad + \mathbb{E}_{\beta_{n+1}}\,\mathrm{tr}\left[ \left( \beta_{n+1} u_{12}^\top \right)^\top u_{21} u_{21}^\top \left( \beta_{n+1} u_{12}^\top \right) \right] + \mathbb{E}_{\beta_{n+1}}\,\mathrm{tr}\left[ U_{11}^\top (u_{-1}\beta_{n+1})(u_{-1}\beta_{n+1})^\top U_{11} \right]$$

$$\quad + 2\mathbb{E}_{\beta_{n+1}}\,\mathrm{tr}\left[ U_{11}^\top (u_{-1}\beta_{n+1})(u_{-1}\beta_{n+1})^\top \left( \beta_{n+1} u_{12}^\top \right) \right] + 2\mathbb{E}_{\beta_{n+1}}\,\mathrm{tr}\left[ \left( \beta_{n+1} u_{12}^\top \right)^\top u_{21} (u_{-1}\beta_{n+1})^\top \left( \beta_{n+1} u_{12}^\top \right) \right]$$

$$\quad + \mathbb{E}_{\beta_{n+1}}\,\mathrm{tr}\left[ \left( \beta_{n+1} u_{12}^\top \right)^\top (u_{-1}\beta_{n+1})(u_{-1}\beta_{n+1})^\top \left( \beta_{n+1} u_{12}^\top \right) \right]$$

$$= \mathrm{tr}\left[ U_{11}^\top u_{21} u_{21}^\top U_{11} \right] + 2\mathbb{E}\,\mathrm{tr}\left( u_{12}\beta_{n+1}^\top u_{21} u_{21}^\top U_{11} \right) + 2u_{-1}\mathbb{E}\,\mathrm{tr}\left( U_{11}^\top \beta_{n+1} u_{21}^\top U_{11} \right)$$

$$\quad + 2u_{-1}\mathbb{E}_{\beta_{n+1}}\left[ \beta_{n+1}^\top \beta_{n+1} \right] u_{21}^\top U_{11} u_{12} + 2u_{-1}\mathbb{E}_{\beta_{n+1}}\,\mathrm{tr}\left[ u_{12}\beta_{n+1}^\top u_{21} \beta_{n+1}^\top U_{11} \right]$$

$$\quad + u_{12}^\top u_{12}\mathbb{E}_{\beta_{n+1}}\,\mathrm{tr}\left[ u_{21} u_{21}^\top \beta_{n+1}\beta_{n+1}^\top \right] + u_{-1}^2\mathbb{E}_{\beta_{n+1}}\,\mathrm{tr}\left[ \beta_{n+1}\beta_{n+1}^\top U_{11} U_{11}^\top \right]$$

$$\quad + 2u_{-1}^2\mathbb{E}_{\beta_{n+1}}\,\mathrm{tr}\left[ U_{11}^\top \beta_{n+1}\beta_{n+1}^\top \beta_{n+1} u_{12}^\top \right] + 2u_{-1}u_{12}^\top u_{12}\mathbb{E}_{\beta_{n+1}}\,\mathrm{tr}\left[ \beta_{n+1}^\top u_{21}\beta_{n+1}^\top \beta_{n+1} \right]$$

$$+ u_{-1}^2 u_{12}^\top u_{12} \mathbb{E}_{\beta_{n+1}} \|\beta_{n+1}\|_2^4$$

and

$$
\begin{aligned}
\text{(II)} &= \mathbb{E}_{\beta_{n+1}} (u_{21} + u_{-1}\beta_{n+1})^\top (u_{21} + u_{-1}\beta_{n+1}) \operatorname{tr}\left[\left(U_{11} + \beta_{n+1}u_{12}^\top\right)^\top \left(U_{11} + \beta_{n+1}u_{12}^\top\right)\right] \\
&= \mathbb{E}_{\beta_{n+1}}\left[u_{21}^\top u_{21} + 2u_{-1}\beta_{n+1}^\top u_{21} + u_{-1}^2\beta_{n+1}^\top\beta_{n+1}\right]\operatorname{tr}\left[U_{11}^\top U_{11} + 2u_{12}\beta_{n+1}^\top U_{11} + u_{12}\beta_{n+1}^\top\beta_{n+1}u_{12}^\top\right] \\
&= u_{21}^\top u_{21}\operatorname{tr}\left[U_{11}^\top U_{11}\right] + 2u_{21}^\top u_{21}\mathbb{E}\beta_{n+1}^\top U_{11}u_{12} + 2u_{-1}\mathbb{E}\beta_{n+1}^\top u_{21}\operatorname{tr}\left[U_{11}^\top U_{11}\right] \\
&\quad + u_{21}^\top u_{21}u_{12}^\top u_{12}\mathbb{E}\beta_{n+1}^\top\beta_{n+1} + 4u_{-1}\mathbb{E}\beta_{n+1}^\top u_{21}\beta_{n+1}^\top U_{11}u_{12} + u_{-1}^2\operatorname{tr}U_{11}U_{11}^\top\mathbb{E}\beta_{n+1}^\top\beta_{n+1} \\
&\quad + 2u_{-1}u_{12}^\top u_{12}\mathbb{E}\beta_{n+1}^\top u_{21}\beta_{n+1}^\top\beta_{n+1} + 2u_{-1}^2\mathbb{E}\beta_{n+1}^\top\beta_{n+1}\beta_{n+1}^\top U_{11}u_{12} \\
&\quad + u_{-1}^2 u_{12}^\top u_{12}\mathbb{E}\|\beta_{n+1}\|_2^4.
\end{aligned}
$$

For the cross term in $S_1$, we have

$$
\begin{aligned}
C(U_{11}, u_{12}, u_{21}, u_{-1}) &= \left\langle I_d, \mathbb{E}\boldsymbol{z}_1\beta_{n+1}^\top\right\rangle = \mathbb{E}\left\{(u_{21} + u_{-1}\beta_{n+1})^\top \cdot \frac{1}{n}X^\top X \left(U_{11} + \beta_{n+1}u_{12}^\top\right)\beta_{n+1}\right\} \\
&= \mathbb{E}\left\{(u_{21} + u_{-1}\beta_{n+1})^\top \left(U_{11} + \beta_{n+1}u_{12}^\top\right)\beta_{n+1}\right\} \\
&= \mathbb{E}_{\beta_{n+1}}\left[u_{21}^\top U_{11}\beta_{n+1}\right] + \mathbb{E}_{\beta_{n+1}}\left[u_{21}^\top\beta_{n+1}u_{12}^\top\beta_{n+1}\right] \\
&\quad + u_{-1}\mathbb{E}_{\beta_{n+1}}\left[\beta_{n+1}^\top U_{11}\beta_{n+1}\right] + u_{-1}\mathbb{E}_{\beta_{n+1}}\left[\beta_{n+1}^\top\beta_{n+1}u_{12}^\top\beta_{n+1}\right].
\end{aligned}
$$

Finally,

$$\left\langle I_d, \mathbb{E}\beta_{n+1}\beta_{n+1}^\top\right\rangle = \mathbb{E}\beta_{n+1}^\top\beta_{n+1},$$

$$
\begin{aligned}
S_2 &= \frac{1}{n^2}\mathbb{E}\Bigg[u_{21}^\top X^\top v u_{12}^\top u_{12}v^\top X u_{21} + u_{-1}^2 v^\top X U_{11}U_{11}^\top X^\top v \\
&\qquad + 2u_{-1}\cdot v^\top X U_{11}u_{12}v^\top X u_{21} + 4u_{-1}^2 v^\top X\beta_{n+1}u_{12}^\top u_{12}\beta_{n+1}^\top X^\top v \\
&\qquad + 4u_{-1}v^\top X\beta_{n+1}u_{12}^\top u_{12}v^\top X u_{21} + 4u_{-1}v^\top X\beta_{n+1}u_{12}^\top U_{11}X^\top v\Bigg] \\
&= \frac{1}{n^2}\mathbb{E}\left[u_{21}^\top X^\top v u_{12}^\top u_{12}v^\top X_{u_{21}}\right] + \frac{\vartheta^2}{n^2}u_{-1}^2\mathbb{E}\operatorname{tr}\left(X U_{11}U_{11}^\top X^\top\right) \\
&\quad + \frac{2u_{-1}}{n^2}\mathbb{E}\left[v^\top X U_{11}u_{12}v^\top X u_{21}\right] + \frac{4u_{-1}^2\vartheta^2}{n^2}\mathbb{E}\operatorname{tr}\left(X\beta_{n+1}u_{12}^\top u_{12}\beta_{n+1}^\top X^\top\right) \\
&\quad + \frac{4u - 1}{n^2}\mathbb{E}\left[v^\top X\beta_{n+1}u_{12}^\top u_{12}v^\top X u_{21}\right] + \frac{4u_{-1}}{n^2}\mathbb{E}\left[v^\top X\beta_{n+1}u_{12}^\top U_{11}X^\top v\right] \\
&= \frac{\vartheta^2}{n}\left[u_{21}^\top u_{21}u_{12}^\top u_{12} + u_{-1}^2\operatorname{tr}\left(U_{11}U_{11}^\top\right) + 2u_{-1}u_{12}^\top U_{11}^\top u_{21} + 4u_{-1}^2 u_{12}^\top u_{12}\operatorname{tr}\left(\mathbb{E}\beta_{n+1}\beta_{n+1}^\top\right)\right] \\
&\quad + \frac{4u_{-1}}{n^2}\mathbb{E}\left[v^\top X\beta_{n+1}u_{12}^\top u_{12}v^\top X u_{21}\right] + \frac{4u_{-1}}{n^2}\mathbb{E}\left[v^\top X\beta_{n+1}u_{12}^\top U_{11}X^\top v\right],
\end{aligned}
$$

$$S_3 = \frac{\sigma^4(n+2)}{n}u_{-1}^2 u_{12}^\top u_{12},$$

and

$$
\begin{aligned}
S_4 &= 2\left\langle I_d, \mathbb{E}\left(z_1 - \beta_{n+1}\right)z_3^\top\right\rangle \\
&= 2\mathbb{E}\left\{\left[(u_{21} + u_{-1}\beta_{n+1})^\top\frac{1}{n}X^\top X\left(u_n + \beta_{n+1}u_{12}^\top\right) - \beta_{n+1}^\top\right]\frac{1}{n}v^\top v u_{-1}u_{12}\right\} \\
&= 2\vartheta^2 u_{-1}\mathbb{E}\left\{\left[(u_{21} + u_{-1}\beta_{n+1})^\top\left(U_{11} + \beta_{n+1}u_{12}^\top\right) - \beta_{n+1}^\top\right]u_{12}\right\}
\end{aligned}
$$

$$= 2\vartheta^2 u_{-1} \mathbb{E}\left\{\left[u_{21}^\top U_{11} + u_{21}^\top \beta_{n+1} u_{12}^\top + u_{-1}\beta_{n+1}^\top U_{11} + u_{-1}\beta_{n+1}^\top \beta_{n+1} u_{12}^\top - \beta_{n+1}^\top\right] u_{12}\right\}$$

$$= 2\vartheta^2 u_{-1} u_{21}^\top U_{11} u_{12} + 2\vartheta^2 u_{-1}\mathbb{E} u_{21}^\top \beta_{n+1} u_{12}^\top u_{12}$$

$$\quad + 2\vartheta^2 u_{-1}^2 \mathbb{E}\beta_{n+1}^\top U_{11} u_{12} + 2\vartheta^2 u_{-1}^2 u_{12}^\top u_{12}\mathbb{E}\beta_{n+1}^\top \beta_{n+1} - 2\vartheta^2 u_{-1}\mathbb{E}\beta_{n+1}^\top u_{12}.$$

Now we compute the derivative of the risk $\mathcal{R}$ w.r.t. parameter $U_{11}, u_{-1}, u_{12}$ and $u_{21}$, where

$$2L_{\mathrm{SA}}(E,\theta) - \vartheta^2 = \frac{n+1}{n}(\mathsf{I}) + \frac{1}{n}(\mathsf{II}) - 2C(U_{11}, u_{12}, u_{21}, u_{-1}) + \mathbb{E}\beta_{n+1}^\top \beta_{n+1} + S_2 + S_3 + S_4.$$

We first calculate the derivative of all components in risk w.r.t. the scalar $u_{-1}$.

$$\frac{\partial(\mathsf{I})}{\partial u_{-1}} = 2\mathbb{E}\,\mathrm{tr}\left(U_{11}^\top \beta_{n+1} u_{21}^\top U_{11}\right) + 2\mathbb{E}\left[\beta_{n+1}^\top \beta_{n+1}\right] u_{21}^\top U_{11} u_{12}$$

$$\quad + 2\mathbb{E}\,\mathrm{tr}\left[u_{12}\beta_{n+1}^\top u_{21}\beta_{n+1}^\top U_{11}\right] + 2u_{-1}u_{12}^\top u_{12}\mathbb{E}\|\beta_{n+1}\|_2^4,$$

$$\frac{\partial(\mathsf{II})}{\partial u_{-1}} = 2\mathbb{E}\beta_{n+1}^\top u_{21}\,\mathrm{tr}\left(U_{11}^\top U_{11}\right) + 4\mathbb{E}\beta_{n+1}^\top u_{21}\beta_{n+1}^\top U_{11} u_{12} + 2u_{-1}\,\mathrm{tr}\left[U_{11}U_{11}^\top\right]\mathbb{E}\beta_{n+1}^\top \beta_{n+1}$$

$$\quad + 2u_{12}^\top u_{12}\mathbb{E}\beta_{n+1}^\top u_{22}\beta_{n+1}^\top \beta_{n+1} + 4u_{-1}\mathbb{E}\beta_{n+1}^\top \beta_{n+1}\beta_{n+1}^\top U_{11}u_{12} + 2u_{-1}u_{12}^\top u_{12}\mathbb{E}\|\beta_{n+1}\|_2^2,$$

$$\frac{\partial C(U_{11}, u_{12}, u_{21}, u_{-1})}{\partial u_{-1}} = \mathbb{E}\left[\beta_{n+1}^\top U_{11}\beta_{n+1}\right] + \mathbb{E}\left[\beta_{n+1}^\top \beta_{n+1} u_{12}^\top \beta_{n+1}\right],$$

and

$$\frac{\partial S_2}{\partial u_{-1}} = \frac{4\vartheta^2}{n}u_{-1}\,\mathrm{tr}\left(U_{11}U_{11}^\top\right) + \frac{2\vartheta^2}{n}u_{12}^\top U_{11}^\top u_{21} + \frac{8\vartheta^2}{n}u_{-1}u_{12}^\top u_{12}\,\mathrm{tr}\left(\mathbb{E}\beta_{n+1}\beta_{n+1}^\top\right),$$

$$\quad + \frac{4}{n^2}\mathbb{E}\left[v^\top X\beta_{n+1}u_{12}^\top u_{12}v^\top X u_{21}\right] + \frac{4}{n^2}\mathbb{E}\left[v^\top X\beta_{n+1}u_{12}^\top U_{11}X^\top v\right],$$

$$\frac{\partial S_3}{\partial u_{-1}} = \frac{2\sigma^4(n+2)}{n}u_{-1}\left(u_{12}^\top u_{12}\right),$$

$$\frac{\partial S_4}{\partial u_{-1}} = 2\vartheta^2 u_{21}^\top U_{11}u_{12} + 2\vartheta^2\mathbb{E}u_{21}^\top \beta_{n+1}u_{12}^\top u_{12}$$

$$\quad + 4\vartheta^2 u_{-1}\mathbb{E}\beta_{n+1}^\top U_{11}u_{12} + 4\vartheta^2 u_{-1}u_{12}^\top u_{12}\mathbb{E}\beta_{n+1}^\top \beta_{n+1} - 2\vartheta^2\mathbb{E}\beta_{n+1}^\top u_{12}.$$

Next, we calculate the derivative of all components in risk w.r.t. the matrix parameter $U_{11}$:

$$\frac{\partial(\mathsf{I})}{\partial U_{11}} = 2u_{21}u_{21}^\top U_{11} + 2\mathbb{E}\left[u_{21}u_{21}^\top \beta_{n+1}u_{12}^\top\right] + 2u_{-1}\mathbb{E}\left[\beta_{n+1}u_{21}^\top U_{11} + u_{21}\beta_{n+1}^\top U_{11}\right]$$

$$\quad + 2u_{-1}u_{21}\mathbb{E}\beta_{n+1}\beta_{n+1}^\top u_{12}^\top + 2u_{-1}\mathbb{E}\,\mathrm{tr}\left[\beta_{n+1}u_{21}^\top \beta_{n+1}u_{12}^\top\right]$$

$$\quad + 2u_{-1}^2\mathbb{E}\beta_{n+1}\beta_{n+1}^\top U_{11} + 2u_{-1}^2\mathbb{E}\beta_{n+1}\beta_{n+1}^\top \beta_{n+1}u_{12}^\top,$$

$$\frac{\partial(\mathsf{II})}{\partial U_{11}} = 2u_{21}^\top u_{21}U_{11} + 2u_{21}^\top u_{21}\mathbb{E}\beta_{n+1}u_{12}^\top + 4u_{-1}\mathbb{E}\beta_{n+1}^\top u_{21}U_{11}$$

$$\quad + 4u_{-1}\mathbb{E}\beta_{n+1}u_{21}^\top \beta_{n+1}u_{12}^\top + 2u_{-1}^2 U_{11}\mathbb{E}\beta_{n+1}^\top \beta_{n+1}$$

$$\quad + 2u_{-1}^2\mathbb{E}\beta_{n+1}\beta_{n+1}^\top \beta_{n+1}u_{12}^\top,$$

$$\frac{\partial C(U_{11}, u_{12}, u_{21}, u_{-1})}{\partial U_{11}} = \mathbb{E}u_{21}\beta_{n+1}^\top + u_{-1}\mathbb{E}\beta_{n+1}\beta_{n+1}^\top,$$

and

$$\frac{\partial S_2}{\partial U_{11}} = \frac{2\vartheta^2}{n}u_{-1}^2 U_{11} + \frac{2\vartheta^2}{n}u_{-1}u_{21}u_{12}^\top + \frac{4u_{-1}}{n^2}\mathbb{E}\left[u_{12}\beta_{n+1}^\top X^\top vv^\top X\right],$$

$$\frac{\partial S_3}{\partial U_{11}} = 0,$$

$$\frac{\partial S_4}{\partial U_{11}} = 2\vartheta^2 u_{-1} u_{21} u_{12}^\top + 2\vartheta^2 u_{-1}^2 \mathbb{E}\beta_{n+1} u_{12}^\top.$$

Then we calculate the derivative of all components in risk w.r.t. the parameter $u_{12}$:

$$\frac{\partial (\mathsf{I})}{\partial u_{12}} = 2\mathbb{E}U_{11}^\top u_{21} u_{21}^\top \beta_{n+1} + 2u_{-1}\mathbb{E}\beta_{n+1}^\top \beta_{n+1} U_{11}^\top u_{21}$$

$$+ 2u_{-1}U_{11}^\top \beta_{n+1} u_{21}^\top \beta_{n+1} + 2u_{12}\mathbb{E}\operatorname{tr}\left[u_{21}u_{21}^\top \beta_{n+1}\beta_{n+1}^\top\right]$$

$$+ 2u_{-1}^2\mathbb{E}U_{11}^\top \beta_{n+1}\beta_{n+1}^\top \beta_{n+1} + 4u_{-1}u_{12}\mathbb{E}\operatorname{tr}\left[\beta_{n+1}^\top u_{21}\beta_{n+1}^\top \beta_{n+1}\right] + 2u_{-1}^2 u_{12}\mathbb{E}\left\|\beta_{n+1}\right\|_2^4,$$

$$\frac{\partial (\mathsf{II})}{\partial u_{12}} = 2U_{11}^\top \mathbb{E}\beta_{n+1} u_{21}^\top u_{21} + 2u_{21}^\top u_{21} u_{12}\mathbb{E}\beta_{n+1}^\top \beta_{n+1} + 4u_{-1}U_{11}^\top \mathbb{E}\beta_{n+1}u_{21}^\top \beta_{n+1}$$

$$+ 4u_{-1}u_{12}\mathbb{E}\beta_{n+1}^\top u_{21}\beta_{n+1}^\top \beta_{n+1} + 2u_{-1}^2\mathbb{E}U_{11}^\top \beta_{n+1}\beta_{n+1}^\top \beta_{n+1} + 2u_{-1}^2 u_{12}\mathbb{E}\left\|\beta_{n+1}\right\|_2^4,$$

$$\frac{\partial C(U_{11}, u_{12}, u_{21}, u_{-1})}{\partial u_{12}} = \mathbb{E}\left[\beta_{n+1}u_{21}^\top \beta_{n+1}\right] + u_{-1}\cdot \mathbb{E}\left[\beta_{n+1}\beta_{n+1}^\top \beta_{n+1}\right]$$

and

$$\frac{\partial S_2}{\partial u_{12}} = \frac{2\vartheta^2}{n}\left(u_{21}^\top u_{21}\right)u_{12} + \frac{2\vartheta^2}{n}u_{-1}U_{11}^\top u_{21} + \frac{8\vartheta^2}{n}u_{-1}^2\operatorname{tr}\left[\mathbb{E}\beta_{n+1}\beta_{n+1}^\top\right]u_{12}$$

$$+ \frac{4u_{-1}}{n^2}\mathbb{E}\left[u_{12}\beta_{n+1}^\top X^\top v u_{21}^\top X^\top v + u_{12}v^\top X u_{21}v^\top X\beta_{n+1}\right] + \frac{4u_{-1}}{n^2}\mathbb{E}U_{11}X^\top vv^\top X\beta_{n+1},$$

$$\frac{\partial S_3}{\partial u_{12}} = \frac{2(n+2)\sigma^4}{n}u_{-1}^2 u_{12},$$

$$\frac{\partial S_4}{\partial u_{12}} = 2\vartheta^2 u_{-1}U_{11}^\top u_{21} + 4\vartheta^2 u_{-1}u_{12}\mathbb{E}\beta_{n+1}^\top u_{21}.$$

Finally, we calculate the derivative of all components in risk w.r.t. the parameter $u_{21}$:

$$\frac{\partial (\mathsf{I})}{\partial u_{21}} = 2U_{11}U_{11}^\top u_{21} + 2\mathbb{E}\left[U_{11}u_{12}\beta_{n+1}^\top + \beta_{n+1}u_{12}^\top U_{11}^\top\right]u_{21} + 2u_{-1}\mathbb{E}U_{11}U_{11}^\top \beta_{n+1}$$

$$+ 2u_{-1}\mathbb{E}\beta_{n+1}u_{12}^\top U_{11}\beta_{n+1} + 2\left(u_{12}^\top u_{12}\right)\mathbb{E}\beta_{n+1}\beta_{n+1}^\top u_{21} + 2u_{-1}u_{12}^\top u_{12}\mathbb{E}\beta_{n+1}\beta_{n+1}^\top \beta_{n+1},$$

$$\frac{\partial (\mathsf{II})}{\partial u_{21}} = 2\operatorname{tr}\left[U_{11}U_{11}^\top\right]u_{21} + 4\left(\mathbb{E}\beta_{n+1}^\top U_{11}u_{12}\right)u_{21} + 2u_{-1}\operatorname{tr}\left[U_{11}U_{11}^\top\right]\mathbb{E}\beta_{n+1}$$

$$+ 2u_{12}^\top u_{12}\left(\mathbb{E}\beta_{n+1}^\top \beta_{n+1}\right)u_{21} + 4u_{-1}\mathbb{E}\beta_{n+1}u_{12}^\top U_{11}^\top \beta_{n+1} + 2u_{-1}u_{12}^\top u_{12}\mathbb{E}\beta_{n+1}\beta_{n+1}^\top \beta_{n+1},$$

$$\frac{\partial C(U_{11}, u_{12}, u_{21}, u_{-1})}{\partial u_{21}} = \mathbb{E}U_{11}\beta_{n+1} + \mathbb{E}\beta_{n+1}u_{12}^\top \beta_{n+1},$$

and

$$\frac{\partial S_2}{\partial u_{21}} = \frac{2\vartheta^2}{n}\left(u_{12}^\top u_{12}\right)u_{21} + \frac{2\vartheta^2}{n}u_{-1}U_{11}u_{12} + \frac{4u_{-1}}{n}\mathbb{E}\left[X^\top vu_{12}^\top u_{12}\beta_{n+1}^\top X^\top v\right],$$

$$\frac{\partial S_3}{\partial u_{21}} = 0,$$

$$\frac{\partial S_4}{\partial u_{21}} = 2\vartheta^2 u_{-1}U_{11}u_{12} + 2\vartheta^2 u_{-1}\mathbb{E}\beta_{n+1}u_{12}^\top u_{12}.$$

When $\mathbb{E}\beta_{n+1} = 0$, then by taking the derivative to zero and solving the equations, we have

$$u_{12}^* = u_{21}^* = 0$$

and

$$\frac{n+1}{n} \cdot 2u_{-1}^2 \mathbb{E}\beta_{n+1}\beta_{n+1}^\top U_{11} + \frac{1}{n} \cdot 2u_{-1}^2 \mathbb{E}\beta_{n+1}^\top \beta_{n+1} U_{11} - 2u_{-1}\mathbb{E}\beta_{n+1}\beta_{n+1}^\top + \frac{2\vartheta^2}{n}u_{-1}^2 U_{11} = 0,$$

which implies

$$u_{-1}^* U_{11}^* = \left[\frac{n+1}{n}\mathbb{E}\beta_{n+1}\beta_{n+1}^\top + \left(\frac{1}{n}\mathbb{E}\beta_{n+1}^\top\beta_{n+1} + \frac{\vartheta^2}{n}\right)I\right]^{-1}\mathbb{E}\beta_{n+1}\beta_{n+1}^\top.$$

### B.2.3 Dynamics of parameters

**Lemma B.2.** Let $E \in \mathbb{R}^{(d+1)\times(n+1)}$ be an embedding matrix corresponding to a prompt of length $n$ and define

$$U = \begin{pmatrix} U_{11} & u_{12} \\ (u_{21})^\top & u_{-1} \end{pmatrix} \in \mathbb{R}^{(d+1)\times(d+1)}, \quad u = \mathrm{Vec}(U) \in \mathbb{R}^{(d+1)^2}. \tag{42}$$

When $\mathbb{E}\beta_i = 0$, then starting with $u_{12} = u_{21} = 0_d$, the dynamics of parameter $U$ follows

$$\frac{dU_{11}}{dt} = -u_{-1}^2 U_{11}\left[\frac{n+1}{n}\mathbb{E}\beta\beta^\top + \left(\frac{1}{n}\mathbb{E}\beta^\top\beta + \frac{\vartheta^2}{n}\right)I\right] + u_{-1}\mathbb{E}\beta\beta^\top, \tag{43}$$

$$\frac{du_{-1}}{dt} = -\mathrm{tr}\left[u_{-1}U_{11}U_{11}^\top\left(\frac{n+1}{n}\mathbb{E}\beta\beta^\top + \left(\frac{1}{n}\mathbb{E}\beta^\top\beta + \frac{\vartheta^2}{n}\right)I\right) - U_{11}\mathbb{E}\beta\beta^\top\right] \tag{44}$$

and $u_{21}(t) = u_{12}(t) = 0$ for all $t \geq 0$.

*Proof.* Based on the calculation in Section B.2.2, we see that if we initialize $u_{12} = u_{21} = 0$ at $t = 0$ and use the identities

$$\frac{du_{12}}{dt} = -\nabla_{u_{12}}L_{\mathrm{SA}}(E;\theta),$$

$$\frac{du_{21}}{dt} = -\nabla_{u_{21}}L_{\mathrm{SA}}(E;\theta),$$

we can see that $\frac{du_{12}}{dt} = \frac{du_{21}}{dt} = 0$. The expression of $\frac{dU_{11}}{dt}$ and $\frac{du_{-1}}{dt}$ can be obtained by plugging $u_{12} = u_{21} = 0$ and $\mathbb{E}\beta = 0$ in the gradient $\nabla_{U_{11}}L_{\mathrm{SA}}(E,\theta)$ and $\frac{d}{du_{-1}}L_{\mathrm{SA}}(E,\theta)$. $\square$

**Remark B.1.** One can show that with $u_{12} = u_{21} = 0_d$,

$$\frac{dU_{11}}{dt}U_{11}^\top = \left[-u_{-1}^2 U_{11}\left[\frac{n+1}{n}\mathbb{E}\beta\beta^\top + \left(\frac{1}{n}\mathbb{E}\beta^\top\beta + \frac{\vartheta^2}{n}\right)I\right] + u_{-1}\mathbb{E}\beta\beta^\top\right]U_{11}^\top,$$

$$\frac{du_{-1}}{dt}u_{-1} = -\mathrm{tr}\left[u_{-1}^2 U_{11}U_{11}^\top\left(\frac{n+1}{n}\mathbb{E}\beta\beta^\top + \left(\frac{1}{n}\mathbb{E}\beta^\top\beta + \frac{\vartheta^2}{n}\right)I\right) - u_{-1}U_{11}\mathbb{E}\beta\beta^\top\right].$$

Hence,

$$\frac{d}{dt}\mathrm{tr}\left[U_{11}U_{11}^\top\right] = \frac{d}{dt}u_{-1}^2. \tag{45}$$

Besides, the loss function could be simplified as

$$\tilde{\ell}(U_{11}, u_{-1}) = \frac{1}{2}u_{-1}^2\,\mathrm{tr}\left[\left(\frac{n+1}{n}\mathbb{E}\beta\beta^\top + \left(\frac{1}{n}\mathbb{E}\beta^\top\beta + \frac{\vartheta^2}{n}\right)I\right)U_{11}U_{11}^\top\right] - u_{-1}\,\mathrm{tr}\left[U_{11}\mathbb{E}\beta\beta^\top\right] + \mathbb{E}\beta^\top\beta + \vartheta^2$$

and thus,

$$\frac{\mathrm{d}}{\mathrm{d}t}U_{11}(t) = -\frac{\partial\tilde{\ell}(U_{11}, u_{-1})}{\partial U_{11}}, \quad \frac{\mathrm{d}}{\mathrm{d}t}u_{-1}(t) = -\frac{\partial\tilde{\ell}(U_{11}, u_{-1})}{\partial u_{-1}}.$$

**Lemma B.3.** Under initialization condition equation 11, when the parameter $\gamma$ satisfies the condition

$$\gamma \leqslant \sqrt{\frac{2\lambda_{\min}\left(\mathbb{E}\beta\beta^\top\right)}{\sqrt{d}\left(\frac{n+d+1}{n}\lambda_{\max}\left(\mathbb{E}\beta\beta^\top\right) + \frac{\vartheta^2}{n}\right)}} \tag{46}$$

we have:

- For all $t \geqslant 0$

$$u_{-1} \geqslant \sqrt{\frac{\gamma^2\|\Theta^\top\mathbb{E}\beta\beta^\top\|_F^2}{2\sqrt{d}\|\mathbb{E}\beta\beta^\top\|_{op}}\left(2 - \sqrt{d}\gamma^2\left\|\frac{n+1}{n}\mathbb{E}\beta\beta^\top + \left(\frac{1}{n}\mathbb{E}\beta^\top\beta + \frac{\vartheta^2}{n}\right)I\right\|_{op}\|\Theta\Theta^\top\|_F\right)} > 0. \tag{47}$$

- $\tilde{\ell}(U_{11}(t), u_{-1}(t))$ satisfies the PL-inequality, i.e. there exist $\mu > 0$ (free of $t$) such that

$$\left\|\nabla\tilde{\ell}(U_{11}(t), u_{-1}(t))\right\|_2^2 \geqslant \mu\left(\tilde{\ell}(U_{11}(t), u_{-1}(t)) - \min\tilde{\ell}(U_{11}, u_{-1})\right). \tag{48}$$

Then gradient flow converges to a global minimum of the population loss. Moreover, $W^{PV}$ and $W^{KQ}$ converge to $W_*^{PV}$ and $W_*^{KQ}$ respectively, where

$$w_{22}^{PV*} = \left\|\left(\frac{n+1}{n}\mathbb{E}\beta\beta^\top + \left(\frac{1}{n}\mathbb{E}\beta^\top\beta + \frac{\vartheta^2}{n}\right)I\right)^{-1}\mathbb{E}\beta\beta^\top\right\|_F^{\frac{1}{2}}, \tag{49}$$

$$W_{11}^{KQ*} = \left(w_{22}^{PV*}\right)^{-1}\left(\frac{n+1}{n}\mathbb{E}\beta\beta^\top + \left(\frac{1}{n}\mathbb{E}\beta^\top\beta + \frac{\vartheta^2}{n}\right)I\right)^{-1}\mathbb{E}\beta\beta^\top \tag{50}$$

*Proof.* We first characterize the behavior of $\tilde{\ell}(U_{11}(t), u_{-1}(t))$ on the gradient flow. Define

$$\Gamma = \frac{n+1}{n}\mathbb{E}\beta\beta^\top + \left(\frac{1}{n}\mathbb{E}\beta^\top\beta + \frac{\vartheta^2}{n}\right)I. \tag{51}$$

Note that

$$\frac{1}{2}\operatorname{tr}\left[\Gamma\left(u_{-1}U_{11} - \Gamma^{-1}\mathbb{E}\beta\beta^\top\right)\left(u_{-1}U_{11} - \Gamma^{-1}\mathbb{E}\beta\beta^\top\right)^\top\right]$$

$$=\frac{1}{2}\operatorname{tr}\left[\Gamma\left(u_{-1}^2U_{11}U_{11}^\top - 2u_{-1}U_{11}\Gamma^{-1}\mathbb{E}\beta\beta^\top + \Gamma^{-1}\mathbb{E}\beta\beta^\top\Gamma^{-1}\mathbb{E}\beta\beta^\top\right)\right]$$

$$=\frac{1}{2}\operatorname{tr}\left[u_{-1}^2U_{11}U_{11}^\top\Gamma - 2u_{-1}U_{11}\mathbb{E}\beta\beta^\top + \mathbb{E}\beta\beta^\top\Gamma^{-1}\mathbb{E}\beta\beta^\top\right]$$

$$=\frac{1}{2}u_{-1}^2\operatorname{tr}\left[U_{11}U_{11}^\top P\right] - u_{-1}\operatorname{tr}\left[U_{11}\mathbb{E}\beta\beta^\top\right] + \frac{1}{2}\operatorname{tr}\left[\Gamma^{-1}\left(\mathbb{E}\beta\beta^\top\right)^2\right].$$

Hence,

$$\tilde{\ell}(U_{11}, u_{-1}) = \frac{1}{2}\operatorname{tr}\left[\Gamma\left(u_{-1}U_{11} - \Gamma^{-1}\mathbb{E}\beta\beta^\top\right)\left(u_{-1}U_{11} - \Gamma^{-1}\mathbb{E}\beta\beta^\top\right)\right] - \frac{1}{2}\operatorname{tr}\left[\Gamma^{-1}\left(\mathbb{E}\beta\beta^\top\right)^2\right] + \mathbb{E}\beta^\top\beta + \vartheta^2.$$

Besides, by chain rule, we have

$$\frac{d\tilde{\ell}}{dt} = \left\langle\frac{d\tilde{\ell}}{dU_{11}}, \frac{dU_{11}}{dt}\right\rangle + \left\langle\frac{d\tilde{\ell}}{du_{-1}}, \frac{du_{-1}}{dt}\right\rangle$$

$$= -\left\langle\frac{dU_{11}}{dt}, \frac{dU_{11}}{dt}\right\rangle - \left\langle\frac{du_{-1}}{dt}, \frac{du_{-1}}{dt}\right\rangle \leqslant 0.$$

Also, $\tilde{\ell}$ is a quadratic function w.r.t, $u_{-1}$ and $\tilde{\ell}(U_{11}, u_{-1}) = \mathbb{E}\beta^\top\beta + \vartheta^2$ when $u_{-1} = 0$. Hence, if we can show that for all $t > 0$, $\tilde{\ell} \leqslant \mathbb{E}\beta^\top\beta + \vartheta^2$, we can claim $u_{-1}(t) \geqslant 0$ by the property of quadratic function. Now under the initialization condition equation 11 with $\operatorname{tr}\left(\Theta\Theta^\top\left(\mathbb{E}\beta\beta^\top\right)\Theta\Theta^\top\left(\mathbb{E}\beta\beta^\top\right)\right) = 1$, at $t = 0$ we have

$$\tilde{\ell}(U_{11}(0), u_{-1}(0))$$

$$
\begin{aligned}
=&\frac{1}{2}\gamma^4\operatorname{tr}\left[\Gamma\left(\mathbb{E}\beta\beta^\top\right)^{\frac{1}{2}}\Theta\Theta^\top\left(\mathbb{E}\beta\beta^\top\right)\Theta\Theta^\top\left(\mathbb{E}\beta\beta^\top\right)^{\frac{1}{2}}\right]-\gamma^2\operatorname{tr}\left[\left(\mathbb{E}\beta\beta^\top\right)^2\Theta\Theta^\top\right]+\mathbb{E}\beta^\top\beta+\vartheta^2\\
=&\frac{1}{2}\gamma^4\operatorname{tr}\left[\Gamma\left(\mathbb{E}\beta\beta^\top\right)\Theta\Theta^\top\left(\mathbb{E}\beta\beta^\top\right)\Theta\Theta^\top\right]-\gamma^2\operatorname{tr}\left[\left(\mathbb{E}\beta\beta^\top\right)^2\Theta\Theta^\top\right]+\mathbb{E}\beta^\top\beta+\vartheta^2\\
\leqslant&\frac{1}{2}\gamma^4\sqrt{d}\|\Gamma\|_{op}\left\|\mathbb{E}\beta\beta^\top\Theta\Theta^\top\mathbb{E}\beta\beta^\top\Theta\Theta^\top\right\|_F-\gamma^2\left\|\Theta^\top\left(\mathbb{E}\beta\beta^\top\right)^{\frac{1}{2}}\right\|_F^2+\mathbb{E}\beta^\top\beta+\vartheta^2\\
=&\frac{1}{2}\gamma^2\left\|\Theta^\top\left(\mathbb{E}\beta\beta^\top\right)\right\|_F^2\left(\sqrt{d}\gamma^2\|\Gamma\|_{op}\left\|\Theta\Theta^\top\right\|_F-2\right)+\mathbb{E}\beta^\top\beta+\vartheta^2
\end{aligned}
$$

where the second equality follows from the fact that $\mathbb{E}\beta\beta^\top$ commute with $\Gamma$. Therefore, as long as $\sqrt{d}\gamma^2\|\Gamma\|_{op}\left\|\Theta\Theta^\top\right\|_F\leqslant 2$, i.e. $\gamma\leqslant\sqrt{\frac{2}{\sqrt{d}\|\Gamma\|_{op}\|\Theta\Theta^\top\|_F}}$, then we will have $u_{-1}(t)\geqslant 0$. Now we just show that condition equation 12 is a sufficient condition by finding a lower bound for $\frac{1}{\|\Theta\Theta^\top\|_F}$.

$$
\begin{aligned}
1=&\operatorname{tr}\left(\mathbb{E}\beta\beta^\top\Theta\Theta^\top\mathbb{E}\beta\beta^\top\Theta\Theta^\top\right)\\
\geqslant&\operatorname{tr}\left(\Theta\Theta^\top\mathbb{E}\beta\beta^\top\Theta\Theta^\top\right)\lambda_{\min}\left(\mathbb{E}\beta\beta^\top\right)\\
\geqslant&\operatorname{tr}\left[\left(\Theta\Theta^\top\right)^2\right]\lambda_{\min}^2\left(\mathbb{E}\beta\beta^\top\right)\\
=&\|\Theta\Theta^\top\|_F^2\lambda_{\min}^2\left(\mathbb{E}\beta\beta^\top\right)
\end{aligned}
$$

we have $\frac{1}{\|\Theta\Theta^\top\|_F^2}\geqslant\lambda_{\min}^2\left(\mathbb{E}\beta\beta^\top\right)$. Besides, $\|\Gamma\|_{op}=\frac{n+1}{n}\lambda_{\max}(\mathbb{E}\beta\beta^\top)+\frac{1}{n}(\vartheta^2+\mathbb{E}\beta^\top\beta)$. Hence, under condition equation 12, we have

$$
\begin{aligned}
\gamma\leqslant&\sqrt{\frac{2\lambda_{\min}\left(\mathbb{E}\beta\beta^\top\right)}{\sqrt{d}\left(\frac{n+d+1}{n}\lambda_{\max}\left(\mathbb{E}\beta\beta^\top\right)+\frac{\vartheta^2}{n}\right)}}\leqslant\sqrt{\frac{2\lambda_{\min}\left(\mathbb{E}\beta\beta^\top\right)}{\sqrt{d}\left(\frac{n+1}{n}\lambda_{\max}\left(\mathbb{E}\beta\beta^\top\right)+\frac{\operatorname{tr}\mathbb{E}\beta\beta^\top}{n}+\frac{\vartheta^2}{n}\right)}}\\
\leqslant&\sqrt{\frac{2}{\sqrt{d}\|\Gamma\|_{op}\|\Theta\Theta^\top\|_F}}
\end{aligned}
$$

and $\tilde{\ell}<\mathbb{E}\beta^\top\beta+\vartheta^2$. Next, we prove equation 47 by contradiction. Note that we can lower bound $\tilde{\ell}(U_{11},u_{-1})$ as

$$
\begin{aligned}
\tilde{\ell}\left(U_{11},u_{-1}\right)=&\operatorname{tr}\left[\frac{1}{2}u_{-1}^2\Gamma U_{11}U_{11}^\top-u_{-1}\mathbb{E}\beta\beta^\top U_{11}^\top\right]+\mathbb{E}\beta^\top\beta+\vartheta^2\\
\geqslant&-\operatorname{tr}\left[u_{-1}\mathbb{E}\beta\beta^\top U_{11}^\top\right]+\mathbb{E}\beta^\top\beta+\vartheta^2\\
\geqslant&-\sqrt{d}u_{-1}\left\|\mathbb{E}\beta\beta^\top\right\|_{op}\|U_{11}\|_F+\mathbb{E}^\top\beta+\vartheta^2
\end{aligned}
$$

where the first inequality follows from that $\Gamma$ and $U_{11}U_{11}^\top$ are positive definite. Besides, at time $t=0$, we have

$$
\tilde{\ell}\left(U_{11}(0),u_{-1}(0)\right)\leqslant-\frac{1}{2}\gamma^2\|\Theta^\top\mathbb{E}\beta\beta^\top\|_F^2\left(2-\sqrt{d}\gamma^2\|\Gamma\|_{op}\left\|\Theta\Theta^\top\right\|_F\right)+\mathbb{E}\beta^\top\beta+\vartheta^2.
$$

Suppose that the opposite of equation 47 holds, i.e.

$$
u_{-1}<\sqrt{\frac{\gamma^2\|\Theta^\top\mathbb{E}\beta\beta^\top\|_F^2}{2\sqrt{d}\|\mathbb{E}\beta\beta^\top\|_{op}}\left(2-\sqrt{d}\gamma^2\|\Gamma\|_{op}\left\|\Theta\Theta^\top\right\|_F\right)},
$$

then we will have

$$
-\sqrt{d}u_{-1}^2\left\|\mathbb{E}\beta\beta^\top\right\|_{op}>-\frac{1}{2}\gamma^2\|\Theta^\top\mathbb{E}\beta\beta^\top\|_F^2\left(2-\sqrt{d}\gamma^2\|\Gamma\|_{op}\left\|\Theta\Theta^\top\right\|_F\right)
$$

and thus,

$$
\tilde{\ell}\left(U_{11},u_{-1}\right)>\tilde{\ell}\left(U_{11}(0),u_{-1}(0)\right).
$$

However, we also have $d\tilde{\ell}/dt \leqslant 0$. Therefore, we must have

$$u_{-1} \geqslant \sqrt{\frac{\gamma^2 \|\Theta^\top \mathbb{E}\beta\beta^\top\|_F^2}{2\sqrt{d}\,\|\mathbb{E}\beta\beta^\top\|_{op}}} \left(2 - \sqrt{d}\gamma^2\|\Gamma\|_{op}\,\|\Theta\Theta^\top\|_F\right) > 0.$$

Now we prove PL-inequality. Note that

$$\tilde{\ell}(U_{11}, u_{-1}) - \min \tilde{\ell}(U_{11}, u_{-1}) = \frac{1}{2}\operatorname{tr}\left[\Gamma\left(u_{-1}U_{11} - \Gamma^{-1}\mathbb{E}\beta\beta^\top\right)\left(u_{-1}U_{11} - \Gamma^{-1}\mathbb{E}\beta\beta^\top\right)\right]$$
$$= \frac{1}{2}\left\|\Gamma^{\frac{1}{2}}\left(u_{-1}U_{11} - \Gamma^{-1}\mathbb{E}\beta\beta^\top\right)\right\|_F^2.$$

Besides

$$\frac{\partial \tilde{\ell}}{\partial U_{11}} = -\frac{dU_{11}}{dt} = u^2_{-1}U_{11}\Gamma - u_{-1}\mathbb{E}\beta\beta^\top,$$

$$\left\|\nabla\tilde{\ell}(U_{11}(t), u_{-1}(t))\right\|_2^2 = \left\|\frac{\partial \tilde{\ell}}{\partial U_{11}}\right\|_F^2 + \left|\frac{\partial \tilde{\ell}}{\partial u_{-1}}\right|^2$$
$$\geqslant \left\|u^2_{-1}U_{11}\Gamma - u_{-1}\mathbb{E}\beta\beta^\top\right\|_F^2$$
$$= u^2_{-1}\left\|u_{-1}U_{11}\Gamma - \mathbb{E}\beta\beta^\top\right\|_F^2.$$

Since $\mathbb{E}\beta\beta^\top$ commutes with $\Gamma$, we have

$$u_{-1}U_{11}\Gamma - \mathbb{E}\beta\beta^\top = u_{-1}U_{11}\Gamma - \Gamma^{-1}\Gamma\mathbb{E}\beta\beta^\top$$
$$= u_{-1}U_{11}\Gamma - \Gamma^{-1}\mathbb{E}\beta\beta^\top\Gamma$$
$$= \left(u_{-1}U_{11} - \Gamma^{-1}\mathbb{E}\beta\beta^\top\right)\Gamma.$$

Therefore, it holds that

$$\Gamma^{\frac{1}{2}}\left(u_{-1}U_{11} - \Gamma^{-1}\mathbb{E}\beta\beta^\top\right) = \Gamma^{\frac{1}{2}}\left(u_{-1}U_{11}\Gamma - \mathbb{E}\beta\beta^\top\right)\Gamma^{-1}$$

and

$$\tilde{\ell}(U_{11}, u_{-1}) - \min \tilde{\ell}(U_{11}, u_{-1}) \leqslant \frac{1}{2}\left\|\Gamma^{\frac{1}{2}}\right\|_F^2 \left\|u_{-1}U_{11}\Gamma - \mathbb{E}\beta\beta^\top\right\|_F^2 \left\|\Gamma^{-1}\right\|_F^2.$$

Now if we set $\mu$ such that

$$u^2_{-1}\left\|u_{-1}U_{11}\Gamma - \mathbb{E}\beta\beta^\top\right\|_F^2 \geqslant \frac{1}{2}\mu\left\|\Gamma^{\frac{1}{2}}\right\|_F^2 \left\|u_{-1}U_{11}\Gamma - \mathbb{E}\beta\beta^\top\right\|_F^2 \left\|\Gamma^{-1}\right\|_F^2,$$

then PL-inequality holds, i.e.

$$\left\|\nabla\tilde{\ell}(U_{11}(t), u_{-1}(t))\right\|_2^2 \geqslant u^2_{-1}\left\|u_{-1}U_{11}\Gamma - \mathbb{E}\beta\beta^\top\right\|_F^2$$
$$\geqslant \frac{1}{2}\mu\left\|\Gamma^{\frac{1}{2}}\right\|_F^2 \left\|u_{-1}U_{11}\Gamma - \mathbb{E}\beta\beta^\top\right\|_F^2 \left\|\Gamma^{-1}\right\|_F^2$$
$$\geqslant \mu\left[\tilde{\ell}(U_{11}, u_{-1}) - \min \tilde{\ell}(U_{11}, u_{-1})\right].$$

In particular, using lower bound of $u_{-1}$, we can set

$$\mu = \frac{\gamma^2\|\Theta^\top\mathbb{E}\beta\beta^\top\|_F^2}{\sqrt{d}\left\|\Gamma^{\frac{1}{2}}\right\|_F^2 \|\Gamma^{-1}\|_F^2 \|\mathbb{E}\beta\beta^\top\|_{op}}\left(2 - \sqrt{d}\gamma^2\|\Gamma\|_{op}\,\|\Theta\Theta^\top\|_F\right) > 0. \tag{52}$$

Now, by the dynamics of gradient flow,

$$\frac{d}{dt}\left[\tilde{\ell}(U_{11}(t), u_{-1}(t))\right] = -\left\|\frac{dU_{11}}{dt}\right\|_F^2 - \left|\frac{du_{-1}}{dt}\right|^2 \leqslant -\mu\left[\tilde{\ell}(U_{11}(t), u_{-1}(t)) - \min \tilde{\ell}(U_{11}, u_{-1})\right].$$

Hence,

$$0 \leqslant \tilde{\ell}\left(U_{11}(t), u_{-1}(t)\right) - \min \tilde{\ell}\left(U_{11}, u_{-1}\right) \leqslant \exp(-\mu t)\left[\tilde{\ell}\left(U_{11}(0), u_{-1}(0)\right) - \min \tilde{\ell}\left(U_{11}, u_{-1}\right)\right] \to 0$$

i.e.

$$\lim_{t \to \infty}\left[\tilde{\ell}\left(U_{11}(t), u_{-1}(t)\right) - \min \tilde{\ell}\left(U_{11}, u_{-1}\right)\right] = \lim_{t \to \infty}\left\|\Gamma^{\frac{1}{2}}\left(u_{-1}U_{11} - \Gamma^{-1}\mathbb{E}\beta\beta^{\top}\right)\right\|_{F}^{2} \to 0.$$

Hence, we have $u_{-1}U_{11} \to \Gamma^{-1}\mathbb{E}\beta\beta^{\top}$. Besides, $u_{-1}^{2} = \operatorname{tr}\left(U_{11}U_{11}^{\top}\right) = \|U_{11}\|_{F}^{2}$, we have as $t \to \infty$

$$u_{-1} \to \left\|\Gamma^{-1}\mathbb{E}\beta\beta^{\top}\right\|_{F}^{\frac{1}{2}},$$
$$U_{11} \to \left\|\Gamma^{-1}\mathbb{E}\beta\beta^{\top}\right\|_{F}^{-\frac{1}{2}}\Gamma^{-1}\mathbb{E}\beta\beta^{\top}.$$

$\square$

## C  Auxiliary Results

**Proposition C.1** (Proposition C.2 in Bai et al. (2024)). *Let $\ell(\cdot, \cdot) : \mathbb{R}^2 \to \mathbb{R}$ be a loss function such that $\partial_1\ell$ is $(\varepsilon, R, M, C)$-approximable by sum of relus with $R = \max\left\{B_x B_w, B_y, 1\right\}$. Let $\widehat{L}_n(\beta) := \frac{1}{n}\sum_{i=1}^{n}\ell\left(\beta^{\top}x_i, y_i\right)$ denote the empirical risk with loss function $\ell$ on dataset $\left\{(x_i, y_i)\right\}_{i \in [n]}$. Then, for any $\varepsilon > 0$, there exists an attention layer $\left\{(Q_m, K_m, V_m)\right\}_{m \in [M]}$ with $M$ heads such that, for any input sequence that takes form $h_i = \left[x_i; y_i'; \beta; 0_{D-2d-3}; 1; t_i\right]$ with $\|\beta\|_2 \leqslant B_w$, it gives output*

$$\widetilde{h}_i = \left[\operatorname{Attn}_{\boldsymbol{\theta}}(H)\right]_i = \left[x_i; y_i'; \widetilde{\beta}; 0_{D-2d-3}; 1; t_i\right]$$

*for all $i \in [N+1]$, where*

$$\left\|\widetilde{\beta} - \left(\beta - \eta\nabla\widehat{L}_n(\beta)\right)\right\|_2 \leqslant \varepsilon \cdot \left(\eta B_x\right).$$

**Proposition C.2** (Proposition 1 in Pathak et al. (2024)). *Given any input matrix $H \in \mathbb{R}^{p \times q}$ that output a matrix $H' \in \mathbb{R}^{p \times q}$, following operators can be implemented by a single layer of an autoregressive transformer:*

- *copy_down$\left(H; k, k', \ell, \mathcal{I}\right)$ : For columns with index $i \in \mathcal{I}$, outputs $H'$ where*

$$H'_{k':\ell',i} = H_{k:\ell,i}$$

  *and the remaining entries are unchanged. Here, $\ell' = k' + (\ell - k)$ and $k' \geqslant k$, so that entries are copied "down" within columns $i \in \mathcal{I}$. Note, we assume $\ell \geqslant k$ and that $k' \leqslant q$ so that the operator is well-defined.*

- *copy_over$\left(H; k, k', \ell, \mathcal{I}\right)$ : For columns with index $i \in \mathcal{I}$, outputs $H'$ with*

$$H'_{k':\ell',i} = H_{k:\ell,i-1}.$$

  *The remaining entries stay the same. Here entries from column $i-1$ are copied "over" to column $i$.*

- *mul$\left(H; k, k', k'', \ell, \mathcal{I}\right)$ : For columns with index $i \in \mathcal{I}$, outputs $H'$ where*

$$H'_{k''+t,i} = H_{k+t,i}H_{k'+t,i}, \quad for \ t \in \{0, \dots, \ell - k\}.$$

  *Note that $\ell'' = k'' + \delta''$ where $W \in \mathbb{R}^{\delta'' \times \delta}, W' \in \mathbb{R}^{\delta'' \times \delta'}$ and $\ell = k + \delta, \ell' = k' + \delta'$. We assume $\delta, \delta', \delta'' \geqslant 0$. The remaining entries of $H$ are copied over to $H'$, unchanged.*

- *scaled_agg$\left(H; \alpha, k, \ell, k', i, \mathcal{I}\right)$ : Outputs a matrix $H'$ with entries*

$$H_{k'+t,i} = \alpha\sum_{j \in \mathcal{I}}H_{k+t,j} \quad for \ t \in \{0, 1, \dots, \ell - k\}.$$

  *The set $\mathcal{I}$ is causal, so that $\mathcal{I} \subset [i-1]$. The remaining entries of $H$ are copied over to $H'$, unchanged.*

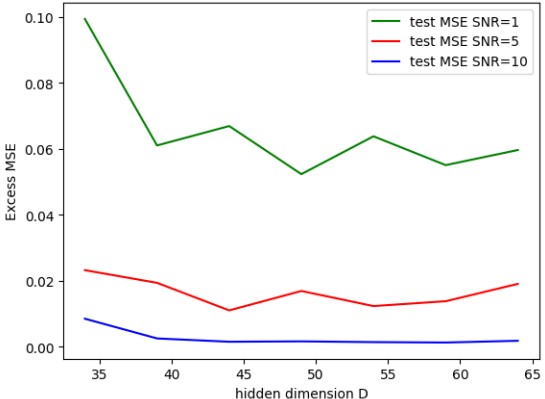

Figure 4: Plot of excess testing risk of the transformer v.s. the hidden dimension D with different SNRs.

- soft $\left(H; k, \ell, k'\right)$ : *For the final column q, outputs a matrix H′ with entries*

$$H'_{k'+t,q} = \frac{e^{H_{k+t,q}}}{\sum_{t'=0}^{\ell-k} e^{H_{k+t',q}}}, \quad for\ t \in \{0, 1, \ldots, \ell - k\}.$$

*The remaining entries of H are copied over to H′, unchanged.*

**Lemma C.1** (Lemma D.2 in Zhang et al. (2023))**.** If x is a Gaussian random vector of $d$ dimension, mean zero and covariance matrix $I_d$, and $\boldsymbol{A} \in \mathbb{R}^{d \times d}$ is a fixed matrix. Then

$$\mathbb{E}\left[\mathbf{x}\mathbf{x}^\top \boldsymbol{A} \mathbf{x}\mathbf{x}^\top\right] = \left(\boldsymbol{A} + \boldsymbol{A}^\top\right) + \mathrm{tr}(\boldsymbol{A})I_d.$$

If $\boldsymbol{A}$ is symmetric and the rows in $\mathbf{X} \in \mathbb{R}^{M \times d}$ are generated independently from $X_i \sim \mathcal{N}(0, I_d)$, then it holds that

$$\mathbb{E}\left[\mathbf{X}^\top \mathbf{X} \boldsymbol{A} \mathbf{X}^\top \mathbf{X}\right] = M \cdot \mathrm{tr}(\boldsymbol{A}) \cdot I_d + M(M+1)\boldsymbol{A}.$$

# D   Additional Details in Simulation

## D.1   Batch EM Algorithm

The procedure of EM algorithm is given by Algorithm 1. In our implementation we stop (or declare the algorithm converged) if the maximum iteration is attain, or if $\max_j \|\hat{\beta}_j^{(t)} - \hat{\beta}_j^{(t-1)}\|_2 \leqslant \varepsilon$, where $\varepsilon = 0.0001$ and the maximum iteration $T_{\max} = 10000$.

## D.2   Performance with different number of hidden dimension

In this experiment, we vary the hidden dimension $D = 34, 64, 128$. For each case, we run the experiment with two components ($K = 2$), different SNR ($\eta = 1, 5, 10$). The $x$-axis is the hidden dimension $D$, and the $y$-axis is the excess test MSE. The performance of the trained transformer is presented in Figure 4. In the low SNR settings, increasing the hidden dimension helps in improving the transformer's ability to learn the mixture of regression problem. However, excessively large hidden dimensions can lead to sparsity in the parameter matrix, which may not significantly enhance performance further.

---

**Algorithm 1** Batch EM algorithm for Mixture of Regression Problem

---

**Input:** Number of prompts $B > 0$, prompts $\{E^{(i)} : i = 1, \ldots, B\}$ of length $n + 1$ containing the data $\{(x_i, y_i), x_{n+1} : i = 1, \ldots, n\}$ generated based on Mixture of regression tasks with noise variance $\vartheta > 0$, number of components $K > 0$.

Initialize $\pi^{(0)} \in [0, 1]^K$, drawn uniformly on the probability simplex.

Initialize $\beta_j^{(0)} \in \mathbb{R}^d$, drawn uniformly on the sphere of radius 1 for $j \in [\ K]$ such that $\cos \angle(\beta_j^*, \beta_j^0) > 0.8$.

Initialize $\gamma_{ij}^{(0)} = 0$ for all $i \in [B], j \in [K]$.

**while** have not converged **do**

   Update prompt-component assignment probabilities, for all $i \in [B], j \in [K]$.

$$
\gamma_{ij}^{(t+1)} = \frac{\pi_j^{(t)} \prod_{\ell=1}^n \exp \left\{ -\frac{(y_\ell^{(i)} - x_\ell^{(i)\top} \beta_j^{(t)})^2}{2\vartheta^2} \right\}}{\sum_{j'=1}^K \pi_{j'}^{(t)} \prod_{\ell=1}^n \exp \left\{ -\frac{(y_\ell^{(i)} - x_\ell^{(i)\top} \beta_{j'}^{(t)})^2}{2\vartheta^2} \right\}}.
$$

   Update the marginal component probabilities for all $j \in [K]$

$$
\pi_j^{(t+1)} = \frac{1}{B} \sum_{i=1}^B \gamma_{ij}^{(t+1)}.
$$

   Update the parameter estimates for all $j \in [K]$

$$
\beta_j^{(t+1)} = \arg \min_{\beta \in \mathbb{R}^d} \left\{ \sum_{i=1}^B \sum_{\ell=1}^n \gamma_{ij}^{(t+1)} \left( y_\ell^{(i)} - \beta^\top x_\ell^{(i)} \right)^2 \right\}.
$$

   Update the iteration counter, $t \leftarrow t + 1$.

**end while**

**Return:** Final set of component centers, $\left\{ \beta_j^{(t)} \right\}_{j=1}^K$

---

