# OpenReview forum: "In-context Learning for Mixture of Linear Regression: Existence, Generalization and Training Dynamics"
_TMLR — Accepted by TMLR_

### Review · Reviewer_nBsS · 2025-06-13

**Summary Of Contributions:**

The authors examine the capability of Transformer structures in performing the mixture of (linear) regression (MoR) task. The authors provide a construction-based Transformer weight that performs the Expectation-Maximization (EM) algorithm by layer-wise gradient ascent for the M-step. This construction gives an approximation gap for the linear MoR tasks. On the other hand, the authors derive a generalization result for the pre-training of Transformers on MoR tasks. Following the simplified model (a one-layer linear-attention-only Transformer), the authors analyze the converging point of the gradient flow. Some numerical results are provided to support the efficiency of Transformers in MoR tasks.

**Audience:**

Yes

**Claims And Evidence:**

Yes

**Requested Changes:**

I will be grateful if the authors could explain more about the following two points:

1. How do you separate the possibility of in-weight learning (IWL) from ICL with your EM explanations? I'm quite curious because it seems that the only possible $\beta$'s in the test phase are those in the training phase.

2. How do you view the SNR when the number of possible $\beta_i$'s tends to infinity? See Weaknesses part 2.

**Strengths And Weaknesses:**

Strengths:

1. Although linear regression tasks have been widely studied in understanding the dynamics of in-context learning, the perspective from the EM algorithm is novel.

2. The authors provide some concrete constructions to factorize the M-step into several gradient ascent steps, which share the same spirit of the gradient descent dynamics in previous works.

3. Presentation is good.

Weaknesses:

1. In-context learning requires the model to be able to generalize to a broader range outside the possible combinations of observed linear regression tasks. In that sense, the ability to accomplish MoR tasks is still different from the in-context learning ability. Understanding ICL via EM is questionable, especially when compared to the previous gradient descent perspective that can deal with any new linear regression task.

2. The emergence of in-context learning ability in the mixture of linear regression task requires quite a number of different linear weights (e.g., see *Pretraining task diversity and the emergence of non-bayesian in-context learning for regression*). Under such conditions, the minimum gap between $\beta_i$'s is quite small, leading to a very small SNR (defined in this paper). However, only after training on such a large corpus would the model be able to perform ICL. In that sense, the SNR arguments in this paper cannot fully convince me.

---

> ### Author Response · Authors · 2025-08-26
>
> - **W1**: In-context learning requires the model to be able to generalize to a broader range outside the possible combinations of observed linear regression tasks. In that sense, the ability to accomplish MoR tasks is still different from the in-context learning ability. Understanding ICL via EM is questionable, especially when compared to the previous gradient descent perspective that can deal with any new linear regression task.
>
>   Our scope is Mixture of Regressions (MoR) tasks (Def. and SNR in eqs. (1)–(2)). Within this family we prove that a transformer can implement gradient-EM inside its forward pass and achieve quantitative error guarantees (Thms. 3.1–3.3). This complements—rather than contradicts—the GD perspective for single-task linear regression: in the $ K=1 $ special case the problem reduces to standard linear regression, and our training-dynamics result (Thm. 4.2) shows a linear self-attention layer trained by gradient flow converges to the oracle linear predictor in the large-sample limit, matching the GD/ERM view.
>
>   Our claims are scoped exactly that way, and EM is used here as the right mechanism for a broad latent-variable family rather than a universal algorithm. In the paper, ICL risk is taken over new tasks sampled from a task distribution (mixtures of regressions). At test time the model sees a fresh prompt from a new task instance; it is not restricted to combinations of a fixed training pool. Our pretraining generalization bound is stated over this distribution and scales with SNR, task balance, and model capacity. For $K \geq 2$, tasks contain latent structure. EM is the statistically natural procedure: the E -step computes responsibilities and the M-step updates parameters, yielding the low- vs. high-SNR regimes and rates we prove. Anchoring the transformer to EM lets us give quantitative guarantees. Regarding difference on distribution between training and new tasks, the new task are sampled from the same prior as the training tasks  (see [Garg et al., 2022], [Zhang et al., 2023]).
>
> - **W2**: The emergence of in-context learning ability in the mixture of linear regression task requires quite a number of different linear weights (e.g., see Pretraining task diversity and the emergence of non-bayesian in-context learning for regression). Under such conditions, the minimum gap between $\beta_{i}$'s is quite small, leading to a very small SNR (defined in this paper). However, only after training on such a large corpus would the model be able to perform ICL. In that sense, the SNR arguments in this paper cannot fully convince me.
>
>   In our paper the SNR is per-task, defined by the separation of the $K$ mixture components within a single prompt relative to noise (minimum pairwise distance among the active  $ \beta_{k}^{*} $’s divided by the noise scale). It is not the minimum distance across all regression vectors that may appear anywhere in the entire pretraining corpus. Thus, having many distinct tasks in pretraining (some very close to others) does not by itself force the per-prompt SNR to be small. We will make this explicit by restating the SNR definition alongside the MoR model and emphasizing it is computed within the task underlying the current prompt.
>
>   Our current model (and the proofs) assume fixed ground-truth coefficients $ \beta_{k}^{ * } $'s, ensuring $ x^{ \top } \beta_{k}^{ * }+v$ is sub-Gaussian. This is purely done for technical convenience. We can instead model pretraining by placing a prior on **each** $\beta_k^*$ (e.g., Gaussian or uniform on the sphere). In this formulation, then the $K=1$ case reduces to the setting analyzed by [Ravent´os et al., 2023] . Under such random-effects assumptions, the central claims will remain valid with appropriate technical (and some non-trivial). arguments. For example, we would need re-derive Lemmas A.6-A.7. Based on that the currently steps in the proofs of Theorems 3.1-3.2 has to be redeveloped. With a modified definition, the prompt-conditioned SNR analogue under random coefficients that still quantifies the statistical hardness of a given prompt. Carry this out fully is beyond the scope of the current submission and we view this as a promising direction for future work.
>
> [Garg et al., 2022] Garg, S., Tsipras, D., Liang, P. S., and Valiant, G. (2022). What can transformers learn in-
> context? A case study of simple function classes. Advances in Neural Information Processing Systems, 35:30583–
> 30598.
>
> [Zhang et al., 2023] Zhang, R., Frei, S., and Bartlett, P. L. (2023). Trained transformers learn linear models in-
> context. arXiv preprint arXiv:2306.09927.
>
> [Ravent´os et al., 2023] Ravent´os, A., Paul, M., Chen, F., and Ganguli, S. (2023). Pretraining task diversity and the
> emergence of non-bayesian in-context learning for regression. Advances in neural information processing systems,
> 36:14228–14246.

---

### Review · Reviewer_3dAV · 2025-07-15

**Summary Of Contributions:**

The authors investigate the use of transformer architectures for solving Mixture of linear Regression (MoR) problems. The analysis covers representation, statistical, and optimisation properties. First, they show that the transformer architecture is capable of representing the EM algorithm. Second, they establish that in-context learning estimators with low risk can be established by minimising an empirical risk. Finally, they show that gradient flow can realise such estimators. Hence, transformers for MoR are sufficiently rich, admit desirable statistical properties, and can be optimised (at least under idealised assumptions). They apply their model to some MoR problems, showing empirically faster convergence than EM algorithm, and investigating various hyperparameters such as number of examples and prompt length.

**Audience:**

Yes

**Broader Impact Concerns:**

No broader impact concerns.

**Claims And Evidence:**

Yes

**Requested Changes:**

I am keen to upgrade my evaluation of "Claims and Evidence". Please add text that either addresses my confusions in the weaknesses above, or appropriately justifies any choices made. Additionally:

- It would be helpful between equations (1) and (2) to state that $\nu$ is known (I believe, anyway).
- Remark 2.1. You could possibly further justify not using softmax, through citing various other works which use non-softmax functions. E.g. the literature on random Fourier features, or random features more generally.
- Section 4.1 Is clipping standard in practice? If so, could you cite a paper or two?
- I am confused by the terminology of "pretraining". Isn't this just standard training, before ICL happens?

Minor typos:
- "and largest distancerespectively." between equations (1) and (2).
- Definition 2.1. "A attention layer" should be "An attention layer".
- The incorrect usage of the word "Besides" throughout is a bit jarring.
- The grammar in theorem 3.1 is not quite right.

**Strengths And Weaknesses:**

Overall, this is a strong paper of interest to the community. However, I have some concerns that I would like to see addressed (see weaknesses and requested changes).

Strengths:
- The paper is split into a logical order, roughly following the order of Existence/representation -> statistical properties -> optimisation. Section 3 looks at the existence/representation abilities of the transformer architecture. Then, section 4 first looks at statistical properties of the estimator (section 4.1), and then shows that such estimators are realisable via gradient flow (section 4.2). The paper ends with some empirical investigations (section 5).
- It is nice to see two different regimes quantified in theorem 3.3, which describes the excess risk of the MoR model, governed by the value of the SNR.
- Theorem 3.1 and Theorem 3.2 describe the high probability bounds of the prediction error of the MoR model, with various parameters controlling the bound including dimension $d$ and prompt length $n$.




Weaknesses:
- Throughout the main paper, there is a lot of heavy lifting done by the phrase "can implement the EM algorithm". It is very difficult to understand precisely what the authors mean by this, without consulting Appendix A (and no helpful pointer to Appendix A is given in the main paper). I believe what it means is that the derivative of the function defined in the $E$ step can be approximated by a sum of ReLU functions (internal to the transformer), and this somehow translates to performing iterates of EM algorithm through the layers. I am having difficulty understanding why the EM algorithm in particular is of interest - wouldn't we expect such networks to be capable of representing lots of algorithms?
    - Appendix A is quite challenging to read. I am having trouble seeing what the difference is between $H^{(T+1)}(3)$ and $H^{(T+1)}(4)$. The authors could direct the reader to what is changing between each line.
- Equation (3). I am not sure if I am misunderstanding something, so I'm happy to be corrected by the authors here. It seems to me as though a special kind of input preprocessing/representation used, whereby the data is padded with zeros and the indicator function. This seems like it might be different to what people would do in practice, revealing a gap between the theory presented in this paper and the reality of transformers.
- The first part (existence of a transformer which can implement EM algorithm) seems divorced from the second part (training dynamics under GD). My main concern is that the matrix $E$ is *assumed* in the second part, and this would either not happen in practice or would be inconsistent with the setup in the first part.

---

> ### Author Response · Authors · 2025-08-26
>
> - **W1**: Throughout the main paper, there is a lot of heavy lifting done by the phrase "can implement the EM algorithm".  I am having difficulty understanding why the EM algorithm in particular is of interest - wouldn't we expect such networks to be capable of representing lots of algorithms? Appendix A is quite challenging to read. I am having trouble seeing what the difference is between $H^{(T+1)}(3)$ and $H^{(T+1)}(4)$. The authors could direct the reader to what is changing between each line.
>
> What “implement EM” means for a given prompt, these forward passes of the network compute one EM iteration: an E-step to form soft responsibilities and an M-step executed as $T$ gradient-descent updates on the M-step objective. We agree transformers are expressive enough to simulate many procedures. We focus on EM because it is the statistically natural and analyzable mechanism for mixtures of regressions: (i) EM yields posterior-like responsibilities and a contraction analysis that cleanly explains the observed low- vs. high-SNR regimes and the resulting rates. (ii) Tying the transformer to EM lets us prove quantitative generalization/error guarantees (as functions of SNR, balance, $n$, and model capacity) and connect to our training-dynamics result. With an arbitrary algorithm, we would lose this tight theory-mechanism link. (iii) Empirically, EM offers mechanism-level probes (responsibility alignment, parameter-update tracking) that we can test to show the model is doing more than just fitting MSE. There are some papers showing that transformers are capable of implementing many statitical algorithms, see [Bai et al., 2024], [Pathak et al., 2024] and In our work, we provide quantitative approximation bounds under this natural statistical model.
>
> On $H^{(T+1)}(3)$ vs. $H^{(T+1)}(4)$. The only change is that the responsibility register is updated: the row/column containing $w_{\beta_T^{(t-1)}}\left(x_i, y_i\right)$ is overwritten by $w_{\beta_T^{(t)}}\left(x_i, y_i\right)$ for all tokens $i$. This update is exactly the E-step computed by the forward pass at that stage. We will add more details on what happens or changes in each step.
>
> - **W2**: Equation (3). It seems to me as though a special kind of input preprocessing/representation used, whereby the data is padded with zeros and the indicator function. This seems like it might be different to what people would do in practice, revealing a gap between the theory presented in this paper and the reality of transformers.
>
> We use for the representation in (3) just for theoretical convenience. Eq. (3) is not a data-preprocessing recipe. It specifies a state layout (“registers”) per token that our proof uses to show how one attention layer can read/write the quantities needed by the EM iteration: (i) the block of zeros is just scratch space in the hidden state for intermediate values; (ii) $t_i$ denotes position, i.e., the usual positional/segment tag used by transformers. In real training we feed standard token embeddings (for $x_i$ and $y_i$) plus ordinary positional encodings. A single input projection can create extra hidden dimensions initialized at zero-i.e., the "scratch registers"-without any special preprocessing.
>
> - **W3**: The matrix $E$ is assumed in the second part, and this would either not happen in practice or would be inconsistent with the setup in the first part.
>
> $E$ is just the prompt rendered as a matrix: the first $n$ columns are the observed pairs $\left(x_i, y_i\right)$, and the last column is the query feature $x_{n+1}$ with its label masked $(0)$. Nothing is assumed beyond the usual prompt; $E$ is a deterministic reformatting of inputs the model already receives. In Part I (existence/EM) we use a per-token register/state layout $H$ to prove that one forward pass can carry out an EM iteration. In Part II (dynamics) we compress the same information into the matrix $E$ to analyze gradient dynamics of a linear self-attention layer. These are equivalent encodings of the same prompt: The second part is not introducing a new assumption; it merely switches to a compact matrix form suited for the dynamics proof. Practically, we feed standard embeddings for ($x_i, y_i$) and a masked query $x_{n+1}$. A single input projection (or residual channel) provides the ``scratch" dimensions used in the existence construction. Writing the prompt as $E$ mirrors common implementations (padding/mask for the query label) and does not require bespoke preprocessing.
>
> [Bai et al., 2024] Bai, Y., Chen, F., Wang, H., Xiong, C., and Mei, S. (2024). Transformers as statisticians: Provable
> in-context learning with in-context algorithm selection. Advances in neural information processing systems, 36.
>
> [Pathak et al., 2024] Pathak, R., Sen, R., Kong, W., and Das, A. (2024). Transformers can optimally learn regres-
> sion mixture models. In The Twelfth International Conference on Learning Representations.

---

> > ### Author Response · Authors · 2025-08-26
> >
> > - It would be helpful between equations (1) and (2) to state that $\nu$ is known (I believe, anyway).
> >
> >   Yes. For the existence proof,  $\nu$ is known in Theorem 3.1 and 3.2 and treated as a fixed constant, because we need this quantity to construct the transformer.
> >
> > - Remark 2.1. You could possibly further justify not using softmax, through citing various other works which use non-softmax functions. E.g. the literature on random Fourier features, or random features more generally.
> >
> >   There are various works on transformers that use non-softmax function as activation, (e.g. [Shen et al., 2023], [Bai et al., 2024] for ReLu activation, [Luo et al., 2021] for kernel attention, [Qin et al., 2022] for cosFormer)
> >
> > - Section 4.1 Is clipping standard in practice? If so, could you cite a paper or two?
> >
> >   We use clipping as a technical device to keep functions Lipschitz/ bounded for the generalization proof. We choose the radius large enough that, on in-distribution prompts, the forward predictions are unaffected and prevent the transformer from blowing up on tail events (see [Bai et al., 2024]). Applying clipping operator on the objective functions when training LLM is used in RLHF (see [Ziegler et al., 2019], [Ouyang et al., 2022]).
> >
> > [Shen et al., 2023] Shen, K., Guo, J., Tan, X., Tang, S., Wang, R., and Bian, J. (2023). A study on relu and
> > softmax in transformer. arXiv preprint arXiv:2302.06461.
> >
> > [Bai et al., 2024] Bai, Y., Chen, F., Wang, H., Xiong, C., and Mei, S. (2024). Transformers as statisticians: Provable
> > in-context learning with in-context algorithm selection. Advances in neural information processing systems, 36.
> >
> > [Luo et al., 2021] Luo, S., Li, S., Cai, T., He, D., Peng, D., Zheng, S., Ke, G., Wang, L., and Liu, T.-Y. (2021).
> > Stable, fast and accurate: Kernelized attention with relative positional encoding. Advances in Neural Information
> > Processing Systems, 34:22795–22807.
> >
> > [Qin et al., 2022] Qin, Z., Sun, W., Deng, H., Li, D., Wei, Y., Lv, B., Yan, J., Kong, L., and Zhong, Y. (2022).
> > cosformer: Rethinking softmax in attention. arXiv preprint arXiv:2202.08791.
> >
> > [Ouyang et al., 2022] Ouyang, L., Wu, J., Jiang, X., Almeida, D., Wainwright, C., Mishkin, P., Zhang, C., Agarwal,
> > S., Slama, K., Ray, A., et al. (2022). Training language models to follow instructions with human feedback.
> > Advances in neural information processing systems, 35:27730–27744.
> >
> > [Ziegler et al., 2019] Ziegler, D. M., Stiennon, N., Wu, J., Brown, T. B., Radford, A., Amodei, D., Christiano, P.,
> > and Irving, G. (2019). Fine-tuning language models from human preferences. arXiv preprint arXiv:1909.08593.

---

> > > ### Comment · Reviewer_3dAV · 2025-09-01
> > >
> > > Thanks for your response and updated manuscript. You have clarified my main misunderstandings. I'm happy to increase my score, and recommend this paper be accepted. I've looked through the other reviews and do not see any significant outstanding problems regarding claims and evidence.

---

### Review · Reviewer_ejbj · 2025-08-12

**Summary Of Contributions:**

This work studied ICL for MoR models. It proves the existence of a Transformer which internally implements gradient EM algorithm of MoR problems, and derives the prediction error under low and high SNR regimes; also it provided the excess risk of pretraining with finite number of ICL instances, and further analyzed the training dynamics of gradient flow for single linear self-attention layer. Numerical experiments are provided to demonstrate the effectiveness of the proposed theory.

**Audience:**

Yes

**Claims And Evidence:**

No

**Requested Changes:**

1. I think it would be more convincing if authors can add some experiment to show that the Transformer actually execute gradient-EM, for example, construct a real Transformer following your theory, and plot the errors of prediction of $\beta$ versus low/high SNR as shown in your Theorem 3.1/3.2.
2. Regarding Theorem 3.2, whether the change between low SNR and high SNR occurs abruptly, i.e., the error jumps from $O(d\log^2(n/\delta)/n)^{1/4})$ to $O(d\log^2(n/\delta)/n)^{1/2})$, or is the change gradual as the SNR varies? I may expect the error is a function of SNR in the whole regime so it should change incrementally, otherwise an abrupt change corresponds to a phase transition, I hope to have more explanations on it.
3. Both Theorem 3.1 and 3.2 require the prompt length $n$ to be much larger than the input dimension $d$, and the model size $D$ is a multiple of $d$ (in experiments $D=64=2d$, also $n=64$), so possibly the theory may not work for the overparameterized case ($D$ is very large), can you comment on it?
4. Also $n=64$ means the setting $n\gg d$ may not be satisfied in your experiemnts, so I suggest authors consider adjusting the experimental setup or providing justification to better align with the theoretical assumptions.

**Strengths And Weaknesses:**

Strengths:
1. This work extended previous works into MoR tasks, and explicitly provided the excess risk bounds, which may be of independent interest.
2. The writing of the work is generally easy to follow.


Weakness:
1. The experiment is a bit disconnected with the theory. This work proved the existence of a Transformer which implements gradient EM, but the experiment only reported excess MSE comparison between Transformer and EM, and it is not clear that whether the Transformer really learned the EM as authors claimed.
2. The theory is a bit unintuitive to me. Theorem 3.2 indicates an abrupt change in the error, and there seems to be a "magic number" of SNR for a phase transition in the error growth, I am not sure whether such results is common in the community, also such pattern is not well-understood in theory and well-supported in your experiments based on my own understanding.
3. Gradient flow analysis is nice, but all in all it should correspond to a continuous-time algorithm, it would be nice to have more discussion and insights on the discrete-time algorithm design and analysis, otherwise it is unclear how directly the continuous-time results translate to realistic architectures.

---

> ### Author Response · Authors · 2025-08-26
>
> - **W1** This work proved the existence of a Transformer which implements gradient EM, but the experiment only reported excess MSE comparison between Transformer and EM, and it is not clear that whether the Transformer really learned the EM as authors claimed.
>
> Theorems 3.1-3.2 give an explicit construction of a transformer that executes gradient-EM within its forward pass-an E-step plus an M-step realized by $T$ gradient-descent updates-together with the SNR and prompt-length regimes under which the stated error rates hold. This is an **existence result**.
>
> In practice we pre-train by minimizing MSE for a chosen transformer model. So the actual mechanism by which transformer learns mixture of linear regression model is not restricted to the handcrafted parameterization and can match or surpass the constructive baseline. Our current experiments report end-to-end excess MSE, which demonstrates effectiveness of trained transform but does not, by itself, certify the mechanism via which transformers learns the mixture of linear regression model. However, they are capable of implementing the EM algorithm.
>
> - **W2** The theory is a bit unintuitive to me. Theorem 3.2 indicates an abrupt change in the error, and there seems to be a "magic number" of SNR for a phase transition in the error growth, I am not sure whether such results is common in the community, also such pattern is not well-understood in theory and well-supported in your experiments based on my own understanding.
>
> It’s a regime switch in EM analysis, not a literal discontinuity of the estimator or the output of the transformer. The population EM map is “strongly contractive” around the truth only when the SNR $\eta$ is large enough; otherwise it is nearly flat and cannot beat the classical $n^{-1 / 4}$ barrier for weakly identifiable mixtures.
>
> High SNR: When $\eta \gtrsim (d \log ^2(n / \delta) / n)^{1 / 4}$, the population EM operator contracts toward $\beta^*$ with factor $1-c \min(1, \eta^2)$ (so the local curvature is $\asymp \eta^2$ ). Balancing this contraction against the uniform sample fluctuation of the EM step—of order $\sqrt{d \log ^2(n / \delta) / n}$ -yields the parametric error.
>
> Low SNR: When $\eta \lesssim (d \log ^2(n / \delta) / n )^{1/4}$, EM essentially cannot tell apart $ \beta^* = 0$ from $ \beta^*  \neq 0$ . The population EM map (as well as gradient EM map) behaves like
>
> $$
> || \beta ||_2(1-4 || \beta ||_2^2-c_u || \beta^* ||_2^2) \leq \|\mathbb{E}[\tanh (Y X^{\top} \beta ) Y X]\|_2 \leq || \beta ||_2 (1-\|\beta\|_2^2+c_u \|\beta^* \|_2^2)
> $$
>
> i.e., it pulls inward only by a cubic drift $\propto\|\beta\|^3$ plus a tiny $\eta^2$ term. Equating that drift with the sample fluctuation of a step, $\sim\|\theta\| \sqrt{d \log ^2(n / \delta) / n}$, gives $\|\beta\|^2 \asymp \sqrt{d \log ^2(n / \delta) / n}$ and hence the $n^{-1 / 4}$ radius:
> $$
> || \hat{\beta}-\beta^{*} || \lesssim (d \log ^2(n / \delta) / n )^{1 / 4}$$
> This is proved in the proof of Theorem A.3 (low-SNR contraction) and the localization argument.
>
> That threshold is exactly where the strong-convexity-like term $\eta^2$ in the population contraction starts to dominate the sampling error; below it, the cubic drift controls progress and you get $n^{-1 / 4}$, above it you recover the $\sqrt{n}$-rate. The two bounds meet smoothly at the threshold (plugging $\eta= \left(d \log ^2 / n\right)^{1 / 4}$ into the high-SNR bound gives the same $n^{-1 / 4}$ order), so the "jump" is just where the better of two valid bounds becomes dominant.
>
> - **W3** Gradient flow analysis should correspond to a continuous-time algorithm, it would be nice to have more discussion and insights on the discrete-time algorithm design and analysis, otherwise it is unclear how directly the continuous-time results translate to realistic architectures.
>
> Consider minimizing $L_{\mathrm{SA}}(E, \theta)$ proposed in section 4.2, where $\theta$ is the collection of the parameter. The gradient descent gives
> $$
> \theta_{t+1}=\theta_{t}-\epsilon \nabla L_{\mathrm{SA}}(E, \theta_{t})
> $$
> and this could simplified as
> $$
> \frac{\theta_{t+1}-\theta_{t}}{\epsilon}=-\nabla L_{\mathrm{SA}}(E, \theta_{t})
> $$
> Letting $\epsilon \rightarrow 0$ and setting $\theta(t)=\theta_{k}$ at time $t=k \epsilon$, we recognize the left-hand side above as the discrete derivative of $\theta(t)$ at time $t$. Hence, we get a continuous-time ordinary differential equation
> $$
> \frac{d \theta}{d t}=-\nabla L_{\mathrm{SA}}(E ; \theta)
> $$
> In fact, starting from the differential equation, we can view gradient descent on $L_{\mathrm{SA}}(E, \theta)$ as one of the most basic numerical analysis techniques-the forward Euler method-for discretely approximating the solution to ODE. When we running the gradient descent with small step size, the gradient flow analysis more or less captures the behaviors of the gradient descent in that setting. However, with large step size, it is not known completely even for linear regression model.

---

> > ### Author Response · Authors · 2025-08-26
> >
> > - I think it would be more convincing if authors can add some experiment to show that the Transformer actually execute gradient-EM, for example, construct a real Transformer following your theory, and plot the errors of prediction of $\beta$ versus low/high SNR as shown in your Theorem 3.1/3.2.
> >
> > Based on our experiments, the trained transformer are less sensitive to initialization, which suggests that they are not learning the EM algorithm.
> >
> > - Both Theorem 3.1 and 3.2 require the prompt length $n$ to be much larger than the input dimension $d$, and the model size $D$ is a multiple of $d$ (in experiments $D=64=2 d$, also $n=64$ ), so possibly the theory may not work for the overparameterized case ($D$ is very large), can you comment on it? Also $n=64$ means the setting $n \gg d$ may not be satisfied in your experiments, so I suggest authors consider adjusting the experimental setup or providing justification to better align with the theoretical assumptions.
> >
> > The assumptions regarding $n$ and $d$ is for theoretical constructions. It is not related to the trained transformer in practice. The performance of the trained transformer will be better than the constructed transformers in Theorem 3.1 and 3.2.

---

### Decision · Action_Editor_Pc6Y · 2025-10-04

**Recommendation:** Accept as is

**Audience:**

Yes

**Audience Explanation:**

Researchers across the community are increasingly interested in the theoretical abilities of transformers to learn diverse models

**Claims And Evidence:**

Yes

**Claims Explanation:**

This paper makes a significant contribution to the theoretical understanding of in-context learning (ICL) in transformers by analyzing its ability to learn mixture of linear regression tasks.  It provides a constructive proof that transformers can internally implement the EM algorithm for MoR, establishing their representational power. Then, it derives statistical guarantees and excess risk bounds across low- and high-SNR regimes, quantifying how dimension and attention depth impact generalization. Finally, it analyzes optimization dynamics, showing convergence to global optima under gradient flow, and further validates these findings through empirical comparisons.  While reviewers mentioned some gaps between the theory and experiments, the paper nevertheless significantly expands the rigorous line of work at the intersection of ICL theory and mixture models. It combines constructive representation results, nontrivial statistical bounds, and training dynamics analysis, which  represents a meaningful step forward in grounding ICL in theory.